# Understanding Deflation Process in Over-parametrized Tensor Decomposition

**Rong Ge**[*]
Duke University
rongge@cs.duke.edu

**Yunwei Ren**[*]
Shanghai Jiao Tong University
2016renyunwei@sjtu.edu.cn

**Xiang Wang**[*]
Duke University
xwang@cs.duke.edu

**Mo Zhou**[*]
Duke University
mozhou@cs.duke.edu

## Abstract

In this paper we study the training dynamics for gradient flow on over-parametrized tensor decomposition problems. Empirically, such training process often first fits larger components and then discovers smaller components, which is similar to a tensor deflation process that is commonly used in tensor decomposition algorithms. We prove that for orthogonally decomposable tensor, a slightly modified version of gradient flow would follow a tensor deflation process and recover all the tensor components. Our proof suggests that for orthogonal tensors, gradient flow dynamics works similarly as greedy low-rank learning in the matrix setting, which is a first step towards understanding the implicit regularization effect of over-parametrized models for low-rank tensors.

## 1 Introduction

Recently, over-parametrization has been recognized as a key feature of neural network optimization. A line of works known as the Neural Tangent Kernel (NTK) showed that it is possible to achieve zero training loss when the network is sufficiently over-parametrized (Jacot et al., 2018; Du et al., 2018; Allen-Zhu et al., 2018b). However, the theory of NTK implies a particular dynamics called lazy training where the neurons do not move much (Chizat et al., 2019), which is not natural in many settings and can lead to worse generalization performance (Arora et al., 2019b). Many works explored other regimes of over-parametrization (Chizat and Bach, 2018; Mei et al., 2018) and analyzed dynamics beyond lazy training (Allen-Zhu et al., 2018a; Li et al., 2020a; Wang et al., 2020).

Over-parametrization does not only help neural network models. In this work, we focus on a closely related problem of tensor (CP) decomposition. In this problem, we are given a tensor of the form

$$T^* = \sum_{i=1}^{r} a_i (U[:, i])^{\otimes 4},$$

where $a_i \geq 0$ and $U[:, i]$ is the $i$-th column of $U \in \mathbb{R}^{d \times r}$. The goal is to fit $T^*$ using a tensor $T$ of a similar form:

$$T = \sum_{i=1}^{m} \frac{(W[:, i])^{\otimes 4}}{\|W[:, i]\|^2}.$$

---

[*]Alphabetical order.

35th Conference on Neural Information Processing Systems (NeurIPS 2021).

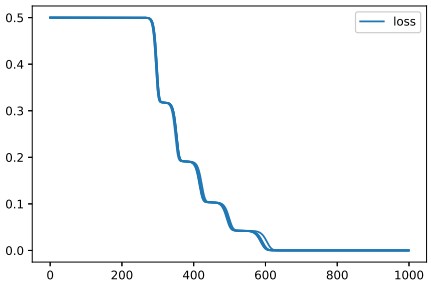 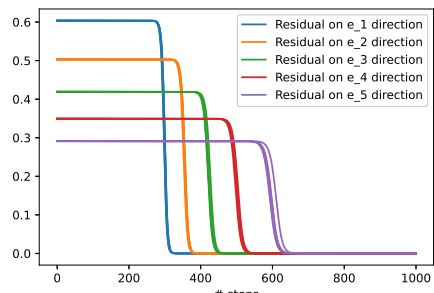

Figure 1: The training trajectory of gradient flow on orthogonal tensor decompositions. We chose $T^* = \sum_{i \in [5]} a_i e_i^{\otimes 4}$ with $e_i \in \mathbb{R}^{10}$ and $a_i/a_{i+1} = 1.2$. Our model $T$ has 50 components and each component is randomly initialized with small norm $10^{-15}$. We ran the experiments from 5 different initialization and plotted the results separately. The left figure shows the loss $\frac{1}{2} \|T - T^*\|_F^2$ and the right figure shows the residual on each $e_i$ direction that is defined as $(T^* - T)(e_i^{\otimes 4})$.

Here $W$ is a $d \times m$ matrix whose columns are components for tensor $T$. The model is over-parametrized when the number of components $m$ is larger than $r$. The choice of normalization factor of $1/\|W[:,i]\|^2$ is made to accelerate gradient flow (similar to Li et al. (2020a); Wang et al. (2020)).

Suppose we run gradient flow on the standard objective $\frac{1}{2}\|T - T^*\|_F^2$, that is, we evolve $W$ according to the differential equation:

$$\frac{\mathrm{d}W}{\mathrm{d}t} = -\nabla \left( \frac{1}{2} \|T - T^*\|_F^2 \right),$$

can we expect $T$ to fit $T^*$ with good accuracy? Empirical results (see Figure 1) show that this is true for orthogonal tensor $T^*$[2] as long as $m$ is large enough. Further, the training dynamics exhibits a behavior that is similar to a *tensor deflation process*: it finds the ground truth components one-by-one from larger component to smaller component (if multiple ground truth components have similar norm they might be found simultaneously).

In this paper we show that with a slight modification, gradient flow on over-parametrized tensor decomposition is guaranteed to follow this tensor deflation process, and can fit any orthogonal tensor to desired accuracy[3](see Section 4 for the algorithm and Theorem 1 for the main theorem). This shows that for orthogonal tensors, the trajectory of modified gradient-flow is similar to a greedy low-rank process that was used to analyze the implicit bias of low-rank matrix factorization (Li et al., 2020b). We emphasize that our goal is not to propose another tensor decomposition algorithm. Instead, we hope our results can serve as a first step in understanding the implicit bias of over-parameterized gradient descent for low-rank tensor problems.

## 1.1 Our approach and technique

To understand the tensor deflation process shown in Figure 1, intuitively we can think about the discovery and fitting of a ground truth component in two phases. Consider the beginning of the gradient flow as an example. Initially all the components in $T$ are small, which makes $T$ negligible compared to $T^*$. In this case each component $w$ in $W$ will evolve according to a simpler dynamics that is similar to tensor power method, where one updates $w$ to $T^*(w^{\otimes 3}, I)/\|T^*(w^{\otimes 3}, I)\|$ (see Section 3 for details).

For orthogonal tensors, it's known that tensor power method with random initializations would be able to discover the largest ground truth components (see Anandkumar et al. (2014)). Once the largest ground truth component has been discovered, the corresponding component (or multiple components) $w$ will quickly grow in norm, which eventually fits the ground truth component. The flat regions in the trajectory in Figure 1 correspond to the period of time where the components $w$'s are small and

---

[2]We say $T^*$ is an orthogonal tensor if the ground truth components $U[:,i]$'s are orthonormal.

[3]Due to some technical challenges, we actually require the target accuracy to be at least $\exp(-o(d/\log d))$. This is only a very mild restriction since the dependence is exponential in $d$, and in practice, $d$ is usually large and this lower bound can easily drop below the numerical precision.

$T - T^*$ remains stable, while the decreasing regions correspond to the period of time where a ground truth component is being fitted.

However, there are many challenges in analyzing this process. The main problem is that the gradient flow would introduce a lot of dependencies throughout the trajectory, making it harder to analyze the fitting of later ground truth components, especially ones that are much smaller. We modify the algorithm to include a reinitialization step per epoch, which alleviates the dependency issue. Even after the modification we still need a few more techniques:

**Local stability**  One major problem in analyzing the dynamics in a later stage is that the components used to fit the previous ground truth components are still moving according to their gradients, therefore it might be possible for these components to move away. To address this problem, we add a small regularizer to the objective, and give a new local stability analysis that bounds the distance to the fitted ground truth component both individually and on average. The idea of bounding the distance on average is important as just assuming each component $w$ is close enough to the fitted ground truth component is not sufficient to prove that $w$ cannot move far. While similar ideas were considered in Chizat (2021), the setting of tensor decomposition is different.

**Norm/Correlation relation**  A key step in our analysis establishes a relationship between norm and correlation: we show if a component $w$ crosses a certain norm threshold, then it must have a very large correlation with one of the ground truth components. This offers an initial condition for local stability and makes sure the residual $T^* - T$ is almost close to an orthogonal tensor. Establishing this relation is difficult as unlike the high level intuition, we cannot guarantee $T^* - T$ remains unchanged even within a single epoch: it is possible that one ground truth component is already fitted while no large component is near another ground truth component of same size. In previous work, Li et al. (2020a) deals with a similar problem for neural networks using gradient truncation that prevents components from growing in the first phase (and as a result has super-exponential dependency on the ratio between largest and smallest $a_i$). We give a new technique to control the influence of ground truth components that are fitted within this epoch, so we do not need the gradient truncation and can characterize the deflation process.

## 1.2 Related works

**Neural Tangent Kernel**  There is a recent line of work showing the connection between Neural Tangent Kernel (NTK) and sufficiently wide neural networks trained by gradient descent (Jacot et al., 2018; Allen-Zhu et al., 2018b; Du et al., 2018, 2019; Li and Liang, 2018; Arora et al., 2019b,c; Zou et al., 2020; Oymak and Soltanolkotabi, 2020; Ghorbani et al., 2021). These papers show when the width of a neural network is large enough, it will stay around the initialization and its training dynamic is close to the dynamic of the kernel regression with NTK. In this paper we go beyond the NTK setting and analyze the trajectory from a very small initialization.

**Mean-field analysis**  There is another line of works that use mean-field approach to study the optimization for infinite-wide neural networks (Mei et al., 2018; Chizat and Bach, 2018; Nguyen and Pham, 2020; Nitanda and Suzuki, 2017; Wei et al., 2019; Rotskoff and Vanden-Eijnden, 2018; Sirignano and Spiliopoulos, 2020). Chizat et al. (2019) showed that, unlike NTK regime, the parameters can move away from its initialization in mean-field regime. However, most of the existing works need width to be exponential in dimension and do not provide a polynomial convergence rate.

**Beyond NTK**  There are many works showing the gap between neural networks and NTK (Allen-Zhu and Li, 2019; Allen-Zhu et al., 2018a; Yehudai and Shamir, 2019; Ghorbani et al., 2019, 2020; Dyer and Gur-Ari, 2019; Woodworth et al., 2020; Bai and Lee, 2019; Bai et al., 2020; Huang and Yau, 2020; Chen et al., 2020). In particular, Li et al. (2020a) and Wang et al. (2020) are closely related with our setting. While Li et al. (2020a) focused on learning two-layer ReLU neural networks with orthogonal weights, they relied on the connection between tensor decomposition and neural networks (Ge et al., 2017) and essentially worked with tensor decomposition problems. In their result, all the $a_i$'s are within a constant factor and all components are learned simultaneously. We allow ground truth components with very different scale and show a deflation phenomenon. Wang et al. (2020) studied learning a low-rank non-orthogonal tensor, but they only showed the learned tensor $T$ will eventually be close to the ground truth tensor $T^*$ and does not guarantee the components of $T$ will

align with the components of $T^*$. On the other hand, we fully characterize the training trajectory and the components of the learned tensor.

**Implicit regularization**    Many works recently showed that different optimization methods tend to converge to different optima and have different optimization trajectories in several settings (Saxe et al., 2014; Soudry et al., 2018; Nacson et al., 2019; Ji and Telgarsky, 2018a,b, 2019, 2020; Gunasekar et al., 2018a,b; Moroshko et al., 2020; Arora et al., 2019a; Lyu and Li, 2019; Chizat and Bach, 2020). In particular, Saxe et al. (2014) related the dynamics of gradient descent to the magnitude of the singular values of the target weight matrices for linear networks with orthogonal inputs. The phenomenon there is qualitatively similar to our results, but the settings and the proof techniques are very different. The more related and recent works are Li et al. (2020b) and Razin et al. (2021). Li et al. (2020b) studied matrix factorization problem and showed gradient descent with infinitesimal initialization is similar to greedy low-rank learning, which is a multi-epoch algorithm that finds the best approximation within the rank constraint and relax the constraint after every epoch. Razin et al. (2021) studied the tensor factorization problem and showed that it biases towards low rank tensor. Both of these works considered partially observable matrix or tensor and are only able to fully analyze the first epoch (i.e., recover the largest direction). We focus on a simpler setting with fully-observable ground truth tensor and give a complete analysis of learning all the ground truth components.

### 1.3   Outline

In Section 2 we introduce the basic notations and problem setup. In Section 3 we review tensor deflation process and tensor power method. We then give our algorithm in Section 4. Section 5 gives the formal main theorem and discusses high-level proof ideas. We conclude in Section 6 and discuss some limitations of the work. The detailed proofs and additional experiments are left in the appendix.

## 2   Preliminaries

**Notations**    We use upper-case letters to denote matrices and tensors, and lower-case letters to denote vectors. For any positive integer $n$, we use $[n]$ to denote the set $\{1, 2, \cdots, n\}$. We use $I_d$ to denote $d \times d$ identity matrix, and omit the subscript $d$ when the dimension is clear. We use $\delta_0 \mathrm{Unif}(\mathbb{S}^{d-1})$ to denote the uniform distribution over $(d-1)$-dimensional sphere with radius $\delta_0$.

For vector $v$, we use $\|v\|$ to denote its $\ell_2$ norm. We use $v_k$ to denote the $k$-th entry of vector $v$, and use $v_{-k}$ to denote vector $v$ with its $k$-th entry removed. We use $\bar{v}$ to denote the normalized vector $\bar{v} = v/\|v\|$, and use $\bar{v}_k$ to denote the $k$-th entry of $\bar{v}$.

For a matrix $A$, we use $A[:, i]$ to denote its $i$-th column and $\mathrm{col}(A)$ to denote the set of all column vectors of $A$. For matrix $M$ or tensor $T$, we use $\|M\|_F$ and $\|T\|_F$ to denote their Frobenius norm, which is equal to the $\ell_2$ norm of their vectorization.

For simplicity we restrict our attention to symmetric 4-th order tensors. For a vector $v \in \mathbb{R}^d$, we use $v^{\otimes 4}$ to denote a $d \times d \times d \times d$ tensor whose $(i, j, k, l)$-th entry is equal to $v_i v_j v_k v_l$. Suppose $T = \sum_w w^{\otimes 4}$, we define $T(v^{\otimes 4})$ as $\sum_w \langle w, v \rangle^4$, $T(v^{\otimes 3}, I)$ as $\sum_w \langle w, v \rangle^3 w$, and $T(v^{\otimes 2}, u, I) = \sum_w \langle w, v \rangle^2 \langle w, u \rangle w$.

For clarity, we always call a component in $T^*$ as ground truth component and call a component in our model $T$ simply as component.

**Problem setup**    We consider the problem of fitting a 4-th order tensor. The components of the ground truth tensor is arranged as columns of a matrix $U \in \mathbb{R}^{d \times r}$, and the tensor $T^*$ is defined as

$$T^* = \sum_{i=1}^{r} a_i (U[:, i]^{\otimes 4}),$$

where $a_1 \geq a_2 \geq \cdots \geq a_r \geq 0$ and $\sum_{i=1}^{r} a_i = 1$. For convenience in the analysis, we assume $a_i \geq \epsilon/\sqrt{d}$ for all $i \in [r]$. This is without loss of generality because the target accuracy is $\epsilon$ and we can safely ignore very small ground truth components with $a_i < \epsilon/\sqrt{d}$. In this paper, we focus on the case where the components are orthogonal—that is, the columns $U[:, i]$'s are orthonormal. For

simplicity we assume without loss of generality that $U[:, i] = e_i$ where $e_i$ is the $i$-th standard basis vector[4]. To reduce the number of parameters we also assume $r = d$, again this is without loss of generality because we can simply set $a_i = 0$ for $i > r$.

There can be many different ways to parametrize the tensor that we use to fit $T^*$. Following previous works (Wang et al., 2020; Li et al., 2020a), we use an over-parameterized and two-homogeneous tensor

$$T = \sum_{i=1}^{m} \frac{W[:, i]^{\otimes 4}}{\|W[:, i]\|^2}.$$

Here $W \in \mathbb{R}^{d \times m}$ is a matrix with $m$ columns that corresponds to the components in $T$. It is overparametrized when $m > r$.

Since the tensor $T$ only depends on the set of columns $W[:, i]$ instead of the orderings of the columns, for the most part of the paper we will instead write the tensor $T$ as

$$T = \sum_{w \in \mathrm{col}(W)} \frac{w^{\otimes 4}}{\|w\|^2},$$

where $\mathrm{col}(W)$ is the set of all the column vectors in $W$. This allows us to discuss the dynamics of coordinates for a component $w$ without using the index for the component. In particular, $w_i$ always represents the $i$-th coordinate of the vector $w$. This representation is similar to the mean-field setup (Chizat and Bach, 2018; Mei et al., 2018) where one considers a distribution on $w$, however since we do not rely on analysis related to infinite-width limit we use the sum formulation instead. For the ease of presentation, we choose to restrict our setting to fourth-order tensor decomposition, but our results can be easily generalized to tensor with order at least three.

## 3 Tensor deflation process and tensor power method

In this section we will first discuss the basic tensor deflation process for orthogonal tensor decomposition. Then we show the connection between the tensor power method and gradient flow.

**Tensor deflation**   For orthogonal tensor decomposition, a popular approach is to first fit the largest ground truth component in the tensor, then subtract it out and recurse on the residual. The general process is given in Algorithm 1. In this process, there are multiple ways to find the best rank-1 approximation. For example, Anandkumar et al. (2014) uses tensor power method, which picks many random vectors $w$, and update them as $w = T^*(w^{\otimes 3}, I) / \|T^*(w^{\otimes 3}, I)\|$.

---

**Algorithm 1** Tensor Deflation Process

---
   **Input:** Tensor $T^*$
   **Output:** Components $W$ such that $T^* \approx \sum_{w \in \mathrm{col}(W)} w^{\otimes 4}/\|w\|^2$
   Initially let the residual $R$ be $T^*$.
   **while** $\|R\|_F$ is large **do**
      Find the best rank 1 approximation $w^{\otimes 4}/\|w\|^2$ for $R$.
      Add $w$ as a new column in $W$, and let $R = R - w^{\otimes 4}/\|w\|^2$.
   **end while**

---

**Tensor power method and gradient flow**   If we run tensor power method using a tensor $T^*$ that is equal to $\sum_{i=1}^{d} a_i e_i^{\otimes 4}$, then a component $w$ will converge to the direction of $e_i$ where $i$ is equal to $\arg\max_i a_i \bar{w}_i^2$. If there is a tie (which happens with probability 0 for random $w$), then the point will be stuck at a saddle point.

Let's consider running gradient flow on $W$ with objective function $\frac{1}{2}\|T - T^*\|_F^2$ as $T := \sum_{w \in \mathrm{col}(W)} w^{\otimes 4}/\|w\|^2$. If $T$ does not change much, the residual $R := T^* - T$ is close to a

---

[4]This is without loss of generality because gradient flow (and our modifications) is invariant under rotation of the ground truth parameters.

constant. In this case the trajectory of one component $w$ is determined by the following differential equation:

$$\frac{\mathrm{d}w}{\mathrm{d}t} = 4R(\bar{w}^{\otimes 2}, w, I) - 2R(\bar{w}^{\otimes 4})w. \tag{1}$$

To understand how this process works, we can take a look at $\frac{\mathrm{d}w_i^2/\mathrm{d}t}{w_i^2}$ (intuitively this corresponds to the growth rate for $w_i^2$). If $R \approx T^*$ then we have:

$$\frac{\mathrm{d}w_i^2/\mathrm{d}t}{w_i^2} \approx 8a_i\bar{w}_i^2 - 4\sum_{j\in[d]} a_j\bar{w}_j^4.$$

From this formula it is clear that the coordinate with larger $a_i\bar{w}_i^2$ has a faster growth rate, so eventually the process will converge to $e_i$ where $i$ is equal to $\arg\max_i a_i\bar{w}_i^2$, same as the tensor power method. Because of their similarity later we refer to dynamics in Eqn. (1) as tensor power dynamics.

## 4 Our algorithm

Our algorithm is a modified version of gradient flow as described in Algorithm 2. First, we change the loss function to

$$L(W) = \frac{1}{2}\|T - T^*\|_F^2 + \frac{\lambda}{2}\|W\|_F^2.$$

The additional small regularization $\frac{\lambda}{2}\|W\|_F^2$ allows us to prove a *local stability* result that shows if there are components $w$ that are close to the ground truth components in direction, then they will not move too much (see Section 5.1).

Our algorithm runs in multiple epochs with increasing length. We use $W^{(s,t)}$ to denote the weight matrix in epoch $s$ at time $t$. We use similar notation for tensor $T^{(s,t)}$. In each epoch we try to fit ground truth components with $a_i \geq \beta^{(s)}$. In general, the time it takes to fit one ground truth direction is inversely proportional to its magnitude $a_i$. The earlier epochs have shorter length so only large directions can be fitted, and later epochs are longer to fit small directions.

At the middle of each epoch, we reinitialize all components that do not have a large norm. This serves several purposes: first we will show that all components that exceed the norm threshold will have good correlation with one of the ground truth components, therefore giving an initial condition to the local stability result; second, the reinitialization will reduce the dependencies between different epochs and allow us to analyze each epoch almost independently. These modifications do not change the dynamics significantly, however they allow us to do a rigorous analysis.

---

**Algorithm 2** Modified Gradient Flow

---

**Input:** Number of components $m$, initialization scale $\delta_0$, re-initialization threshold $\delta_1$, increasing rate of epoch length $\gamma$, target accuracy $\epsilon$, regularization coefficient $\lambda$

**Output:** Tensor $T$ satisfying $\|T - T^*\|_F \leq \epsilon$.

Initialize $W^{(0,0)}$ as a $d \times m$ matrix with each column $w^{(0,0)}$ i.i.d. sampled from $\delta_0\mathrm{Unif}(\mathbb{S}^{d-1})$.

$\beta^{(0)} \leftarrow \|T^{(0,0)} - T^*\|_F$; $s \leftarrow 0$

**while** $\|T^{(s,0)} - T^*\|_F > \epsilon$ **do**

    Phase 1: Starting from $W^{(s,0)}$, run gradient flow for time $t_1^{(s)} = O(\frac{d}{\beta^{(s)}\log(d)})$.

    Reinitialize all components that have $\ell_2$ norm less than $\delta_1$ by sampling i.i.d. from $\delta_0\mathrm{Unif}(\mathbb{S}^{d-1})$.

    Phase 2: Starting from $W^{(s,t_1^{(s)})}$, run gradient flow for $t_2^{(s)} - t_1^{(s)} = O(\frac{\log(1/\delta_1)+\log(1/\lambda)}{\beta^{(s)}})$ time

    $W^{(s+1,0)} \leftarrow W^{(s,t_2^{(s)})}$; $\beta^{(s+1)} \leftarrow \beta^{(s)}(1-\gamma)$; $s \leftarrow s+1$

**end while**

---

## 5    Main theorem and proof sketch

In this section we discuss the ideas to prove the following main theorem[5]

**Theorem 1.** *For any $\epsilon \geq \exp(-o(d/\log d))$, there exists $\gamma = \Theta(1)$, $m = poly(d)$, $\lambda = \min\{O(\log d/d), O(\epsilon/d^{1/2})\}$, $\alpha = \min\{O(\lambda/d^{3/2}), O(\lambda^2), O(\epsilon^2/d^4)\}$, $\delta_1 = O(\alpha^{3/2}/m^{1/2})$, $\delta_0 = \Theta(\delta_1\alpha/\log^{1/2}(d))$ such that with probability $1 - 1/poly(d)$ in the (re)-initializations, Algorithm 2 terminates in $O(\log(d/\epsilon))$ epochs and returns a tensor $T$ such that*

$$\|T - T^*\|_F \leq \epsilon.$$

Intuitively, epoch $s$ of Algorithm 2 will try to discover all ground truth components with $a_i$ that is at least as large as $\beta^{(s)}$. The algorithm does this in two phases. In Phase 1, the small components $w$ will evolve according to tensor power dynamics. For each ground truth component with large enough $a_i$ that has not been fitted yet, we hope there will be at least one component in $W$ that becomes large and correlated with $e_i$. We call such ground truth components "discovered". Phase 1 ends with a check that reinitializes all components with small norm. Phase 2 is relatively short, and in Phase 2 we guarantee that every ground truth component that has been discovered become "fitted", which means the residual $T - T^*$ becomes small in this direction.

However, there are still many difficulties in analyzing each of the steps. In particular, why would ground truth components that are fitted in previous epochs remain fitted? How to guarantee only components that are correlated with a ground truth component grow to a large norm? Why wouldn't the gradient flow in Phase 2 mess up with the initialization we require in Phase 1? We discuss the high level ideas to solve these issues. In particular, in Section 5.1 we first give an induction hypothesis that is preserved throughout the algorithm, which guarantees that every ground truth component that is fitted remains fitted. In Section 5.2 we discuss the properties in Phase 1, and in Section 5.3 we discuss the properties in Phase 2.

### 5.1    Induction hypothesis and local stability

In order to formally define what it means for a ground truth component to be "discovered" or "fitted", we need some more definitions and notations.

**Definition 1.** *Define $S_i^{(s,t)} \subseteq [m]$ as the subset of components that satisfy the following conditions: the $k$-th component is in $S_i^{(s,t)}$ if and only if there exists some time $(s', t')$ that is no later than $(s, t)$ and no earlier than the latest re-initialization of $W[:, k]$ such that*

$$\left\|W^{(s',t')}[:, k]\right\| = \delta_1 \text{ and } \left[\overline{W^{(s',t')}[:, k]}_i\right]^2 \geq 1 - \alpha^2.$$

*We say that ground truth component $i$ is* discovered *in epoch $s$ at time $t$, if $S_i^{(s,t)}$ is not empty.*

Intuitively, $S_i^{(s,t)}$ is a subset of components in $W$ such that they have large enough norm and good correlation with the $i$-th ground truth component. Although such components may not have a large enough norm to fit $a_i$ yet, their norm will eventually grow. Therefore we say ground truth component $i$ is discovered when such components exist.

For convenience, we shorthand $w^{(s,t)} \in \{W^{(s,t)}[:, j] | j \in S_i^{(s,t)}\}$ by $w^{(s,t)} \in S_i^{(s,t)}$. Now we will discuss when a ground truth component is fitted, for that, let

$$\hat{a}_i^{(s,t)} = \sum_{w^{(s,t)} \in S_i^{(s,t)}} \left\|w^{(s,t)}\right\|^2.$$

Here $\hat{a}_i^{(s,t)}$ is the total squared norm for all the components in $S_i^{(s,t)}$. We say a ground truth component is *fitted* if $a_i - \hat{a}_i^{(s,t)} \leq 2\lambda$.

---

[5]In the theorem statement, we have a parameter $\alpha$ that is not used in our algorithm but is very useful in the analysis (see for example Definition 1). Basically, $\alpha$ measures the closeness between a component and its corresponding ground truth direction (see more in Section 5.1).

Note that one can partition the columns in $W$ using sets $S_i^{(s,t)}$, giving $d$ groups and one extra group that contains everything else. We define the extra group as $S_\varnothing^{(s,t)} := [m] \setminus \bigcup_{k \in [d]} S_k^{(s,t)}$.

For each of the non-empty $S_i^{(s,t)}$, we can take the average of its component (weighted by $\left\|w^{(s,t)}\right\|^2$):

$$\mathbb{E}_{i,w}^{(s,t)} f(w^{(s,t)}) := \frac{1}{\hat{a}_i^{(s,t)}} \sum_{w^{(s,t)} \in S_i^{(s,t)}} \left\|w^{(s,t)}\right\|^2 f(w^{(s,t)}).$$

If $S_i^{(s,t)} = \varnothing$, we define $\mathbb{E}_{i,w}^{(s,t)} f(w^{(s,t)})$ as zero. Now we are ready to state the induction hypothesis:

**Proposition 1** (Induction hypothesis). *In the setting of Theorem 1, for any epoch $s$ and time $t$ and every $k \in [d]$, the following hold.*

(a) *For any $w^{(s,t)} \in S_k^{(s,t)}$, we have $\left[\bar{w}_k^{(s,t)}\right]^2 \geq 1 - \alpha$.*

(b) *If $S_k^{(s,t)}$ is nonempty, $\mathbb{E}_{k,w}^{(s,t)} \left[\bar{w}_k^{(s,t)}\right]^2 \geq 1 - \alpha^2 - 4sm\delta_1^2$.*

(c) *We always have $a_k - \hat{a}_k^{(s,t)} \geq \lambda/6 - sm\delta_1^2$; if $a_k \geq \frac{\beta^{(s)}}{1-\gamma}$, we further know $a_k - \hat{a}_k^{(s,t)} \leq \lambda + sm\delta_1^2$.*

(d) *If $w^{(s,t)} \in S_\varnothing^{(s,t)}$, then $\|w^{(s,t)}\| \leq \delta_1$.*

We choose $\delta_1^2$ small enough so that $sm\delta_1^2$ is negligible compared with $\alpha^2$ and $\lambda$. Note that if Proposition 1 is maintained throughout the algorithm, all the large components will be fitted, which directly implies Theorem 1. Detailed proof is deferred to Appendix D.

Condition (c) shows that for a ground truth component $k$ with large enough $a_k$, it will always be fitted after the corresponding epoch (recall from Theorem 1 that $\lambda = O(\varepsilon/\sqrt{d})$). Condition (d) shows that components that did not discover any ground truth components will always have small norm (hence negligible in most parts of the analysis). Conditions (a)(b) show that as long as a ground truth component $k$ has been discovered, all components that are in $S_k^{(s,t)}$ will have good correlation, while the *average* of all such components will have even better correlation. The separation between individual correlation and average correlation is important in the proof. With only individual bound, we cannot maintain the correlation no matter how small $\alpha$ is. Here is an example below:

**Claim 2.** *Suppose $T^* = e_k^{\otimes 4}$ and $T = v^{\otimes 4}/\|v\|^2 + w^{\otimes 4}/\|w\|^2$ with $\|w\|^2 + \|v\|^2 \in [2/3, 1]$. Suppose $\bar{v}_k^2 = 1 - \alpha$ and $\bar{v}_k = \bar{w}_k, \bar{v}_{-k} = -\bar{w}_{-k}$. Assuming $\|v\|^2 \leq c_1$ and $\alpha \leq c_2$ for small enough constants $c_1, c_2$, we have $\frac{d}{dt}\bar{v}_k^2 < 0$.*

In the above example, both $\bar{v}$ and $\bar{w}$ are close to $e_k$ but they are opposite in other directions ($\bar{v}_{-k} = \bar{w}_{-k}$). The norm of $v$ is very small compared with that of $w$. Intuitively, we can increase $v_{-k}$ so that the average of $v$ and $w$ is more aligned with $e_k$. See the rigorous analysis in Appendix A.6.

The induction hypothesis will be carefully maintained throughout the analysis. The following lemma guarantees that in the gradient flow steps the individual and average correlation will be maintained.

**Lemma 3.** *In the setting of Theorem 1, suppose Proposition 1 holds in epoch $s$ at time $t$, we have*

$$\frac{d}{dt}[\bar{w}^{(s,t)}]^2 \geq 8\left(a_k - \hat{a}_k^{(s,t)}\right)\left(1 - [\bar{w}_k^{(s,t)}]^2\right) - O\left(\alpha^{1.5}\right),$$

$$\frac{d}{dt}\mathbb{E}_{k,w}^{(s,t)}[\bar{w}_k^{(s,t)}]^2 \geq 8\left(a_k - \hat{a}_k^{(s,t)}\right)\left(1 - \mathbb{E}_{k,w}^{(s,t)}[\bar{w}_k^{(s,t)}]^2\right) - O(\alpha^3).$$

*In particular, when $a_k - \hat{a}_k^{(s,t)} \geq \Omega(\lambda) = \Omega(\sqrt{\alpha})$, we have $\frac{d}{dt}[\bar{w}_k^{(s,t)}]^2 > 0$ when $[\bar{w}_k^{(s,t)}]^2 = 1 - \alpha$ and $\frac{d}{dt}\mathbb{E}_{k,w}^{(s,t)}[\bar{w}_k^{(s,t)}]^2 > 0$ when $\mathbb{E}_{k,w}^{(s,t)}[\bar{w}_k^{(s,t)}]^2 = 1 - \alpha^2$.*

The detailed proof for the local stability result can be found in Appendix A. Of course, to fully prove the induction hypothesis one needs to talk about what happens when a component enters $S_i^{(s,t)}$, and what happens at the reinitialization steps. We discuss these details in later subsections.

## 5.2 Analysis of Phase 1

In Phase 1 our main goal is to discover all the components that are large enough. We also need to maintain Proposition 1. Formally we prove the following:

**Lemma 4** (Main Lemma for Phase 1). *In the setting of Theorem 1, suppose Proposition 1 holds at $(s, 0)$. For $t_1^{(s)} := t_1^{(s)'} + t_1^{(s)''} + t_1^{(s)'''}$ with $t_1^{(s)'} = \Theta(d/(\beta^{(s)} \log d))$, $t_1^{(s)''} = \Theta(d/(\beta^{(s)} \log^3 d))$, $t_1^{(s)'''} = \Theta(\log(d/\alpha)/\beta^{(s)})$, with probability $1 - 1/poly(d)$ we have*

1. *Proposition 1 holds at $(s, t)$ for any $0 \leq t < t_1^{(s)}$, and also for $t = t_1^{(s)}$ after reinitialization.*

2. *If $a_k \geq \beta^{(s)}$ and $S_k^{(s,0)} = \varnothing$, we have $S_k^{(s,t_1^{(s)})} \neq \varnothing$ and $\hat{a}_k^{(s,t_1^{(s)})} \geq \delta_1^2$.*

3. *If $S_k^{(s,0)} = \varnothing$ and $S_k^{(s,t_1^{(s)})} \neq \varnothing$, we have $a_k \geq C\beta^{(s)}$ for universal constant $0 < C < 1$.*

Property 2 shows that large enough ground truth components are always discovered, while Property 3 guarantees that no small ground truth components can be discovered. Our proof relies on initial components being "lucky" and having higher than usual correlation with one of the large ground truth components. To make this clear we separate components into different sets (here we use $v$ to denote a component in $W$):

**Definition 2** (Partition of (re-)initialized components). *For each direction $i \in [d]$, define the set of good components $S_{i,good}^{(s)}$ and the set of potential components $S_{i,pot}^{(s)}$ as follow, where $\Gamma_i^{(s)} := 1/(8a_i t_1^{(s)'})$ if $S_i^{(s,0)} = \varnothing$, and $\Gamma_i^{(s)} := 1/(8\lambda t_1^{(s)'})$ otherwise. Here $\rho_i^{(s)} := c_\rho \Gamma_i^{(s)}$ and $c_\rho$ is a small enough absolute constant.*

$$S_{i,good}^{(s)} := \{k \mid [\bar{v}_i^{(s,0)}]^2 \geq \Gamma_i^{(s)} + \rho_i^{(s)}, [\bar{v}_j^{(s,0)}]^2 \leq \Gamma_j^{(s)} - \rho_j^{(s)}, \forall j \neq i \text{ and } v^{(s,0)} = W^{(s,0)}[:, k]\},$$

$$S_{i,pot}^{(s)} := \{k \mid [\bar{v}_i^{(s,0)}]^2 \geq \Gamma_i^{(s)} - \rho_i^{(s)} \text{ and } v^{(s,0)} = W^{(s,0)}[:, k]\}.$$

*Let $S_{good}^{(s)} := \cup_i S_{i,good}^{(s)}$ and $S_{pot}^{(s)} := \cup_i S_{i,pot}^{(s)}$. We also define the set of bad components $S_{bad}^{(s)}$.*

$$S_{bad}^{(s)} := \{k \mid \exists i \neq j \text{ s.t. } [\bar{v}_i^{(s,0)}]^2 \geq \Gamma_i^{(s)} - \rho_i^{(s)}, [\bar{v}_j^{(s,0)}]^2 \geq \Gamma_j^{(s)} - \rho_j^{(s)} \text{ and } v^{(s,0)} = W^{(s,0)}[:, k]\}.$$

For convenience, we shorthand $v^{(s,t)} \in \{W^{(s,t)}[:, j] \mid j \in S_{i,good}\}$ by $v^{(s,t)} \in S_{i,good}$ (same for $S_{i,pot}$ and $S_{bad}$). Intuitively, the good components will grow very quickly and eventually pass the norm threshold. Since both good and potential components only have one large coordinate, they will become correlated with that ground truth component when their norm is large. The bad components are correlated with two ground truth components so they can potentially have a large norm while not having a very good correlation with either one of them. In the proof we will guarantee with probability at least $1 - 1/poly(d)$ that good components exists for all large enough ground truth components and there are no bad components. The following lemma characterizes the trajectories of different type of components:

**Lemma 5.** *In the setting of Lemma 4, for every $i \in [d]$*

1. *(Only good/potential components can become large) If $v^{(s,t)} \notin S_{pot}^{(s)}$, $\|v^{(s,t)}\| = O(\delta_0)$ and $[\bar{v}_i^{(s,t)}]^2 = O(\log(d)/d)$ for all $i \in [d]$ and $t \leq t_1^{(s)}$.*

2. *(Good components discover ground truth components) If $S_{i,good}^{(s)} \neq \varnothing$, there exists $v^{(s,t_1^{(s)})}$ such that $\left\| v^{(s,t_1^{(s)})} \right\| \geq \delta_1$ and $S_i^{(s,t_1^{(s)})} \neq \varnothing$.*

3. *(Large components are correlated with ground truth components) If $\left\| v^{(s,t)} \right\| \geq \delta_1$ for some $t \leq t_1^{(s)}$, there exists $i \in [d]$ such that $v^{(s,t)} \in S_i^{(s,t)}$.*

The proof of Lemma 5 is difficult as one cannot guarantee that all the ground truth components that we are hoping to fit in the epoch will be fitted simultaneously. However we are able to show that $T - T^*$ remains near-orthogonal and control the effect of changing $T - T^*$ within this epoch. The details are in Appendix B.

### 5.3 Analysis of Phase 2

In Phase 2 we will show that every ground truth component that's discovered in Phase 1 will become fitted, and the reinitialized components will preserve the desired initialization conditions.

**Lemma 6** (Main Lemma for Phase 2). *In the setting of Theorem 1, suppose Proposition 1 holds at $(s, t_1^{(s)})$, we have for $t_2^{(s)} - t_1^{(s)} := O\left(\frac{\log(1/\delta_1) + \log(1/\lambda)}{\beta^{(s)}}\right)$*

1. *Proposition 1 holds at $(s, t)$ for any $t_1^{(s)} \leq t \leq t_2^{(s)}$.*

2. *If $S_k^{(s, t_1^{(s)})} \neq \varnothing$, we have $a_k - \hat{a}_k^{(s, t_2^{(s)})} \leq 2\lambda$.*

3. *For any component $v$ that was reinitialized at $t_1^{(s)}$, we have $\left\| v^{(s, t_2^{(s)})} \right\|^2 = \Theta(\delta_0^2)$ and $\left[ \bar{v}_i^{(s, t_2^{(s)})} \right]^2 = \left[ \bar{v}_i^{(s, t_1^{(s)})} \right]^2 \pm o\left( \frac{\log d}{d} \right)$ for every $i \in [d]$.*

The main idea is that as long as a direction has been discovered, the norm of the corresponding components will increase very fast. The rate of that is characterized by the following lemma.

**Lemma 7** (informal). *In the setting of Theorem 6, for any $t_1^{(s)} \leq t \leq t_2^{(s)}$,*

$$\frac{\mathrm{d}}{\mathrm{d}t} \hat{a}_k^{(s,t)} \geq \left( 2(a_k - \hat{a}_k^{(s,t)}) - \lambda - O\left(\alpha^2\right) \right) \hat{a}_k^{(s,t)}.$$

*In particular, after $O\left(\frac{\log(1/\delta_1) + \log(1/\lambda)}{a_k}\right)$ time, we have $a_k - \hat{a}_k^{(s,t)} \leq \lambda$.*

By the choice of $\delta_1$ and $\lambda$, the length of Phase 2 is much smaller than the amount of time needed for the reinitialized components to move far, allowing us to prove the third property in Lemma 6. Detailed analysis is deferred to Appendix C.

## 6 Conclusion

In this paper we analyzed the dynamics of gradient flow for over-parametrized orthogonal tensor decomposition. With very mild modification to the algorithm (a small regularizer and some re-initializations), we showed that the trajectory is similar to a tensor deflation process and the greedy low-rank procedure in Li et al. (2020b). These modifications allowed us to prove strong guarantees for orthogonal tensors of any rank, while not changing the empirical behavior of the algorithm. We believe such techniques would be useful in later analysis for the implicit bias of tensor problems.

A major limitation of our work is that it only applies to orthogonal tensors. Going beyond this would require significantly new ideas—we observed that for general tensors, overparametrized gradient flow may have a very different behavior compared to the greedy low-rank procedure, as it is possible for two large component in the same direction to split into two different directions (see more details in Appendix E). We leave that as an interesting open problem.

## Acknowledgements

Rong Ge, Xiang Wang and Mo Zhou are supported in part by NSF Award CCF-1704656, CCF-1845171 (CAREER), CCF-1934964 (Tripods), a Sloan Research Fellowship, and a Google Faculty Research Award.

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
