## Overview of Supplementary Materials

In the supplementary material we will give detailed proof for Theorem 1. We will first highlight a few technical ideas that goes into the proof, and then give details for each part of the proof.

**Continuity Argument**   Continuity argument is the main tool we use to prove Proposition 1. Intuitively, the continuity argument says that if whenever a property is about to be violated, there exists a positive speed that pulls it back, then that property will never be violated. In some sense, this is the continuous version of the mathematical induction or, equivalently, the minimal counterexample method. See Section 1.3 of Tao (2006) for a short discussion on this method.

However, since our algorithm is not just gradient flow, and in particular involves reinitialization steps that are not continuous, we need to generalize continuity argument to handle impulses. We give detailed lemmas in Section A.1 as the continuity argument is mostly used to prove Proposition 1.

**Approximating residual**   In many parts of the proof, we approximate the residual $T^* - T$ as:

$$T^* - T = \sum_{i=1}^{d} \tilde{a}_i e_i^{\otimes 4} + \Delta,$$

where $\tilde{a}_i = a_i - \hat{a}_i$. That is, we think of $T^* - T$ as an orthogonal tensor with some perturbations. The norm of the perturbation $\|\Delta\|_F$ is going to be bounded by $O(\alpha + m\delta_1^2)$, which is sufficient in several parts of the proof that only requires crude estimates. However, in several key steps of our proof (including conditions (a) and (b) of Proposition 1 and the analysis of the first phase), it is important to use extra properties of $\Delta$. In particular we will expand $\Delta$ to show that for a basis vector $e_i$ we always have $\Delta(e_i^{\otimes 4}) = o(\alpha)$, which gives us tighter bounds when we need them.

**Radial and tangent movement**   Throughout the proof, we often need to track the movement of a particular component $w$ (a column in $W$). It is beneficial to separate the movement of $w$ into radial and tangent movement, where radial movement is defined as $\left\langle \frac{dw}{dt}, w \right\rangle$ and tangent movement is defined as $P_{w^\perp} \frac{dw}{dt}$ (where $P_{w^\perp}$ is the projection to the orthogonal subspace of $w$). Intuitively, the radial movement controls the norm of the component $w$, and the tangent movement controls the direction of $w$. When the component $w$ has small norm, it will not significantly change the residual $T^* - T$, therefore we mostly focus on the tangent movement; on the other hand when norm of $w$ becomes large in our proof we show that it must already be correlated with one of the ground truth components, which allow us to better control its norm growth.

**Overall structure of the proof**   The entire proof is a large induction/continuity argument which maintains Proposition 1 as well as properties of the two phases (summarized later in Assumption 1). In each part of the proof, we show that if we assume these conditions hold for the previous time, then they will continue to hold during the phase/after reinitialization.

In Section A we prove Proposition 1 assuming Assumption 1 holds before. In Section B.2 we prove guarantees of Phase 1 and reinitialization assuming Proposition 1. In Seciton C we prove guarantees for Phase 2 assuming Proposition 1. Finally in Section D we give the proof of the main theorem.

**Experiments**   Finally in Section E.1 we give details about experiments that illustrate the deflation process, and show why such a process may not happen for non-orthogonal tensors.

## A   Proofs for Proposition 1

The goal of this section is to prove Proposition 1 under Assumption 1. We also prove Claim 2 in Section A.6.

**Notations**   Recall we defined

$$\mathbb{E}_{i,w}^{(s,t)} f(w^{(s,t)}) := \frac{1}{\hat{a}_i^{(s,t)}} \sum_{w^{(s,t)} \in S_i^{(s,t)}} \left\| w^{(s,t)} \right\|^2 f(w^{(s,t)}).$$

We will use this notation extensively in this section. For simplicity, we shall drop the superscript of epoch $s$. Further, we sometimes consider expectation with two variables $v$ and $w$:

$$\mathbb{E}_{i,v,w}^{(s,t)} f(w^{(s,t)}) := \frac{1}{\left[\hat{a}_i^{(s,t)}\right]^2} \sum_{v^{(s,t)},w^{(s,t)} \in S_i^{(s,t)}} \left\|v^{(s,t)}\right\|^2 \left\|w^{(s,t)}\right\|^2 f(w^{(s,t)}, v^{(s,t)}).$$

We will also use $z_t$ to denote $z^{(t)} := \langle \bar{v}^{(t)}, \bar{w}^{(t)} \rangle$ and $\tilde{a}_k^{(t)} := a_k - \hat{a}_k^{(t)}$. Note that $v$ and $w$ in this section (and later in the proof) just serve as arbitrary components in columns of $W$.

**Assumption 1.** *Throughout this section, we assume the following.*

(a) *For any $k \in [d]$, in phase 1, when $\|v^{(t)}\|$ enters $S_k^{(t)}$, that is, $\|v^{(t)}\| = \delta_1$, we have $[\bar{v}_k^{(t)}]^2 \geq 1 - \alpha^2$ if $\hat{a}_k^{(t)} < \alpha$ and $[\bar{v}_k^{(t)}]^2 \geq 1 - \alpha$ if $\hat{a}_k^{(t)} \geq \alpha$.*

(b) *There exists a small constant $c > 0$ s.t. for any $k \in [d]$ with $a_k < c\beta^{(s)}$, in phase 1, no components will enter $S_k^{(t)}$.*

(c) *For any $k \in [d]$, in phase 2, no components will enter $S_k^{(t)}$.*

(d) *For the parameters, we assume $m\delta_1^2 \leq \alpha^3$ and $\Omega(\sqrt{\alpha}) \leq \lambda \leq O\left(\min_s \beta^{(s)}\right) = O(\varepsilon/\sqrt{d})$.*

**Remark.** As we mentioned, the entire proof is an induction and we only need the assumption up to the point that we are analyzing. The assumption will be proved later in Appendix B and C to finish the induction/continuity argument. The reason we state this assumption here, and state it as an assumption, is to make the dependencies more transparent.

**Remark on the choice of $\lambda$.** The lower bound $\lambda = \Omega(\sqrt{\alpha})$ comes from Lemma A.1. For the upper bound, first note that when $\lambda$ is larger than $a_k$, actually the norm of components in $S_k^{(t)}$ can decrease (cf. Lemma A.6). Hence, we require $\lambda < c \min_s \beta^{(s)}/10$ where $c$ is the constant in (c). This makes sure in phase 2 the growth rate of $\hat{a}_k^{(t)}$ is not too small.

**Proposition 1** (Induction hypothesis). *In the setting of Theorem 1, for any epoch $s$ and time $t$ and every $k \in [d]$, the following hold.*

(a) *For any $w^{(s,t)} \in S_k^{(s,t)}$, we have $\left[\bar{w}_k^{(s,t)}\right]^2 \geq 1 - \alpha$.*

(b) *If $S_k^{(s,t)}$ is nonempty, $\mathbb{E}_{k,w}^{(s,t)} \left[\bar{w}_k^{(s,t)}\right]^2 \geq 1 - \alpha^2 - 4sm\delta_1^2$.*

(c) *We always have $a_k - \hat{a}_k^{(s,t)} \geq \lambda/6 - sm\delta_1^2$; if $a_k \geq \frac{\beta^{(s)}}{1-\gamma}$, we further know $a_k - \hat{a}_k^{(s,t)} \leq \lambda + sm\delta_1^2$.*

(d) *If $w^{(s,t)} \in S_\varnothing^{(s,t)}$, then $\|w^{(s,t)}\| \leq \delta_1$.*

Before we move on to the proof, we collect some further remarks on Proposition 1 and the proof overview here.

**Remark on the epoch correction term.** Note that conditions (b) and (c) have an additional term with form $O(sm\delta_1^2)$. This is because these average bounds may deteriorate a little when the content of $S_k^{(t)}$ changes, which will happen when new components enter $S_k^{(t)}$ or the reinitialization throw some components out of $S_k^{(t)}$. The norm of the components involved in these fluctuations is upper bounded by $\delta_1$ and the number by $m$. Thus the $O(m\delta_1^2)$ factor. The factor $s$ accounts for the accumulation across epochs. We need this to guarantee at the beginning of each epoch, the conditions hold with some slackness (cf. Lemma A.5). Though this issue can be fixed by a slightly sharper estimations for the ending state of each epoch, adding one epoch correction term is simpler and, since we only have $\log(d/\epsilon)$ epochs, it does not change the bounds too much and, in fact, we can always absorb them into the coefficients of $\lambda$ and $\alpha^2$, respectively.

**Remark on condition (a).** Note that Assumption 1 makes sure that when a component enters $S_k^{(t)}$, we always have $[\bar{v}_k^{(t)}]^2 \geq 1 - \alpha$. Hence, essentially this condition says that it will remain basis-like. Following the spirit of the continuity argument, to maintain this condition, it suffices to prove Lemma A.1, the proof of which is deferred to Section A.3. Also note that by Assumption 1 and the definition of $S_k^{(s,t)}$, neither the entrance of new components nor the reinitialization will break this condition.

**Lemma A.1.** *Suppose that at time t, Proposition 1 is true. Assuming $\delta_1^2 = O(\alpha^{1.5}/m)$, then for any $v^{(t)} \in S_k^{(t)}$, we have*

$$\frac{\mathrm{d}}{\mathrm{d}t}[\bar{v}^{(t)}]^2 \geq 8\tilde{a}^{(t)} \left(1 - [\bar{v}_k^{(t)}]^2\right) [\bar{v}_k^{(t)}]^4 - O\left(\alpha^{1.5}\right),$$

*In particular, if $\lambda = \Omega\left(\sqrt{\alpha}\right)$, then $\frac{\mathrm{d}}{\mathrm{d}t}[\bar{v}^{(t)}]^2 > 0$ whenevner $[\bar{v}_k^{(t)}]^2 = 1 - \alpha$.*

**Remark on condition (b).** The proof idea of condition (b) is similar to condition (a) and we prove Lemma A.2 in Section A.4. In Section A.4, we also handle the impulses caused by the entrance of new components and the reinitialization.

**Lemma A.2.** *Suppose that at time t, Proposition 1 is true and $S_k^{(t)} \neq \varnothing$. Assuming $\delta_1^2 = O(\alpha^3/m)$, we have*

$$\frac{\mathrm{d}}{\mathrm{d}t}\mathbb{E}_{k,v}^{(t)}[\bar{v}_k^{(t)}]^2 \geq 8\tilde{a}_k^{(t)}(1 - \mathbb{E}_{k,v}^{(t)}[\bar{v}_k^{(t)}]^2) - O(\alpha^3).$$

*In particular, if $\lambda = \Omega(\alpha)$, then $\frac{\mathrm{d}}{\mathrm{d}t}\mathbb{E}_{k,v}^{(t)}[\bar{v}_k^{(t)}]^2 > 0$ when $\mathbb{E}_{k,v}^{(t)}[\bar{v}_k^{(t)}]^2 < 1 - \alpha^2/2$.*

**Remark on condition (c).** This condition says that the residual along direction $k$ is always $\Omega(\lambda)$. This guarantees the existence of a small attraction region around $e_k$, which will keep basis-like components basis-like. We rely on the regularizer to maintain this condition. The second part of condition (c) means fitted directions will remain fitted. We prove Lemma A.3 and handle the impulses in Section A.5.

**Lemma A.3** (Lemma A.17 and Lemma A.18). *Suppose that at time t, Proposition 1 is true. and no impulses happen at time t. Then at time t, we have*

$$\frac{1}{\hat{a}_k^{(t)}} \frac{\mathrm{d}}{\mathrm{d}t}\hat{a}_k^{(t)} = 2\tilde{a}_k^{(t)} - \lambda \pm O\left(\alpha^2\right).$$

*In particular, $\frac{\mathrm{d}}{\mathrm{d}t}\hat{a}_k^{(t)}$ is negative (resp. positive) when $\hat{a}_k^{(t)} > a_k - \lambda/6$ (resp. $\hat{a}_k^{(t)} < a_k - \lambda$).*

## A.1   Continuity argument

We mostly use the following version of continuity argument, which is adapted from Proposition 1.21 of Tao (2006).

**Lemma A.4.** *Let $\mathbf{I}^{(t)}$ be a statement about the structure of some object. $\mathbf{I}^{(t)}$ is true for all $t \geq 0$ as long as the following hold.*

*(a) $\mathbf{I}^{(0)}$ is true.*

*(b) $\mathbf{I}$ is closed in the sense that for any sequence $t_n \to t$, if $\mathbf{I}^{(t_n)}$ is true for all $n$, then $\mathbf{I}^{(t)}$ is also true.*

*(c) If $\mathbf{I}^{(t)}$ is true, then there exists some $\delta > 0$ s.t. $\mathbf{I}^{(s)}$ is true for $s \in [t, t + \delta)$.*

*In particular, if $\mathbf{I}^{(t)}$ has form $\bigwedge_{i=1}^{N} \bigvee_{j=1}^{N} p_{i,j}^{(t)} \leq q_{i,j}$. Then, we can replace (b) and (c) by the following.*

*(b') $p_{i,j}^{(t)}$ is $C^1$ for all $i, j$.*

*(c')* *Suppose at time $t$, $\mathbf{I}^{(t)}$ is true but some clause $\bigvee_{j=1}^{N} p_{i,j}^{(t)} \leq q_{i,j}$ is tight, in the sense that $p_{i,j}^{(t)} \geq q_{i,j}$ for all $j$ with at least one equality. Then there exists some $k$ s.t. $p_{i,k}^{(t)} = q_{i,k}$ and $\dot{p}_{i,k}^{(t)} < 0$.*

*Proof.* Define $t' := \sup\{t \geq 0 \ : \ \mathbf{I}^{(t)} \text{ is true}\}$. Since $\mathbf{I}^{(0)}$ is true, $t' \geq 0$. Assume, to obtain a contradiction, that $t' < \infty$. Since $\mathbf{I}$ is closed, $\mathbf{I}^{(t')}$ is true, whence there exists a small $\delta > 0$ s.t. $\mathbf{I}^{(t)}$ is true in $[t', t' + \delta)$. Contradiction.

For the second set of conditions, first note that the continuity of $p_{i,j}^{(t)}$ and the non-strict inequalities imply that $\mathbf{I}$ is closed. Now we show that (b') and (c') imply (c). If none of the clause is tight at time $t$, by the continuity of $p_{i,j}^{(t)}$, $\mathbf{I}$ holds in a small neighborhood of $t$. If some constraint is tight, by (c') and the $C^1$ condition, we have $p_{i,k}^{(t)} < q_{i,k}$ in a right small neighborhood of $t$. $\qquad\square$

**Remark.** Despite the name "continuity argument", it is possible to generalize it to certain classes of discontinuous functions. In particular, we consider impulsive differential equations here, that is, for almost every $t$, $p^{(t)}$ behaves like a usual differential equation, but at some $t_i$, it will jump from $p^{(t_i-)}$ to $p^{(t_i)} = p^{(t_i-)} + \delta_i$. See, for example, Lakshmikantham et al. (1989) for a systematic treatment on this topic. Suppose that we still want to maintain the property $p^{(t)} \leq 0$. If the total amount of impulses is small and we have some cushion in the sense that $\dot{p}^{(t)} < 0$ whenever $p^{(t)} \in [-\varepsilon, 0]$, then we can still hope $p^{(t)} \leq 0$ to hold for all $t$, since, intuitively, only the jumps can lead $p^{(t)}$ into $[-\varepsilon, 0]$, and the normal $\dot{p}^{(t)}$ will try to take it back to $(-\infty, -\varepsilon)$. As long as the amount of impulses is smaller than the size $\varepsilon$ of the cushion, then the impulses will never break things. We formalize this idea in the next lemma.

**Lemma A.5** (Continuity argument with impulses)**.** *Let $0 < t_1 < \cdots < t_N < \infty$ be the moments at which the impulse happens and $\delta_1, \ldots, \delta_N \in \mathbb{R}$ the size of the impulses at each $t_i$. Let $p : [0, \infty) \to \mathbb{R}$ be a function that is $C^1$ on $[0, t_1)$, every $(t_i, t_{i+1})$ and $(t_N, \infty)$, and $p^{(t_i)} = p^{(t_i-)} + \delta_i$. Write $\Delta = \sum_{i=1}^{N} \max\{0, \delta_i\}$. If (a) $p^{(0)} \leq -\Delta$ and (b) for every $t \notin \{t_i\}_{i=1}^{N}$ with $p^{(t)} \in [-\Delta, 0]$, we have $\dot{p}^{(t)} < 0$, then $p^{(t)} \leq 0$ always holds.*

**Remark.** Note that if there is no impulses, then $p^{(t)}$ is a usual $C^1$ function and we recover conditions (b') and (c') of Lemma A.4. Also, though the statement here only concerns one $a_t$, one can incorporate it into Lemma A.4 by replacing (b') and (c') with the hypotheses of this lemma and modify (a) to be $p_{i,j}^{(0)} \leq p_{i,j} - \Delta_{i,j}$.

*Proof.* We claim that $p^{(t)} \leq -\Delta + \sum_{i=1}^{N} \mathbb{1}_{t \leq t_k} \max\{0, \delta_i\} =: q^{(t)}$. Define $t' = \sup\{t \geq 0 \ : \ p^{(t)} \leq q^{(t)}\}$. Since $p^{(t)} \leq -\Delta$ and $t_1 > 0$, $t' \geq 0$. Assume, to obtain a contradiction, that $t' < \infty$ and consider $p^{(t')}$. If $t' = t_k$ for some $k$, then, by the definition of $t'$, $p^{(t'-)} \leq -\Delta + \sum_{i=1}^{k-1} \max\{0, \delta_i\}$, whence, $p^{(t')} = p^{(t'-)} + \delta_k \leq -\Delta + \sum_{i=1}^{k} \max\{0, \delta_i\}$. Contradiction. If $t' \notin \{t_i\}_{i=1}^{N}$, then by the continuity of $p$, we have $p^{(t')} = q^{(t')}$. Then, since $\dot{p}^{(t')} < 0$ and $p$ is $C^1$, we have $p^{(t)} < p^{(t')} = q^{(t')} = q^{(t)}$ in $[t', t'+\tau]$ for some small $\tau > 0$, which contradicts the maximality of $t'$. Thus, $p^{(t)} \leq 0$ holds for all $t \geq 0$. $\qquad\square$

## A.2 Preliminaries

The next two lemmas give formulas for the norm growth rate and tangent speed of each component.

**Lemma A.6** (Norm growth rate)**.** *For any $v^{(t)}$, we have*

$$\frac{1}{2\|v^{(t)}\|^2} \frac{\mathrm{d}}{\mathrm{d}t} \left\|v^{(t)}\right\|^2 = \sum_{i=1}^{d} a_i [\bar{v}_i^{(t)}]^4 - \sum_{i=1}^{d} \hat{a}_i^{(t)} \mathbb{E}_{i,w}^{(t)} \left\{[z^{(t)}]^4\right\} - T_\varnothing^{(t)} \left([\bar{v}^{(t)}]^{\otimes 4}\right) - \frac{\lambda}{2}.$$

*Proof.* Due to the 2-homogeneity, we have[6]

$$\frac{1}{2\left\|v^{(t)}\right\|^2}\frac{\mathrm{d}}{\mathrm{d}t}\left\|v^{(t)}\right\|^2 = \left(T^* - T^{(t)}\right)\left([\bar{v}^{(t)}]^{\otimes 4}\right) - \frac{\lambda}{2}.$$

The ground truth terms can be rewritten as

$$T^*\left([\bar{v}^{(t)}]^{\otimes 4}\right) = \sum_{i=1}^d a_i[\bar{v}_i^{(t)}]^4.$$

Decompose the $T^{(t)}$ term accordingly and we get

$$T^{(t)}\left([\bar{v}^{(t)}]^{\otimes 4}\right) = \sum_{i=1}^d \hat{a}^{(t)}\mathbb{E}_{i,w}^{(t)}\left\{[z^{(t)}]^4\right\} + T_\varnothing^{(t)}\left([\bar{v}^{(t)}]^{\otimes 4}\right).$$

$\square$

**Lemma A.7** (Tangent speed)**.** *Suppose that at time $t$, Proposition 1 is true. Then at time $t$, for any $v^{(t)} \in W^{(t)}$ and any $k \in [d]$, we have*

$$\frac{\mathrm{d}}{\mathrm{d}t}[\bar{v}^{(t)}]^2 = G_1 - G_2 - G_3 \pm O(m\delta_1^2),$$

*where*

$$G_1 := 8a_k\left(1 - [\bar{v}_k^{(t)}]^2\right)[\bar{v}_k^{(t)}]^4 - 8\hat{a}_k^{(t)}\left(1 - [\bar{v}_k^{(t)}]^2\right)\mathbb{E}_{k,w}^{(t)}\left\{[z^{(t)}]^4\right\}$$
$$+ 8\hat{a}_k^{(t)}\mathbb{E}_{k,w}^{(t)}\left\{[z^{(t)}]^3\left\langle\bar{w}_{-k},\bar{v}_{-k}\right\rangle\right\},$$
$$G_2 = 8\sum_{i\neq k}\hat{a}_i^{(t)}\mathbb{E}_{i,w}^{(t)}\left\{[z^{(t)}]^3 v_k^{(t)}w_k^{(t)}\right\},$$
$$G_3 = 8[\bar{v}_k^{(t)}]^2\sum_{i\neq k}\left(a_i[\bar{v}_i^{(t)}]^4 - \hat{a}_i^{(t)}\mathbb{E}_{i,w}^{(t)}\left\{[z^{(t)}]^4\right\}\right).$$

**Remark.** Intuitively, $G_1$ captures the local dynamics around $e_k$ and $G_2$ characterize the cross interaction between different ground truth directions.

*Proof.* Let's compute the derivative of $[\bar{v}_k^{(t)}]^2$ in terms of time $t$:

$$\frac{\mathrm{d}[\bar{v}_k^{(t)}]^2}{\mathrm{d}t} = 2\bar{v}_k^{(t)}\cdot\frac{d}{dt}\frac{v_k^{(t)}}{\left\|v^{(t)}\right\|}$$
$$= 2\bar{v}_k^{(t)}\cdot\frac{1}{\left\|v^{(t)}\right\|}\frac{d}{dt}v_k^{(t)} + 2[\bar{v}_k^{(t)}]^2\cdot\frac{d}{dt}\frac{1}{\left\|v^{(t)}\right\|}$$
$$= 2\bar{v}_k^{(t)}\cdot\frac{1}{\left\|v^{(t)}\right\|}[-\nabla L(v^{(t)})]_k - 2[\bar{v}_k^{(t)}]^2\cdot\frac{\left\langle\bar{v}^{(t)},-\nabla L(v^{(t)})\right\rangle}{\left\|v^{(t)}\right\|}$$
$$= 2\bar{v}_k^{(t)}\cdot\frac{1}{\left\|v^{(t)}\right\|}[-(I - \bar{v}^{(t)}[\bar{v}^{(t)}]^\top)\nabla L(v^{(t)})]_k.$$

Note that

$$\nabla f(v^{(t)}) = 4(T^{(t)} - T^*)([\bar{v}^{(t)}]^{\otimes 2}, \bar{v}^{(t)}, I) - 2(T^{(t)} - T^*)([\bar{v}^{(t)}]^{\otimes 4})\bar{v}^{(t)} + \lambda\bar{v}^{(t)},$$

where the last two terms left multiplied by $(I - \bar{v}^{(t)}[\bar{v}^{(t)}]^\top)$ equals to zero. Therefore,

$$\frac{\mathrm{d}[\bar{v}_k^{(t)}]^2}{\mathrm{d}t} = 8\bar{v}_k^{(t)}\left[(T^* - T^{(t)})([\bar{v}^{(t)}]^{\otimes 3}, I) - (T^* - T^{(t)})([\bar{v}^{(t)}]^{\otimes 4})\bar{v}^{(t)}\right]_k$$

---

[6]In the mean-field terminologies, the RHS is just the first variation (or functional derivative) of the loss at $\bar{v}^{(t)}$.

We can write $T^*$ as $\sum_{i\in[d]} a_i e_i^{\otimes 4}$ and write $T^{(t)}$ as $\sum_{i\in[d]} T_i^{(t)} + T_\varnothing^{(t)}$. Since Proposition 1 is true at time $t$, we know any $w^{(t)}$ in $W_\varnothing^{(t)}$ has norm upper bounded by $\delta_1$, which implies $\left\|T_\varnothing^{(t)}\right\|_F \le m\delta_1^2$. Therefore, we have

$$\left|8\bar{v}_k^{(t)}\left[-T_\varnothing^{(t)}([\bar{v}^{(t)}]^{\otimes 3}, I) + T_\varnothing^{(t)}([\bar{v}^{(t)}]^{\otimes 4})\bar{v}^{(t)}\right]_k\right| \le O(m\delta_1^2).$$

For any $i \in [d]$, we have

$$\left[T_i^{(t)}([\bar{v}^{(t)}]^{\otimes 3}, I)\right]_k = \sum_{w^{(t)}\in S_i^{(t)}} \left\|w^{(t)}\right\|^2 \left\langle \bar{w}^{(t)}, \bar{v}^{(t)}\right\rangle^3 \bar{w}_k^{(t)}$$

$$= \hat{a}_k^{(t)}\mathbb{E}_{k,w}^{(t)}\left\langle \bar{w}^{(t)}, \bar{v}^{(t)}\right\rangle^3 \bar{w}_k^{(t)},$$

and

$$\left[T_i^{(t)}([\bar{v}^{(t)}]^{\otimes 4})\bar{v}^{(t)}\right]_k = \sum_{w^{(t)}\in S_i^{(t)}} \left\|w^{(t)}\right\|^2 \left\langle \bar{w}^{(t)}, \bar{v}^{(t)}\right\rangle^4 \bar{v}_k^{(t)}$$

$$= \hat{a}_k^{(t)}\mathbb{E}_{k,w}^{(t)}\left\langle \bar{w}^{(t)}, \bar{v}^{(t)}\right\rangle^4 \bar{v}_k^{(t)}.$$

For any $i \in [d]$, we have

$$\left[T^*([\bar{v}^{(t)}]^{\otimes 3}, I)\right]_k = [\bar{v}_k^{(t)}]^3 \mathbb{1}\{i = k\}$$

and

$$\left[T^*([\bar{v}^{(t)}]^{\otimes 4})\bar{v}^{(t)}\right]_k = [\bar{v}_i^{(t)}]^4 \bar{v}_k^{(t)}$$

Based on the above calculations, we can see that

$$G_1 = 8\bar{v}_k^{(t)}\left[(T_k^* - T_k^{(t)})([\bar{v}^{(t)}]^{\otimes 3}, I) - (T_k^* - T_k^{(t)})([\bar{v}^{(t)}]^{\otimes 4})\bar{v}^{(t)}\right]_k$$

$$G_2 = 8\bar{v}_k^{(t)}\left[\sum_{i\neq k} T_i^{(t)}([\bar{v}^{(t)}]^{\otimes 3}, I)\right]_k$$

$$G_3 = 8[\bar{v}_k^{(t)}]^2 \sum_{i\neq k}(T_i^* - T_i^{(t)})([\bar{v}^{(t)}]^{\otimes 4}),$$

and the error term $O(m\delta_1^2)$ comes from $T_\varnothing^{(t)}$. To complete the proof, use the identity $\langle \bar{w}, \bar{v}\rangle = \bar{w}_k\bar{v}_k + \langle \bar{w}_{-k}, \bar{v}_{-k}\rangle$ to rewrite $G_1$. $\qquad\square$

One may wish to skip all following estimations and come back to them when needed.

**Lemma A.8.** *For any $\bar{v}$ with $\bar{v}_k^2 \ge 1 - \alpha$ and any $\bar{w} \in \mathbb{S}^{d-1}$, we have $|\langle \bar{v}, \bar{w}\rangle| = |\bar{w}_k| \pm \sqrt{\alpha}$.*

*Proof.* Assume w.o.l.g. that $k = 1$. Note that the set $\{\bar{v} \in \mathbb{S}^{d-1} : \bar{v}_k^2 \ge 1 - \alpha\}$ is invariant under rotation of other coordinates, whence we may further assume w.o.l.g. that $\bar{w} = \bar{w}_1 e_1 + \sqrt{1 - \bar{w}_1^2}e_2$. Then,

$$|\langle \bar{w}, \bar{v}\rangle| = \left|\bar{w}_1\bar{v}_1 + \sqrt{1 - \bar{v}_1^2}\sqrt{1 - \bar{w}_1^2}\right|$$

$$\ge |\bar{w}_1|\sqrt{1 - \alpha} - \sqrt{\alpha}\sqrt{1 - \bar{w}_1^2}$$

$$= \frac{\bar{w}_1^2(1 - \alpha) - \alpha(1 - \bar{w}_1^2)}{|\bar{w}_1|\sqrt{1 - \alpha} + \sqrt{\alpha}\sqrt{1 - \bar{w}_1^2}}$$

$$= \frac{\bar{w}_1^2 - \alpha}{|\bar{w}_1|\sqrt{1 - \alpha} + \sqrt{\alpha}\sqrt{1 - \bar{w}_1^2}} \ge \frac{\bar{w}_1^2 - \alpha}{|\bar{w}_1| + \sqrt{\alpha}} = |\bar{w}_1| - \sqrt{\alpha}.$$

The other direction follows immediately from

$$| \langle \bar{w}, \bar{v} \rangle | \leq |\bar{w}_1||\bar{v}_1| + \left| \sqrt{1 - \bar{v}_1^2} \sqrt{1 - \bar{w}_1^2} \right| \leq |\bar{w}_1| + \sqrt{\alpha}.$$

$\square$

The next two lemmas bound the cross interaction between different $S_k^{(t)}$.

**Lemma A.9.** *Suppose that at time $t$, Proposition 1 is true. Then for any $v^{(t)} \in S_k^{(t)}$ and $l \neq k$, the following hold.*

*(a)* $[\bar{v}_l^{(t)}]^4 \leq \alpha^2$.

*(b)* $\mathbb{E}_{l,w}^{(t)} \left\{ [z_t]^4 \right\} \leq O(\alpha^2)$.

*(c)* $\mathbb{E}_{l,w}^{(t)} \left\{ [z_t]^3 \bar{v}_l \bar{w}_l \right\} \leq O(\alpha^2)$.

*Proof.* (a) follows immediately from $[v_l^{(t)}]^4 \leq (1 - [v_l^{(t)}]^2) \leq \alpha^2$. For (b), apply Lemma A.8 and we get

$$\mathbb{E}_{l,w}^{(t)} \left\{ [z_t]^4 \right\} \leq \mathbb{E}_{l,w}^{(t)} \left\{ \left( |\bar{w}_k| + \sqrt{\alpha} \right)^4 \right\} \leq \mathbb{E}_{l,w}^{(t)} \left\{ [\bar{w}_k]^4 + 4|\bar{w}_k|^3 \sqrt{\alpha} + 6[\bar{w}_k]^2 \alpha + 4|\bar{w}_k|\alpha^{1.5} + \alpha^2 \right\}.$$

For the first three terms, it suffices to note that $\mathbb{E}_{l,w}^{(t)} \left\{ [\bar{w}_k]^2 \right\} \leq \alpha^2$. For the fourth term, it suffices to additionally recall Jensen's inequality. Combine these together and we get $\mathbb{E}_{l,w}^{(t)} \left\{ [z_t]^4 \right\} = O(\alpha^2)$. The proof of (b), *mutatis mutandis*, yields (c). $\square$

**Lemma A.10.** *Suppose that at time $t$, Proposition 1 is true. Then for any $k \neq l$, the following hold.*

*(a)* $\mathbb{E}_{k,v}^{(t)}[\bar{v}_l^{(t)}]^4 \leq O(\alpha^3)$.

*(b)* $\mathbb{E}_{k,v}^{(t)} \mathbb{E}_{l,w}^{(t)}[z^{(t)}]^4 \leq O(\alpha^3)$.

*(c)* $\mathbb{E}_{k,v}^{(t)} \mathbb{E}_{l,w}^{(t)} \left\{ [z^{(t)}]^3 \bar{v}_k \bar{w}_k \right\} \leq O(\alpha^3)$.

*Proof.* For (a), we compute

$$\mathbb{E}_{k,v}^{(t)}[\bar{v}_l^{(t)}]^4 \leq \mathbb{E}_{k,v}^{(t)} \left\{ \left( 1 - [\bar{v}_k^{(t)}]^2 \right)^2 \right\} \leq \alpha \mathbb{E}_{k,v}^{(t)} \left\{ 1 - [\bar{v}_k^{(t)}]^2 \right\} \leq O(\alpha^3),$$

where the second inequality comes from the condition (a) of Proposition 1 and the third from condition (b) of Proposition 1. Now we prove (b). (c) can be proved in a similar fashion. For simplicity, write $x^{(t)} = \left\langle \bar{w}_{-l}^{(t)}, \bar{v}_{-l}^{(t)} \right\rangle$. Clear that $|x^{(t)}| \leq \sqrt{1 - [\bar{w}_l^{(t)}]^2}$ and by Jensen's inequality and condition (b) of Proposition 1, $\mathbb{E}_{l,w}^{(t)} \sqrt{1 - [\bar{w}_l^{(t)}]^2} \leq O(\alpha)$. We compute

$$\mathbb{E}_{k,v}^{(t)} \mathbb{E}_{l,w}^{(t)}[z^{(t)}]^4 = \mathbb{E}_{k,v}^{(t)} \mathbb{E}_{l,w}^{(t)} \left\{ [\bar{w}_l^{(t)}]^4 [\bar{v}_l^{(t)}]^4 + 4[\bar{w}_l^{(t)}]^3 [\bar{v}_l^{(t)}]^3 x^{(t)} + 6[\bar{w}_l^{(t)}]^2 [\bar{v}_l^{(t)}]^2 [x^{(t)}]^2 \right.$$
$$\left. + 4\bar{w}_l^{(t)} \bar{v}_l^{(t)} [x^{(t)}]^3 + [x^{(t)}]^4 \right\}.$$

We bound each of these five terms as follows.

$$\mathbb{E}^{(t)}_{k,v}\mathbb{E}^{(t)}_{l,w}\left\{[\bar{w}^{(t)}_l]^4[\bar{v}^{(t)}_l]^4\right\} \le \mathbb{E}^{(t)}_{k,v}[\bar{v}^{(t)}_l]^4 \le O(\alpha^3),$$

$$\mathbb{E}^{(t)}_{k,v}\mathbb{E}^{(t)}_{l,w}\left\{[\bar{w}^{(t)}_l]^3[\bar{v}^{(t)}_l]^3 x^{(t)}\right\} \le \mathbb{E}^{(t)}_{k,v}[\bar{v}^{(t)}_l]^3\mathbb{E}^{(t)}_{l,w}\left\{\sqrt{1-[\bar{w}^{(t)}_l]^2}\right\} \le O(\alpha^3),$$

$$\mathbb{E}^{(t)}_{k,v}\mathbb{E}^{(t)}_{l,w}\left\{[\bar{w}^{(t)}_l]^2[\bar{v}^{(t)}_l]^2[x^{(t)}]^2\right\} \le \mathbb{E}^{(t)}_{k,v}[\bar{v}^{(t)}_l]^2\mathbb{E}^{(t)}_{l,w}\left\{1-[\bar{w}^{(t)}_l]^2\right\} \le O(\alpha^3),$$

$$\mathbb{E}^{(t)}_{k,v}\mathbb{E}^{(t)}_{l,w}\left\{\bar{w}^{(t)}_l\bar{v}^{(t)}_l[x^{(t)}]^3\right\} \le \mathbb{E}^{(t)}_{k,v}\bar{v}^{(t)}_l\mathbb{E}^{(t)}_{l,w}\left\{\left(1-[\bar{w}^{(t)}_l]^2\right)^{1.5}\right\} \le O(\alpha^3),$$

$$\mathbb{E}^{(t)}_{k,v}\mathbb{E}^{(t)}_{l,w}[x^{(t)}]^4 \le \mathbb{E}^{(t)}_{l,w}\left\{\left(1-[\bar{w}^{(t)}_l]^2\right)^2\right\} \le O(\alpha^3).$$

Combine these together and we complete the proof. $\qquad\square$

**Lemma A.11.** *Suppose that at time t, Proposition 1 is true. Then, for any $v^{(t)} \in S^{(t)}_k$, we have* $\mathbb{E}^{(t)}_{k,w}\left\{[z^{(t)}]^4\right\} = [\bar{v}^{(t)}_k]^4 \pm O(\alpha^{1.5})$.

*Proof.* For simplicity, put $x^{(t)} = \left\langle \bar{w}^{(t)}_{-k}, \bar{v}^{(t)}_{-k}\right\rangle$. Note that $|x^{(t)}| \le \sqrt{1-[\bar{v}^{(t)}_k]^2}\sqrt{1-[\bar{w}^{(t)}_k]^2} \le \sqrt{\alpha}\sqrt{1-[\bar{w}^{(t)}_k]^2}$. Then

$$\mathbb{E}^{(t)}_{k,w}\left\{[z^{(t)}]^4\right\} = \mathbb{E}^{(t)}_{k,w}\left\{\left[\bar{w}^{(t)}_k\bar{v}^{(t)}_k + x^{(t)}\right]^4\right\} = [\bar{v}^{(t)}_k]^4\mathbb{E}^{(t)}_{k,w}\left\{[\bar{w}^{(t)}_k]^4\right\} \pm O(1)\mathbb{E}^{(t)}_{k,w}x^{(t)}.$$

For the first term, note that

$$\mathbb{E}^{(t)}_{k,w}\left\{[\bar{w}^{(t)}_k]^4\right\} = 1 - \mathbb{E}^{(t)}_{k,w}\left\{(1-[\bar{w}^{(t)}_k]^2)(1+[\bar{w}^{(t)}_k]^2)\right\} \ge 1 - 2\alpha^2.$$

For the second term, by Jensen's inequality, we have

$$\left|\mathbb{E}^{(t)}_{k,w}x^{(t)}\right| \le \sqrt{\alpha\mathbb{E}^{(t)}_{k,w}[1-[\bar{w}^{(t)}_k]^2]} \le \alpha^{1.5}.$$

Thus,

$$\mathbb{E}^{(t)}_{k,w}\left\{[z^{(t)}]^4\right\} = [\bar{v}^{(t)}_k]^4\left(1 \pm 2\alpha^2\right) \pm O(\alpha^{1.5}) = [\bar{v}^{(t)}_k]^4 \pm O(\alpha^{1.5}).$$

$\qquad\square$

**Lemma A.12.** *Suppose that at time t, Proposition 1 is true. Then we have* $\mathbb{E}^{(t)}_{k,v,w}\left\{[z^{(t)}]^4\right\} \ge 1 - O(\alpha^2)$.

*Proof.* For simplicity, put $x^{(t)} = \left\langle \bar{w}^{(t)}_{-k}, \bar{v}^{(t)}_{-k}\right\rangle$. We have

$$\mathbb{E}^{(t)}_{k,v,w}\left\{[z^{(t)}]^4\right\} = \mathbb{E}^{(t)}_{k,v,w}\left\{\left(\bar{w}^{(t)}_k\bar{v}^{(t)}_k + x^{(t)}\right)^4\right\}$$

$$\ge \mathbb{E}^{(t)}_{k,v,w}\left\{[\bar{w}^{(t)}_k]^4[\bar{v}^{(t)}_k]^4 + [\bar{w}^{(t)}_k]^3[\bar{v}^{(t)}_k]^3 x + \bar{w}^{(t)}_k\bar{v}^{(t)}_k x^3\right\}.$$

Note that

$$\mathbb{E}^{(t)}_{k,v,w}\left\{[\bar{w}^{(t)}_k]^3[\bar{v}^{(t)}_k]^3 x\right\} = \sum_{i \ne k}\mathbb{E}^{(t)}_{k,v,w}\left\{[\bar{w}^{(t)}_k]^3[\bar{v}^{(t)}_k]^3\bar{w}^{(t)}_i\bar{v}^{(t)}_i\right\}$$

$$= \sum_{i \ne k}\left(\mathbb{E}^{(t)}_{k,v,w}\left\{[\bar{w}^{(t)}_k]^3\bar{w}^{(t)}_i\right\}\right)^2 \ge 0.$$

(2)

Similarly, $\mathbb{E}^{(t)}_{k,v,w}\left\{\bar{w}^{(t)}_k\bar{v}^{(t)}_k x^3\right\} \ge 0$ also holds. Finally, by Jensen's inequality, we have

$$\mathbb{E}^{(t)}_{k,v,w}\left\{[z^{(t)}]^4\right\} \ge \mathbb{E}^{(t)}_{k,v,w}\left\{[\bar{w}^{(t)}_k]^4[\bar{v}^{(t)}_k]^4\right\}$$

$$= \left(\mathbb{E}^{(t)}_{k,w}\left\{[\bar{w}^{(t)}_k]^4\right\}\right)^2 \ge \left(\mathbb{E}^{(t)}_{k,w}\left\{[\bar{w}^{(t)}_k]^2\right\}\right)^4 \ge (1-\alpha^2)^4 = 1 - O(\alpha^2).$$

$\qquad\square$

## A.3 Condition (a): the individual bound

In this section, we show Lemma A.1, which implies condition ( a) of Proposition 1 always holds.

**Lemma A.1.** *Suppose that at time t, Proposition 1 is true. Assuming $\delta_1^2 = O(\alpha^{1.5}/m)$, then for any $v^{(t)} \in S_k^{(t)}$, we have*

$$\frac{\mathrm{d}}{\mathrm{d}t}[\bar{v}^{(t)}]^2 \geq 8\tilde{a}^{(t)}\left(1 - [\bar{v}_k^{(t)}]^2\right)[\bar{v}_k^{(t)}]^4 - O\left(\alpha^{1.5}\right),$$

*In particular, if $\lambda = \Omega\left(\sqrt{\alpha}\right)$, then $\frac{\mathrm{d}}{\mathrm{d}t}[\bar{v}^{(t)}]^2 > 0$ whenevner $[\bar{v}_k^{(t)}]^2 = 1 - \alpha$.*

*Proof.* Recall the definition of $G_1$, $G_2$ and $G_3$ from Lemma A.7. Now we estimate each of these three terms. By Lemma A.11, the first two terms of $G_1$ can be lower bounded by $8\tilde{a}^{(t)}\left(1 - [\bar{v}_k^{(t)}]^2\right)[\bar{v}_k^{(t)}]^4 - O(\hat{a}_k^{(t)}\alpha^{1.5})$ and, for the third term, replace $|z^{(t)}|$ with 1, and then, by the Cauchy-Schwarz inequality and Jensen's inequality, it is bounded $O(\hat{a}_k^{(t)}\alpha^{1.5})$. By Lemma A.9, $G_2$ and $G_3$ can be bounded by $O(1)\sum_{i\neq k}\hat{a}_i^{(t)}\alpha^2$. Thus,

$$\frac{\mathrm{d}}{\mathrm{d}t}[\bar{v}^{(t)}]^2 \geq 8\tilde{a}^{(t)}\left(1 - [\bar{v}_k^{(t)}]^2\right)[\bar{v}_k^{(t)}]^4 - O(1)\sum_{i=1}^{d}\hat{a}_k^{(t)}\alpha^{1.5} - O(m\delta_1^2)$$

$$\geq 8\tilde{a}^{(t)}\left(1 - [\bar{v}_k^{(t)}]^2\right)[\bar{v}_k^{(t)}]^4 - O\left(\alpha^{1.5}\right).$$

Now suppose that $[\bar{v}_k^{(t)}]^2 = 1 - \alpha$. By Proposition 1, we have $\tilde{a}^{(t)} \geq \lambda/6$. Hence,

$$\frac{\mathrm{d}}{\mathrm{d}t}[\bar{v}^{(t)}]^2 \geq \lambda\alpha(1-\alpha)^2 - O\left(\alpha^{1.5}\right) \geq \lambda\alpha - O\left(\alpha^{1.5}\right).$$

$\square$

## A.4 Condition (b): the average bound

**Bounding the total amount of impulses**

Note that there are two sources of impulses. First, when $\hat{a}_k^{(t)}$ is larger, the correlation of the newly-entered components is $1 - \alpha$ instead of $1 - \alpha^2$ and, second, the reinitialization may throw some components out of $S_k^{(t)}$.

First we consider the first type of impulses. Suppose that at time $t$, $\hat{a}_k^{(t)} \geq \alpha$, $\mathbb{E}_{k,w}^{(t)}\left\{[\bar{w}_k^{(t)}]^2\right\} = B$, and one particle $v^{(t)}$ enters $S_k^{(t)}$. The deterioration of the average bound can be bounded as

$$B - \left(\frac{\hat{a}_k^{(t)}}{\hat{a}_k^{(t)} + \left\|v^{(t)}\right\|^2}B + \frac{\left\|v^{(t)}\right\|^2}{\hat{a}_k^{(t)} + \left\|v^{(t)}\right\|^2}(1-\alpha)\right) = \frac{\left\|v^{(t)}\right\|^2}{\hat{a}_k^{(t)} + \left\|v^{(t)}\right\|^2}(B - (1-\alpha))$$

$$\leq \frac{\left\|v^{(t)}\right\|^2}{\alpha}2\alpha$$

$$= 2\left\|v^{(t)}\right\|^2.$$

Hence, the total amount of impulses caused by the entrance of new components can be bounded by $2m\delta_1^2$.

Now we consider the reinitialization. Again, it suffices to consider the case where $\hat{a}_k^{(t)} \geq \alpha$. Suppose that at time $t$, $\hat{a}_k^{(t)} \geq \alpha$, $\mathbb{E}_{k,w}^{(t)}\left\{[\bar{w}_k^{(t)}]^2\right\} = B$ and one particle $v^{(t)} \in S_k^{(t)}$ is reinitialized. By the definition of the algorithm, its norm is at most $\delta_1$. Hence, The deterioration of the average bound can

be bounded as[7]

$$B - \frac{\hat{a}_k^{(t)}}{\hat{a}_k^{(t)} - \left\|v^{(t)}\right\|^2} \left(B - \frac{\left\|v^{(t)}\right\|^2}{\hat{a}_k^{(t)}}[\bar{v}_k^{(t)}]^2\right) = \frac{\left\|v^{(t)}\right\|^2}{\hat{a}_k^{(t)} - \left\|v^{(t)}\right\|^2} \left([\bar{v}_k^{(t)}]^2 - B\right)$$

$$\leq \frac{\left\|v^{(t)}\right\|^2}{\hat{a}_k^{(t)}} 2\alpha$$

$$\leq 2\left\|v^{(t)}\right\|^2.$$

Since there are at most $m$ components, the amount of impulses caused by reinitialization is bounded by $2m\delta_1^2$.

Combine these two estimations together and we know that the total amount of impulses is bounded by $4m\delta_1^2$. This gives the epoch correction term of condition (c).

**The average bound**

First we derive a formula for the evolution of $\mathbb{E}_{k,w}^{(t)}\left\{[\bar{v}_k^{(t)}]^2\right\}$.

**Lemma A.13.** *For any $k$ with $S_k^{(t)} \neq \varnothing$, we have*

$$\frac{\mathrm{d}}{\mathrm{d}t}\mathbb{E}_{k,v}^{(t)}[\bar{v}_k^{(t)}]^2 = \mathbb{E}_{k,v}^{(t)}\left[\frac{d}{dt}[\bar{v}_k^{(t)}]^2\right]$$
$$+ 4\mathbb{E}_{k,v}^{(t)}\left[\left((T^* - T^{(t)})([\bar{v}^{(t)}]^{\otimes 4})\right)\left([\bar{v}_k^{(t)}]^2\right)\right] - 4\left(\mathbb{E}_{k,v}^{(t)}(T^* - T^{(t)})([\bar{v}^{(t)}]^{\otimes 4})\right)\left(\mathbb{E}_{k,v}^{(t)}[\bar{v}_k^{(t)}]^2\right).$$

**Remark**. The first term corresponds to the tangent movement and the two terms in the second line correspond to the norm change of the components.

*Proof.* Recall that

$$\mathbb{E}_{k,v}^{(t)}[\bar{v}_k^{(t)}]^2 = \frac{1}{\hat{a}_k^{(t)}} \sum_{v^{(t)} \in S_k^{(t)}} \left\|v^{(t)}\right\|^2 [\bar{v}_k^{(t)}]^2.$$

Taking the derivative, we have

$$\frac{d}{dt}\mathbb{E}_{k,v}^{(t)}[\bar{v}_k^{(t)}]^2 = \frac{1}{\hat{a}_k^{(t)}} \sum_{v^{(t)} \in S_k^{(t)}} \left\|v^{(t)}\right\|^2 \left(\frac{d}{dt}[\bar{v}_k^{(t)}]^2\right) + \frac{1}{\hat{a}_k^{(t)}} \sum_{v^{(t)} \in S_k^{(t)}} \left(\frac{d}{dt}\left\|v^{(t)}\right\|^2\right)[\bar{v}_k^{(t)}]^2$$
$$+ \left(\frac{d}{dt}\frac{1}{\hat{a}_k^{(t)}}\right) \sum_{v^{(t)} \in S_k^{(t)}} \left\|v^{(t)}\right\|^2 [\bar{v}_k^{(t)}]^2.$$

The first term is just $\mathbb{E}_{k,v}^{(t)}\frac{d}{dt}[\bar{v}_k^{(t)}]^2$. Denote $R(\bar{v}^{(t)}) = 2(T^* - T^{(t)})([\bar{v}^{(t)}]^{\otimes 4}) - \lambda$. We can write the second term as follows:

$$\frac{1}{\hat{a}_k^{(t)}} \sum_{v^{(t)} \in S_k^{(t)}} \left(\frac{d}{dt}\left\|v^{(t)}\right\|^2\right)[\bar{v}_k^{(t)}]^2 = \frac{1}{\hat{a}_k^{(t)}} \sum_{v^{(t)} \in S_k^{(t)}} 2R(\bar{v}^{(t)})\left\|v^{(t)}\right\|^2 [\bar{v}_k^{(t)}]^2$$
$$= 2\mathbb{E}_{k,v}^{(t)}\left[R(\bar{v}^{(t)})[\bar{v}_k^{(t)}]^2\right]$$

---

[7]The second term is obtained by solving the equation $B = \frac{\hat{a}_k^{(t)} - \left\|v^{(t)}\right\|^2}{\hat{a}_k^{(t)}}B' + \frac{\left\|v^{(t)}\right\|^2}{\hat{a}_k^{(t)}}[\bar{v}_k^{(t)}]^2$ for $B'$.

Finally, let's consider $\frac{d}{dt}\frac{1}{\hat{a}_k^{(t)}}$ in the third term,

$$
\begin{aligned}
\frac{d}{dt}\frac{1}{\hat{a}_k^{(t)}} &= -\frac{1}{[\hat{a}_k^{(t)}]^2}\frac{d}{dt}\hat{a}_k^{(t)} \\
&= -\frac{1}{[\hat{a}_k^{(t)}]^2}\frac{d}{dt}\sum_{v^{(t)}\in S_k^{(t)}}\left\|v^{(t)}\right\|^2 \\
&= -\frac{2}{[\hat{a}_k^{(t)}]^2}\sum_{v^{(t)}\in S_k^{(t)}}R(\bar{v}^{(t)})\left\|v^{(t)}\right\|^2 \\
&= -\frac{2}{\hat{a}_k^{(t)}}\mathbb{E}_{k,v}^{(t)}R(\bar{v}^{(t)}).
\end{aligned}
$$

Overall, we have

$$
\begin{aligned}
\frac{d}{dt}\mathbb{E}_{k,v}^{(t)}[\bar{v}_k^{(t)}]^2 =&\mathbb{E}_{k,v}^{(t)}\left[\frac{d}{dt}[\bar{v}_k^{(t)}]^2\right] \\
&+ 4\mathbb{E}_{k,v}^{(t)}\left[\left((T^*-T^{(t)})([\bar{v}^{(t)}]^{\otimes 4})\right)\left([\bar{v}_k^{(t)}]^2\right)\right] - 4\left(\mathbb{E}_{k,v}^{(t)}(T^*-T^{(t)})([\bar{v}^{(t)}]^{\otimes 4})\right)\left(\mathbb{E}_{k,v}^{(t)}[\bar{v}_k^{(t)}]^2\right)
\end{aligned}
$$

$\square$

**Lemma A.14** (Bound for the average tangent speed)**.** *Suppose that $m\delta_1^2 = O(\alpha^3)$ and, at time $t$, Proposition 1 is true and $S_k^{(t)}\neq\varnothing$. Then we have*

$$
\mathbb{E}_{k,v}^{(t)}\left[\frac{d}{dt}[\bar{v}_k^{(t)}]^2\right] \geq 8(a_k-\hat{a}_k^{(t)})(1-\mathbb{E}_{k,v}^{(t)}[\bar{v}_k^{(t)}]^2) - O(\alpha^3).
$$

*Proof.* Recall the definition of $G_1$, $G_2$ and $G_3$ from Lemma A.7.

- **Lower bound for $\mathbb{E}_{k,v}^{(t)}G_1$.** By (2), we have $\mathbb{E}_{k,v,w}^{(t)}\left\{[z^{(t)}]^3\langle\bar{w}_{-k},\bar{v}_{-k}\rangle\right\}\geq 0$, whence can be ignored. Meanwhile, note that $\mathbb{E}_{k,w}^{(t)}\left\{[z^{(t)}]^4\right\}\leq 1$. Therefore,

$$
\mathbb{E}_{k,v}^{(t)}G_1 \geq 8a_k\mathbb{E}_{k,v}^{(t)}\left\{\left(1-[\bar{v}_k^{(t)}]^2\right)[\bar{v}_k^{(t)}]^4\right\} - 8\hat{a}_k^{(t)}\mathbb{E}_{k,v}^{(t)}\left\{1-[\bar{v}_k^{(t)}]^2\right\}.
$$

For the first term, we compute

$$
\begin{aligned}
\mathbb{E}_{k,v}^{(t)}\left\{\left(1-[\bar{v}_k^{(t)}]^2\right)[\bar{v}_k^{(t)}]^4\right\} &= \mathbb{E}_{k,v}^{(t)}\left\{\left(1-[\bar{v}_k^{(t)}]^2\right)\left(1-\left(1+[\bar{v}_k^{(t)}]^4\right)\right)\right\} \\
&= \mathbb{E}_{k,v}^{(t)}\left\{1-[\bar{v}_k^{(t)}]^2\right\} - \mathbb{E}_{k,v}^{(t)}\left\{\left(1-[\bar{v}_k^{(t)}]^2\right)^2\left(1+[\bar{v}_k^{(t)}]^2\right)\right\} \\
&\geq \mathbb{E}_{k,v}^{(t)}\left\{1-[\bar{v}_k^{(t)}]^2\right\} - 2\mathbb{E}_{k,v}^{(t)}\left\{\left(1-[\bar{v}_k^{(t)}]^2\right)^2\right\} \\
&\geq \mathbb{E}_{k,v}^{(t)}\left\{1-[\bar{v}_k^{(t)}]^2\right\} - O(\alpha^3).
\end{aligned}
$$

Thus,

$$
\mathbb{E}_{k,v}^{(t)}G_1 \geq 8\tilde{a}_k^{(t)}\mathbb{E}_{k,v}^{(t)}\left\{1-[\bar{v}_k^{(t)}]^2\right\} - O\left(\hat{a}_k^{(t)}\alpha^3\right).
$$

- **Upper bound for $\mathbb{E}_{k,v}^{(t)}|G_2|$ and $\mathbb{E}_{k,v}^{(t)}|G_2|$.** It follows from Lemma A.10 that both terms are $O(1)\sum_{i\neq k}\hat{a}_i^{(t)}\alpha^3$.

Combine these two bounds together, absorb $m\delta_1^2$ into $O(\alpha^3)$, and we complete the proof. $\square$

**Lemma A.15** (Bound for the norm fluctuation)**.** *Suppose that at time $t$, Proposition 1 is true and $S_k^{(t)}\neq\varnothing$. Then at time $t$, we have*

$$
4\mathbb{E}_{k,v}^{(t)}\left[\left((T^*-T^{(t)})([\bar{v}^{(t)}]^{\otimes 4})\right)\left([\bar{v}_k^{(t)}]^2\right)\right] - 4\left(\mathbb{E}_{k,v}^{(t)}(T^*-T^{(t)})([\bar{v}^{(t)}]^{\otimes 4})\right)\left(\mathbb{E}_{k,v}^{(t)}[\bar{v}_k^{(t)}]^2\right) \geq -O(\alpha^3)
$$

*Proof.* We can express $(T^* - T^{(t)})([\bar{v}^{(t)}]^{\otimes 4})$ as follows:

$$(T^* - T^{(t)})([\bar{v}^{(t)}]^{\otimes 4})$$

$$=(a_k - \hat{a}_k^{(t)})[\bar{v}_k^{(t)}]^4 + \hat{a}_k^{(t)}\left([\bar{v}_k^{(t)}]^4 - \mathbb{E}_{k,w}^{(t)}\left\langle \bar{w}^{(t)}, \bar{v}^{(t)}\right\rangle^4\right) + \sum_{i \neq k} a_i[\bar{v}_i^{(t)}]^4 - \sum_{i \neq k} \hat{a}_i^{(t)}\mathbb{E}_{i,w}^{(t)}\left\langle \bar{w}^{(t)}, \bar{v}^{(t)}\right\rangle^4 \pm O(m\delta_1^2)$$

It's clear that $\mathbb{E}_{k,v}^{(t)}\sum_{i \neq k} a_i[\bar{v}_i^{(t)}]^4 = O(\alpha^3)$ and $\mathbb{E}_{k,v}^{(t)}\sum_{i \neq k} \hat{a}_i^{(t)}\mathbb{E}_{i,w}^{(t)}\left\langle \bar{w}^{(t)}, \bar{v}^{(t)}\right\rangle^4 = O(\alpha^3)$, so their influence can be bounded by $O(\alpha^3)$. Let's then focus on the first two terms in $(T^* - T^{(t)})([\bar{v}^{(t)}]^{\otimes 4})$. For the first term, we have

$$4\mathbb{E}_{k,v}^{(t)}(a_k - \hat{a}_k^{(t)})[\bar{v}_k^{(t)}]^4[\bar{v}_k^{(t)}]^2 - 4\mathbb{E}_{k,v}^{(t)}(a_k - \hat{a}_k^{(t)})[\bar{v}_k^{(t)}]^4\mathbb{E}_{k,v}^{(t)}[\bar{v}_k^{(t)}]^2$$

$$=4(a_k - \hat{a}_k^{(t)})\left(\mathbb{E}_{k,v}^{(t)}[\bar{v}_k^{(t)}]^6 - \mathbb{E}_{k,v}^{(t)}[\bar{v}_k^{(t)}]^4\mathbb{E}_{k,v}^{(t)}[\bar{v}_k^{(t)}]^2\right) \geq 0.$$

Let's now turn our focus to the second term. Denote $x = \left\langle \bar{w}_{-k}^{(t)}, \bar{v}_{-k}^{(t)}\right\rangle$ and write $\left\langle \bar{w}^{(t)}, \bar{v}^{(t)}\right\rangle^4 = [\bar{w}_k^{(t)}]^4[\bar{v}_k^{(t)}]^4 + 4[\bar{w}^{(t)}]_k^3[\bar{v}_k^{(t)}]^3 x + O(x^2)$. Suppose $m = \mathbb{E}_{k,v}^{(t)}[\bar{v}_k^{(t)}]^2$, we know $m \in [1 - O(\alpha^2), 1]$. We also know that $[\bar{v}_k^{(t)}]^2 \in [1 - \alpha, 1]$ for every $\bar{v}^{(t)} \in S_i^{(t)}$, so we have $|[\bar{v}_k^{(t)}]^2 - m| = O(\alpha)$. We have

$$\left|\mathbb{E}_{k,v}^{(t)}\mathbb{E}_{k,w}^{(t)}([\bar{v}_k^{(t)}]^2 - m)[\bar{v}_k^{(t)}]^4(1 - [\bar{w}_k^{(t)}]^4)\right| = O(\alpha^3)$$

$$\left|\mathbb{E}_{k,v}^{(t)}\mathbb{E}_{k,w}^{(t)}([\bar{v}_k^{(t)}]^2 - m)(\bar{w}_k^{(t)}\bar{v}_k^{(t)})^3 x\right| = O(\alpha^3)$$

$$\mathbb{E}_{k,v}^{(t)}\mathbb{E}_{k,w}^{(t)} x^2 = O(\alpha^4)$$

Therefore,

$$4\mathbb{E}_{k,v}^{(t)}\left[\hat{a}_k^{(t)}\left([\bar{v}_k^{(t)}]^4 - \mathbb{E}_{k,w}^{(t)}\left\langle \bar{w}^{(t)}, \bar{v}^{(t)}\right\rangle^4\right)[\bar{v}_k^{(t)}]^2\right] - 4\mathbb{E}_{k,v}^{(t)}\hat{a}_k^{(t)}\left([\bar{v}_k^{(t)}]^4 - \mathbb{E}_{k,w}^{(t)}\left\langle \bar{w}^{(t)}, \bar{v}^{(t)}\right\rangle^4\right)\mathbb{E}_{k,v}^{(t)}[\bar{v}_k^{(t)}]^2$$

$$\geq -O(\hat{a}_k^{(t)}\alpha^3).$$

Combining the bounds for all four terms, we conclude that

$$4\mathbb{E}_{k,v}^{(t)}\left[(T^* - T^{(t)})([\bar{v}^{(t)}]^{\otimes 4})[\bar{v}_k^{(t)}]^2\right] - 4\mathbb{E}_{k,v}^{(t)}(T^* - T^{(t)})([\bar{v}^{(t)}]^{\otimes 4})\mathbb{E}_{k,v}^{(t)}[\bar{v}_k^{(t)}]^2 \geq -O(\alpha^3).$$

$\square$

**Lemma A.2.** *Suppose that at time t, Proposition 1 is true and $S_k^{(t)} \neq \varnothing$. Assuming $\delta_1^2 = O(\alpha^3/m)$, we have*

$$\frac{d}{dt}\mathbb{E}_{k,v}^{(t)}[\bar{v}_k^{(t)}]^2 \geq 8\tilde{a}_k^{(t)}(1 - \mathbb{E}_{k,v}^{(t)}[\bar{v}_k^{(t)}]^2) - O(\alpha^3).$$

*In particular, if $\lambda = \Omega(\alpha)$, then $\frac{d}{dt}\mathbb{E}_{k,v}^{(t)}[\bar{v}_k^{(t)}]^2 > 0$ when $\mathbb{E}_{k,v}^{(t)}[\bar{v}_k^{(t)}]^2 < 1 - \alpha^2/2$.*

*Proof.* It suffices to combine the previous three lemmas together. $\square$

## A.5 Condition (c): bounds for the residual

In this section, we consider condition (c) of Proposition 1. Again, we need to estimate the derivative of $\tilde{a}_k^{(t)}$ when $\tilde{a}_k^{(t)}$ touches the boundary.

**On the impulses** Similar to the average bound in condition (b), we need to take into consideration the impulses. For the lower bound on $\tilde{a}_k^{(t)}$, we only need to consider the impulses caused by the entrance of new components since the reinitialization will only increase $\tilde{a}_k^{(t)}$. By Proposition 1 and Assumption 1, the total amount of impulses is upper bounded by $m\delta_1^2$. At the beginning of epoch $s$, we have $\tilde{a}_k^{(t)} \geq \lambda/6 - (s-1)m\delta_1^2$, which is guaranteed by the induction hypothesis from the last epoch. (At the beginning of the first epoch, we have $\tilde{a}_k^{(t)} = a_k$). Thus, following Lemma A.5, it suffices to show that $\frac{\mathrm{d}}{\mathrm{d}t}\tilde{a}_k^{(t)} > 0$ when $\tilde{a}_k^{(t)} \leq \lambda/6$. The upper bound on $\tilde{a}_k^{(t)}$ can be proved in a similar fashion. The only difference is that now the impulses that matter are caused by the reinitialization, the total amount of which can again be bounded by $m\delta_1^2$.

**Lemma A.16.** *Suppose that at time $t$, Proposition 1 is true and no impulses happen at time $t$. Then we have*

$$\frac{1}{\hat{a}_k^{(t)}}\frac{\mathrm{d}}{\mathrm{d}t}\hat{a}_k^{(t)} = 2\sum_{i=1}^{d} a_i \mathbb{E}_{k,v}^{(t)}[\bar{v}_i^{(t)}]^4 - 2\sum_{i=1}^{d}\hat{a}_i^{(t)}\mathbb{E}_{k,v}^{(t)}\mathbb{E}_{i,w}^{(t)}[z^{(t)}]^4 - \lambda - O(m\delta_1^2).$$

*Proof.* Recall that $\hat{a}_k^{(t)} = \sum_{v^{(t)}\in S_k^{(t)}}\left\|v^{(t)}\right\|^2$ and Lemma A.6 implies that

$$\frac{\mathrm{d}}{\mathrm{d}t}\left\|v^{(t)}\right\|^2 = 2\sum_{i=1}^{d} a_i \left\|v^{(t)}\right\|^2 [\bar{v}_i^{(t)}]^4 - 2\sum_{i=1}^{d}\hat{a}_i^{(t)}\left\|v^{(t)}\right\|^2 \mathbb{E}_{i,w}^{(t)}\left\{[z^{(t)}]^4\right\}$$
$$- \lambda\left\|v^{(t)}\right\|^2 - \left\|v^{(t)}\right\|^2 O(m\delta_1^2).$$

Sum both sides and we complete the proof. $\qquad\square$

**Lemma A.17.** *Suppose that at time $t$, Proposition 1 is true and no impulses happen at time $t$. Assume $\delta_1^2 = O(\alpha^2/m)$. Then we have*

$$\frac{1}{\hat{a}_k^{(t)}}\frac{\mathrm{d}}{\mathrm{d}t}\hat{a}_k^{(t)} \leq 2\tilde{a}_k^{(t)} - \lambda + O(\alpha^2).$$

*In particular, when $\tilde{a}_k^{(t)} \leq \lambda/6$, we have $\frac{\mathrm{d}}{\mathrm{d}t}\hat{a}_k^{(t)} < 0$.*

*Proof.* By Lemma A.16, we have

$$\frac{1}{\hat{a}_k^{(t)}}\frac{\mathrm{d}}{\mathrm{d}t}\hat{a}_k^{(t)} \leq 2a_k - 2\hat{a}_k^{(t)}\mathbb{E}_{k,v}^{(t)}\mathbb{E}_{k,w}^{(t)}[z^{(t)}]^4 + 2\sum_{i\neq k} a_i\mathbb{E}_{k,v}^{(t)}[\bar{v}_i^{(t)}]^4 - \lambda.$$

By Lemma A.12, we have

$$2a_k - 2\hat{a}_k^{(t)}\mathbb{E}_{k,v}^{(t)}\mathbb{E}_{k,w}^{(t)}[z^{(t)}]^4 \leq 2\tilde{a}_k^{(t)} + O(a_k\alpha^2)$$

For each term in the summation, we have

$$\mathbb{E}_{k,v}^{(t)}[\bar{v}_i^{(t)}]^4 \leq \mathbb{E}_{k,v}^{(t)}\left\{\left(1 - [\bar{v}_k^{(t)}]^2\right)^2\right\} \leq \alpha\mathbb{E}_{k,v}^{(t)}\left\{1 - [\bar{v}_k^{(t)}]^2\right\} \leq \alpha^3.$$

Thus,

$$\frac{1}{\hat{a}_k^{(t)}}\frac{\mathrm{d}}{\mathrm{d}t}\hat{a}_k^{(t)} \leq 2\tilde{a}_k^{(t)} + O(a_k\alpha^2) + 2\sum_{i\neq k} a_i^2\alpha^3 - \lambda$$
$$\leq 2\tilde{a}_k^{(t)} - \lambda + O(\alpha^2).$$

$\qquad\square$

**Lemma A.18.** *Suppose that at time $t$, Proposition 1 is true. and no impulses happen at time $t$. Then at time $t$, we have*

$$\frac{1}{\hat{a}_k^{(t)}}\frac{\mathrm{d}}{\mathrm{d}t}\hat{a}_k^{(t)} \geq 2\tilde{a}_k^{(t)} - \lambda - O\left(\alpha^2\right).$$

*In particular, when $\tilde{a}_k^{(t)} \geq \lambda$, we have $\frac{\mathrm{d}}{\mathrm{d}t}\hat{a}_k^{(t)} > 0$.*

*Proof.* By Lemma A.16 (and the fact $\hat{a}_i^{(t)} \leq a_i$), we have

$$\frac{1}{\hat{a}_k^{(t)}} \frac{\mathrm{d}}{\mathrm{d}t} \hat{a}_k^{(t)} \geq 2a_k \mathbb{E}_{k,v}^{(t)}[\bar{v}_k^{(t)}]^4 - 2\hat{a}_k^{(t)} - 2\sum_{i\neq k} a_i \mathbb{E}_{k,v}^{(t)} \mathbb{E}_{i,w}^{(t)}[z^{(t)}]^4 - \lambda - O(m\delta_1^2).$$

Note that $\mathbb{E}_{k,v}^{(t)}[\bar{v}_k^{(t)}]^4 \geq 1 - O(\alpha^2)$, whence

$$2a_k \mathbb{E}_{k,v}^{(t)}[\bar{v}_k^{(t)}]^4 - 2\hat{a}_k^{(t)} \geq 2\tilde{a}_k^{(t)} - O\left(a_k\alpha^2\right).$$

For each term in the summation, by Lemma A.10, we have $\mathbb{E}_{k,v}^{(t)} \mathbb{E}_{i,w}^{(t)}[z^{(t)}]^4 \leq O(\alpha^3)$. Thus,

$$\frac{1}{\hat{a}_k^{(t)}} \frac{\mathrm{d}}{\mathrm{d}t} \hat{a}_k^{(t)} \geq 2\tilde{a}_k^{(t)} - \lambda - O\left(\alpha^2\right).$$

$\square$

### A.6 Counterexample

We prove Claim 2 as follows.

**Claim 2.** *Suppose $T^* = e_k^{\otimes 4}$ and $T = v^{\otimes 4}/\|v\|^2 + w^{\otimes 4}/\|w\|^2$ with $\|w\|^2 + \|v\|^2 \in [2/3, 1]$. Suppose $\bar{v}_k^2 = 1 - \alpha$ and $\bar{v}_k = \bar{w}_k, \bar{v}_{-k} = -\bar{w}_{-k}$. Assuming $\|v\|^2 \leq c_1$ and $\alpha \leq c_2$ for small enough constants $c_1, c_2$, we have $\frac{\mathrm{d}}{\mathrm{d}t}\bar{v}_k^2 < 0$.*

*Proof.* Similar as in Lemma A.7, we can compute $\frac{\mathrm{d}}{\mathrm{d}t}\bar{v}_k^2$ as follows,

$$\begin{aligned}
\frac{\mathrm{d}}{\mathrm{d}t}\bar{v}_k^2 =& 8(1 - \bar{v}_k^2)\bar{v}_k^4 \\
& - 8(1 - \bar{v}_k^2)\left(\|v\|^2 \langle\bar{v},\bar{v}\rangle^4 + \|w\|^2 \langle\bar{w},\bar{v}\rangle^4\right) \\
& + 8\left(\|w\|^2 \langle\bar{w},\bar{v}\rangle^3 \langle\bar{w}_{-k},\bar{v}_{-k}\rangle + \|v\|^2 \langle\bar{v},\bar{v}\rangle^3 \langle\bar{v}_{-k},\bar{v}_{-k}\rangle\right).
\end{aligned}$$

Since $\bar{v}_k^2 = 1 - \alpha, \bar{v}_k = \bar{w}_k$ and $\bar{v}_{-k} = -\bar{w}_{-k}$, we have $\langle\bar{w},\bar{v}\rangle^4, \langle\bar{w},\bar{v}\rangle^3 \geq 1 - O(\alpha)$ and $\langle\bar{w}_{-k},\bar{v}_{-k}\rangle = -\alpha$. Therefore, we have

$$\frac{\mathrm{d}}{\mathrm{d}t}\bar{v}_k^2 \leq 8\alpha - 8\alpha(\|v\|^2 + \|w\|^2(1 - O(\alpha))) - 8\|w\|^2(1 - O(\alpha))\alpha + 8\|v\|^2\alpha$$

We have

$$\frac{\mathrm{d}}{\mathrm{d}t}\bar{v}_k^2 \leq 8\alpha\left((1 - \|w\|^2 - \|v\|^2) - \|w\|^2(1 - O(\alpha)) + \|v\|^2\right) < 0,$$

where the last inequality assumes $\|w\|^2 + \|v\|^2 \in [2/3, 1]$ and $\|v\|^2, \alpha$ smaller than certain constant.

$\square$

## B   Proofs for (Re)-initialization and Phase 1

We specify the constants that will be used in the proof of initialization (Section B.1) and Phase 1 (Section B.2). We will assume it always hold in the proof of Section B.1 and Section B.2. We omit superscript $s$ for simplicity.

**Proposition 2** (Choice of parameters). *The following hold with proper choices of constants $\gamma, c_e, c_\rho, c_{max}, c_t$*

  *1. $t_1' := \frac{c_t d}{8\beta \log d} \leq t_1 \leq \frac{(1-\gamma)}{8\beta c_e} \cdot \frac{d}{\log d}$,*

  *2. $\Gamma_i = \frac{1}{8a_i t_1'}$ if $S_i^{(s,0)} = \varnothing$, and $\Gamma_i = \frac{1}{8\lambda t_1'}$ otherwise. $\rho_i = c_\rho \Gamma_i$. $\Gamma_{max} = c_{max} \log d/d$.*

  *3. $c_e < \frac{c_\rho c_{max}}{2(1-c_\rho)}$, $c_\rho/c_t > 4c_e$, $c_t c_{max} \geq 4$.*

  *4. $c_a = (1 - c_\rho)/(c_t c_{max})$*

*Proof.* The results hold if let $\gamma, c_e, c_\rho, c_t$ be small enough constant and $c_{max}$ be large enough constant. For example, we can choose $c_e < c_\rho/4 < 0.01$, $c_t, \gamma < 0.01$ and $c_{max} > 10/c_t$. $\square$

## B.1 Initialization

We give a more detailed version of initialization with specified constants to fit the definition of $S_{good}$, $S_{pot}$ and $S_{bad}$. We show that at the beginning of any epoch $s$, the following conditions hold with high probability. Intuitively, it suggests all directions that we will discover satisfy $a_i = \Omega(\beta)$ as $S_{i,pot} \neq \varnothing$.

**Lemma B.1** ((Re-)Initialization space). *In the setting of Theorem 1, the following hold at the beginning of current epoch with probability $1 - 1/poly(d)$.*

1. *For all $a_i - \hat{a}_i^{(0)} \geq \beta$, we have $S_{i,good} \neq \varnothing$.*

2. *For all $a_i - \hat{a}_i^{(0)} < \beta c_a$, we have $S_{i,pot} = \varnothing$.*

3. *$S_{bad} = \varnothing$*

4. *$\left\| v^{(0)} \right\|_2 = \Theta(\delta_0)$, $[\bar{v}_i^{(0)}]^2 \leq \Gamma_{max} = c_{max} \log d/d$*

5. *For every $v$, there are at most $O(\log d)$ many $i \in [d]$ such that $[\bar{v}_i^{(0)}]^2 \geq c_e \log(d)/(10d)$.*

6. *$|\{v|v \text{ was reinitialized in epoch } s\}| = (1 - O(1/\log^2 d))m$.*

*Proof.* Let the constants in Lemma B.2 be $\eta = 1/c_t$, $c_i = \Gamma_i d/\log d$ and satisfy Proposition 2, then we know at the time of (re-)initialization, all statements hold. Since we further know from Lemma 6 that $\|v\| = \Theta(\delta_0)$ and $\bar{v}_i^2$ will only change $o(\log d/d)$, we have at the beginning of every epoch, all statements hold. $\square$

**Lemma B.2.** *There exist $m_0 = poly(d)$ and $m_1 = poly(d)$ such that if $m \in [m_0, m_1]$ and we random sample $m$ vectors $v$ from $Unif(\mathbb{S}^{d-1})$, with probability $1 - 1/poly(d)$ the following hold with proper absolute constant $\eta$, $\gamma$, $c_\rho$, $c_i$, $c_e$, $c_{max}$ satisfying $\eta(1 - \gamma) \leq c_i$, $c_{max} \geq 4\eta$, $\gamma, c_\rho$ are small enough and $c_{max}, \eta$ are large enough*

1. *For every $i \in [d]$ such that $c_i \leq \eta$, there exists $v$ such that $[\bar{v}_i^{(0)}]^2 \geq c_i(1 + 2c_\rho) \log d/d$ and $[\bar{v}_j^{(t)}]^2 \leq c_j(1 - 2c_\rho) \log d/d$ for $j \neq i$.*

2. *For every $v$, there does not exist $i \neq j$ such that $[\bar{v}_i^{(0)}]^2 \geq c_i(1 - 2c_\rho) \log d/d$ and $[\bar{v}_j^{(0)}]^2 \geq c_j(1 - 2c_\rho) \log d/d$.*

3. *For every $v$ and $i \in [d]$, $[\bar{v}_i^{(0)}]^2 \leq c_{max} \log d/2d$.*

4. *For every $v$, there are at most $O(\log d)$ many $i \in [d]$ such that $[\bar{v}_i^{(0)}]^2 \geq c_e \log(d)/11d$.*

5. *$|\{v|\text{there exists } i \in [d] \text{ such that } [\bar{v}_i^{(0)}]^2 \geq c_i(1 - 2c_\rho) \log d/d\}| \leq m/\log^2(d)$.*

*Proof.* It is equivalent to consider sample $v$ from $\mathcal{N}(0, I)$. Let $x \in \mathbb{R}$ be a standard Gaussian variable, according to Proposition 2.1.2 in Vershynin (2018), we have for any $t > 0$

$$\left(\frac{2}{t} - \frac{2}{t^3}\right) \cdot \frac{1}{\sqrt{2\pi}} e^{-t^2/2} \leq \Pr\left[x^2 \geq t^2\right] \leq \frac{2}{t} \cdot \frac{1}{\sqrt{2\pi}} e^{-t^2/2}.$$

Therefore, for any $i \in [d]$, we have for any constant $c > 0$

$$\Pr\left[v_i^2 \geq c \log(d)\right] = \Theta(d^{-c/2} \log^{-1/2} d).$$

According to Theorem 3.1.1 in Vershynin (2018), we know with probability at least $1 - 2\exp(-\Omega(d))$, $(1 - r)d \leq \|v\|^2 \leq (1 + r)d$ for any constant $0 < r < 1$. Hence, we have

$$\Pr\left[\bar{v}_i^2 \geq \frac{c \log(d)}{d}\right] \geq \Theta(d^{-c(1+r)/2} \log^{-1/2} d),$$

$$\Pr\left[\bar{v}_i^2 \geq \frac{c \log(d)}{d}\right] \leq \Theta(d^{-c(1-r)/2} \log^{-1/2} d).$$

**Part 1.** For fixed $i \in [d]$ such that $\eta(1 - \gamma) \leq c_i \leq \eta$, we have

$$\Pr\left[\bar{v}_i^2 \geq c_i(1 + 2c_\rho)\log(d)/d\right] \geq \Theta(d^{-c_i(1+2c_\rho)(1+r)/2}\log^{-1/2}d),$$

For a given $j \neq i$, we have

$$\Pr\left[\bar{v}_i^2 \geq c_i(1 + 2c_\rho)\log(d)/d, \ \bar{v}_j^2 \geq c_j(1 - 2c_\rho)\log(d)/d\right]$$
$$\leq \Theta(d^{-c_i(1+2c_\rho)(1-r)/2 - c_j(1-2c_\rho)(1-r)/2}) = O(d^{-\eta(1-\gamma)(1-r)}).$$

Since $c_i \leq \eta$, we know the desired event happens with probability $\Theta(d^{-\eta(1+2c_\rho)(1+r)/2} - d^{-\eta(1-\gamma)(1-r)+1})$. Since $\gamma, c_\rho$ are small enough constant, when $m_0 \geq \Omega(d^{\eta(1+2c_\rho)(1+r)/2+1})$, with probability $1 - O(e^{-d})$ there exists at least one $v$ such that $\bar{v}_i^2 \geq c_i(1 + 2c_\rho)\log(d)$ and $[\bar{v}_j^{(t)}]^2 \leq c_j(1 - 2c_\rho)\log d/d$ for $j \neq i$. Take the union bound for all $i \in [d]$, we know when $m_0 \geq \Omega(d^{\eta(1+2c_\rho)(1+r)/2+2})$, the desired statement holds with probability $1 - O(de^{-d})$.

**Part 2.** For any given $i \neq j$, we have

$$\Pr\left[[\bar{v}_i^{(0)}]^2 \geq c_i(1 - 2c_\rho)\log d/d, \ [\bar{v}_j^{(0)}]^2 \geq c_j(1 - 2c_\rho)\log d/d\right] \leq O(d^{-(c_i+c_j)(1-2c_\rho)(1-r)/2}).$$

Since $\eta(1 - \gamma) \leq c_i$, the probability that there exist $i \neq j$ such that the above happens is at most $O(d^{-\eta(1-\gamma)(1-2c_\rho)(1-r)+2})$. Thus, with $m_1 \leq O(d^{\eta(1-\gamma)(1-2c_\rho)(1-r)-2}/\text{poly}(d))$, the desired statement holds with probability $1 - 1/\text{poly}(d)$.

**Part 3.** We know

$$\Pr\left[\text{for all } i \in [d], \ \bar{v}_i^2 \leq c_{max}\log d/2d\right] \geq 1 - O(d^{-c_{max}(1-r)/4+1}).$$

With $m_1 \leq O(d^{c_{max}(1-r)/4-1}/\text{poly}(d))$ the desired statement holds with probability $1 - 1/\text{poly}(d)$.

**Part 4.** Since $m \leq m_1 = \text{poly}(d)$, we know for any constant $c_e$, this statement holds with probability $1 - O(e^{-\log^2 d})$.

**Part 5.** We have

$$\Pr\left[\text{there exists } i \in [d] \text{ such that } [\bar{v}_i^{(0)}]^2 \geq c_i(1 - 2c_\rho)\log d/d\right] \leq O(d^{-c_i(1-2c_\rho)/2+1}).$$

Let $p$ be the above probability and set $A$ as the $v$ satisfy above condition, by Chernoff's bound we have

$$\Pr\left[|A| \geq m/\log^2 d\right] \leq e^{-pm}\left(\frac{epm}{m/\log^2 d}\right)^{m/\log^2 d} = O(e^{-d}).$$

Combine all parts above, we know as long as $r, \gamma, c_\rho$ are small enough, $c_{max} \geq 4\eta$ and $\eta$ is large enough, we have when $m_0 \geq \Omega(d^{0.6\eta})$ and $m_1 \leq O(d^{0.9\eta})$, the results hold. $\qquad\square$

## B.2 Proof of Phase 1

In this section, we first give a proof overview of Phase 1 and then give the detailed proof for each lemma in later subsections.

### B.2.1 Proof overview

We give the proof overview in this subsection and present the proof of Lemma 5 and Lemma 4 at the end of this subsection. We remark that the proof idea in this phase is inspired by (Li et al., 2020a).

We describe the high-level proof plan for phase 1. Recall that at the beginning of this epoch, we know $S_{bad} = \varnothing$ which implies there is at most one large coordinate for every component. Roughly speaking, we will show that for those small coordinate they will remain small in phase 1, and the only possibility for one component to have larger norm is to grow in the large direction. This intuitively suggests all components that have a relatively large norm in phase 1 are basis-like components.

We first show within $t_1' = c_t d/(8\beta \log d))$ time, there are components that can improve their correlation with some ground truth component $e_i$ to a non-trivial $\mathrm{polylog}(d)/d$ correlation. This lemma suggests that there is at most one coordinate can grow above $O(\log d/d)$.

Note that we should view the analysis in this section and the analysis in Appendix A as a whole induction/continuity argument. It's easy to verify that at any time $0 \le t \le t_1^{(s)}$, Assumption 1 holds and Proposition 1 holds.

**Lemma B.3.** *In the setting of Lemma 4, suppose $\left\|\bar{v}^{(0)}\right\|_\infty^2 \le \log^4(d)/d$. Then, for every $k \in [d]$*

    *1. for $v \notin S_{pot}$, $[\bar{v}_i^{(t)}]^2 = O(\log(d)/d)$ for all $i \in [d]$ and $t \le t_1'$.*

    *2. if $S_k^{(t)} = \varnothing$ for $t \le t_1'$, then for $v \in S_{k,good}$, there exists $t \le t_1'$ such that $[\bar{v}_k^{(t)}]^2 \ge \log^4(d)/d$ and $[\bar{v}_i^{(t)}]^2 = O(\log(d)/d)$ for all $i \ne k$.*

    *3. for $v \in S_{k,pot} \setminus (S_{good} \cup S_{bad})$, $[\bar{v}_i^{(t)}]^2 = O(\log(d)/d)$ for all $i \ne k$ and $t \le t_1'$.*

The above lemma is in fact a direct corollary from the following lemma when considering the definition of $S_{good}$ and $S_{pot}$. It says if a direction is below certain threshold, it will remain $O(\log d/d)$, while if a direction is above certain threshold and there are no basis-like components for this direction, it will grow to have a $\mathrm{polylog}(d)$ improvement.

**Lemma B.4.** *In the setting of Lemma 4, we have*

    *1. if $[\bar{v}_k^{(0)}]^2 \le \min\{\Gamma_k - \rho_k, \Gamma_{max}\}$, then $[\bar{v}_k^{(t)}]^2 = O(\log(d)/d)$ for $t \le t_1'$.*

    *2. if $S_k^{(t)} = 0$ for $t \le t_1'$, $[\bar{v}_k^{(0)}]^2 \ge \Gamma_k + \rho_k$, $[\bar{v}_i^{(0)}]^2 \le \Gamma_i - \rho_i$ for all $i \ne k$ and $\left\|\bar{v}^{(0)}\right\|_\infty^2 \le \log^4(d)/d$, then there exists $t \le t_1'$ such that $[\bar{v}_k^{(t)}]^2 \ge \log^4(d)/d$.*

The following lemma shows if $[\bar{v}_i^{(t_1')}]^2 = O(\log d/d)$ at $t_1'$, it will remain $O(\log d/d)$ to the end of phase 1. This implies for components that are not in $S_{pot}$, they will not have large correlation with any ground truth component in phase 1.

**Lemma B.5.** *In the setting of Lemma 4, suppose $[\bar{v}_i^{(t_1')}]^2 = O(\log(d)/d)$. Then we have $[\bar{v}_i^{(t)}]^2 = O(\log(d)/d)$ for $t_1' \le t \le t_1$.*

The following two lemmas show good components (those have $\mathrm{polylog}(d)/d$ correlation before $t_1'$) will quickly grow to have constant correlation and $\delta_1$ norm. Note that the following condition $a_k = \Omega(\beta)$ holds in our setting because when $a_i < \beta c_a$, we have $S_{i,good} = S_{i,pot} = \varnothing$ (this means for those small directions there are no components that can have $\mathrm{polylog}(d)/d$ correlation as shown in Lemma B.3).

**Lemma B.6** (Good component, constant correlation). *In the setting of Lemma 4, suppose $S_k^{(t)} = \varnothing$ for $t \le t_1$, $a_k = \Omega(\beta)$. If there exists $\tau_0 \le t_1$ such that $[\bar{v}_k^{(\tau_0)}]^2 > \log^4(d)/d$ and $[\bar{v}_i^{(\tau_0)}]^2 = O(\log(d)/d)$ for all $i \ne k$, then for any constant $c \in (0,1)$ we have $[\bar{v}_k^{(t)}]^2 > c$ and $[\bar{v}_i^{(t)}]^2 = O(\log(d)/d)$ for all $i \ne k$ when $\tau_0 + t_1'' \le t \le t_1$ with $t_1'' = \Theta(d/(\beta \log^3 d))$.*

**Lemma B.7** (Good component, norm growth). *In the setting of Lemma 4, suppose $S_k^{(t)} = \varnothing$ for $t \le t_1$, $a_k = \Omega(\beta)$. If there exists $\tau_0' \le t_1$ such that $[\bar{v}_k^{(\tau_0')}]^2 > c$ and $[\bar{v}_i^{(\tau_0')}]^2 = O(\log(d)/d)$ for all $i \ne k$, then we have $\left\|v^{(t)}\right\|_2 \ge \delta_1$ for some $\tau_0' \le t \le \tau_0' + t_1'''$ with $t_1''' = \Theta(\log(d/\alpha)/\beta)$.*

Recall from Lemma B.4 we know there is at most one coordinate that can be large. Thus, intuitively we can expect if the norm is above certain threshold, the component will become basis-like, since this large direction will contribute most of the norm and other directions will remain small. In fact, we can show (1) norm of "small and dense" components (e.g., those are not in $S_{pot}$) is smaller than $\delta_1$; (2) once a component reaches norm $\delta_1$, it is a basis-like component.

**Lemma B.8.** *In the setting of Lemma 4, we have*

    *1. if $\left\|\bar{v}^{(t)}\right\|_\infty^2 \le \log^4(d)/d$ for all $t \le t_1$, then $\left\|v^{(t)}\right\|_2 = O(\delta_0)$ for all $t \le t_1$.*

2. Let $\tau_0 = \inf\{t \in [0, t_1] | \|\bar{v}^{(t)}\|_\infty^2 \geq \log^4 d/d\}$. Suppose $[\bar{v}_k^{(\tau_0)}]^2 \geq \log^4 d/d$ and $[\bar{v}_i^{(\tau_0)}]^2 = O(\log d/d)$ for $i \neq k$. If there exists $\tau_1$ such that $\tau_0 < \tau_1 \leq t_1$ and $\|v^{(\tau_1)}\|_2 \geq \delta_1$ for the first time, then there exists $k \in [d]$ such that $[\bar{v}_k^{(\tau_1)}]^2 \geq 1 - \alpha^2$ if $\hat{a}_k^{(t)} \leq \alpha$ for $t \leq \tau_1$ and $[\bar{v}_k^{(\tau_1)}]^2 \geq 1 - \alpha$ otherwise.

One might worry that a component can first exceeds the $\delta_1$ threshold then drop below it and eventually gets re-initialized. Next, we show that re-initialization at the end of Phase 1 cannot remove all the components in $S_k^{(t_1)}$.

**Lemma B.9.** *If $S_k^{(0)} = \varnothing$ and $S_k^{(t')} \neq \varnothing$ for some $t' \in (0, t_1]$, we have $S_k^{(t_1)} \neq \varnothing$ and $\hat{a}_k^{(t_1)} \geq \delta_1^2$.*

Given above lemma, we now are ready to prove Lemma 5 and the main lemma for Phase 1.

**Lemma 5.** *In the setting of Lemma 4, for every $i \in [d]$*

1. *(Only good/potential components can become large) If $v^{(s,t)} \notin S_{pot}^{(s)}$, $\|v^{(s,t)}\| = O(\delta_0)$ and $[\bar{v}_i^{(s,t)}]^2 = O(\log(d)/d)$ for all $i \in [d]$ and $t \leq t_1^{(s)}$.*

2. *(Good components discover ground truth components) If $S_{i,good}^{(s)} \neq \varnothing$, there exists $v^{(s,t_1^{(s)})}$ such that $\left\| v^{(s,t_1^{(s)})} \right\| \geq \delta_1$ and $S_i^{(s,t_1^{(s)})} \neq \varnothing$.*

3. *(Large components are correlated with ground truth components) If $\left\| v^{(s,t)} \right\| \geq \delta_1$ for some $t \leq t_1^{(s)}$, there exists $i \in [d]$ such that $v^{(s,t)} \in S_i^{(s,t)}$.*

*Proof.* We show statements one by one.

**Part 1.** The statement follows from Lemma B.3, Lemma B.5 and Lemma B.8.

**Part 2.** Suppose $S_k^{(t)} = \varnothing$ for all $t \leq t_1$. By Lemma B.1 we know $S_{k,good} \neq \varnothing$. Then by Lemma B.3, Lemma B.6 and Lemma B.7, we know there exists $v$ such that $\|v^{(t)}\|_2 \geq \delta_1$ within time $t_1 = t_1' + t_1'' + t_1'''$. Then by Lemma B.8 we know $[\bar{v}_k^{(t)}]^2 \geq 1 - \alpha$. Therefore, we know there exists $t \leq t_1$ such that $S_k^{(t)} \neq \varnothing$. Finally we know it will keep until $t_1$ by Lemma B.9.

**Part 3.** The statement directly follows from Lemma B.8 and Lemma B.9. $\qquad\square$

**Lemma 4** (Main Lemma for Phase 1). *In the setting of Theorem 1, suppose Proposition 1 holds at $(s, 0)$. For $t_1^{(s)} := t_1^{(s)\prime} + t_1^{(s)\prime\prime} + t_1^{(s)\prime\prime\prime}$ with $t_1^{(s)\prime} = \Theta(d/(\beta^{(s)} \log d))$, $t_1^{(s)\prime\prime} = \Theta(d/(\beta^{(s)} \log^3 d))$, $t_1^{(s)\prime\prime\prime} = \Theta(\log(d/\alpha)/\beta^{(s)})$, with probability $1 - 1/poly(d)$ we have*

1. *Proposition 1 holds at $(s, t)$ for any $0 \leq t < t_1^{(s)}$, and also for $t = t_1^{(s)}$ after reinitialization.*

2. *If $a_k \geq \beta^{(s)}$ and $S_k^{(s,0)} = \varnothing$, we have $S_k^{(s,t_1^{(s)})} \neq \varnothing$ and $\hat{a}_k^{(s,t_1^{(s)})} \geq \delta_1^2$.*

3. *If $S_k^{(s,0)} = \varnothing$ and $S_k^{(s,t_1^{(s)})} \neq \varnothing$, we have $a_k \geq C\beta^{(s)}$ for universal constant $0 < C < 1$.*

*Proof.* By Lemma B.1 we know the number of reinitialized components are always $\Theta(m)$ so Lemma B.1 holds with probability $1 - 1/poly(d)$ for every epoch. In the following assume Lemma B.1 holds. The second and third statement directly follow from Lemma B.1 and Lemma 5 as $S_{k,pot} = \varnothing$ when $a_k \leq \beta c_a$. For the first statement, combing the proof in Appendix A and Lemma B.8, we know the statement holds (see also the remark at the beginning of Appendix A). $\qquad\square$

### B.2.2 Preliminary

To simplify the proof in this section, we introduce more notations and give the following lemma.

**Lemma B.10.** *In the setting of Lemma 4, we have* $T^* - T^{(t)} = \sum_{i \in [d]} \tilde{a}_i^{(t)} e_i^{\otimes 4} + \Delta^{(t)}$, *where* $\tilde{a}_i^{(t)} = a_i - \hat{a}_i^{(t)}$ *and* $\|\Delta\|_F = O(\alpha + m\delta_1^2)$. *We know* $\tilde{a}_i^{(0)} = a_i$ *if* $S_i^{(s,0)} = \varnothing$ *and* $\tilde{a}_i^{(t)} = \Theta(\lambda)$ *if* $S_i^{(s,0)} \neq \varnothing$. *That is, the residual tensor is roughly the ground truth tensor* $T^*$ *with unfitted directions at the beginning of this epoch and plus a small perturbation* $\Delta$.

*Proof.* We can decompose $T^{(t)}$ as

$$T^{(t)} = \sum_{i \in [d]} T_i^{(t)} + T_\varnothing^{(t)} = \sum_{i \in [d]} \left( \hat{a}_i^{(t)} e_i^{\otimes 4} + (T_i^{(t)} - \hat{a}_i^{(t)} e_i^{\otimes 4}) \right) + T_\varnothing^{(t)},$$

where $T_i^{(t)} = \sum_{w \in S_i^{(t)}} \|w\|^2 \bar{w}^{\otimes 4}$ and $T_\varnothing^{(t)} = \sum_{w \in S_\varnothing^{(t)}} \|w\|^2 \bar{w}^{\otimes 4}$. Note that when $S_i^{(t)} = \varnothing$, $\hat{a}_i^{(t)} = 0$ and when $S_i^{(t)} \neq \varnothing$ we have $\left\| (T_i^{(t)} - \hat{a}_i^{(t)} e_i^{\otimes 4}) \right\|_F = O(\hat{a}_i^{(t)} \alpha)$ and $\left\| T_\varnothing^{(t)} \right\|_F \leq m\delta_1^2$. This gives the desired form of $T^* - T^{(t)}$.

$\square$

We give the dynamic of $[\bar{v}_k^{(t)}]^2$ and $[v_k^{(t)}]^2$ here, which will be frequently used in our analysis.

$$
\begin{aligned}
\frac{d[\bar{v}_k^{(t)}]^2}{dt} &= 2\bar{v}_k^{(t)} \cdot \frac{d}{dt} \frac{v_k^{(t)}}{\|v^{(t)}\|} \\
&= 2\bar{v}_k^{(t)} \cdot \frac{1}{\|v^{(t)}\|} \frac{d}{dt} v_k^{(t)} + 2[\bar{v}_k^{(t)}]^2 \cdot \frac{d}{dt} \frac{1}{\|v\|} \\
&= 2\bar{v}_k^{(t)} \cdot \frac{1}{\|v^{(t)}\|} [-\nabla L(v^{(t)})]_k - 2[\bar{v}_k^{(t)}]^2 \cdot \frac{\langle \bar{v}^{(t)}, -\nabla L(v^{(t)}) \rangle}{\|v^{(t)}\|} \\
&= 2\bar{v}_k^{(t)} \cdot \frac{1}{\|v^{(t)}\|} [-(I - \bar{v}^{(t)}[\bar{v}^{(t)}]^\top) \nabla L(v^{(t)})]_k \\
&= 8\bar{v}_k^{(t)} \left[ (T^* - T^{(t)})([\bar{v}^{(t)}]^{\otimes 3}, I) - (T^* - T^{(t)})([\bar{v}^{(t)}]^{\otimes 4}) \bar{v}^{(t)} \right]_k \\
&= 8[\bar{v}_k^{(t)}]^2 \left( \tilde{a}_k^{(t)}[\bar{v}_k^{(t)}]^2 - \sum_{i \in [d]} \tilde{a}_i^{(t)}[\bar{v}_i^{(t)}]^4 \pm \frac{\|\Delta^{(t)}\|_F}{|\bar{v}_k^{(t)}|} \right).
\end{aligned}
\tag{3}
$$

$$
\begin{aligned}
\frac{d[v_k^{(t)}]^2}{dt} &= 2v_k^{(t)} \cdot \frac{dv_k^{(t)}}{dt} \\
&= 2v_k^{(t)} \cdot [-\nabla L(v^{(t)})]_k \\
&= 4v_k^{(t)} \left[ 2(T^* - T^{(t)})([\bar{v}^{(t)}]^{\otimes 3}, I) \left\| v^{(t)} \right\|_2 - (T^* - T^{(t)})([\bar{v}^{(t)}]^{\otimes 4}) v^{(t)} \right]_k \\
&= 4[v_k^{(t)}]^2 \left( 2\tilde{a}_k^{(t)}[\bar{v}_k^{(t)}]^2 - \sum_{i \in [d]} \tilde{a}_i^{(t)}[\bar{v}_i^{(t)}]^4 \pm \frac{\|\Delta^{(t)}\|_F \|v^{(t)}\|_2}{|v_k^{(t)}|} \right).
\end{aligned}
\tag{4}
$$

The following lemma allows us to ignore these already fitted direction as they will remain as small as their (re-)initialization in phase 1.

**Lemma B.11.** *In the setting of Lemma 4, if direction* $e_k$ *has been fitted before current epoch (i.e.,* $S_k^{(s,0)} \neq \varnothing$*), then for* $v$ *that was reinitialized in the previous epoch, we have* $[\bar{v}_k^{(t)}]^2 = O(\log(d)/d)$ *for all* $t \leq t_1$.

*Proof.* Since direction $e_k$ has been fitted before current epoch, we know $\tilde{a}_k^{(t)} = \Theta(\lambda)$. We only need to consider the time when $[\bar{v}_k^{(t)}]^2 \geq \log d/d$. By (3) we have

$$\frac{d[\bar{v}_k^{(t)}]^2}{dt} = 8[\bar{v}_k^{(t)}]^2 \left( \tilde{a}_k^{(t)}[\bar{v}_k^{(t)}]^2 - \sum_{i \in [d]} \tilde{a}_i^{(t)}[\bar{v}_i^{(t)}]^4 \pm \frac{\|\Delta^{(t)}\|_F}{|\bar{v}_k^{(t)}|} \right) \leq [\bar{v}_k^{(t)}]^2 O\left( \lambda + d \left\| \Delta^{(t)} \right\|_F \right).$$

Since $\lambda$ and $\left\|\Delta^{(t)}\right\|_F = O(\alpha + m\delta_1^2)$ are small enough and $[\bar{v}_k^{(0)}]^2 = O(\log d/d)$, we know $[\bar{v}_k^{(t)}]^2 = O(\log d/d)$ for $t \le t_1$.

$\qquad\qquad\qquad\qquad\qquad\qquad\qquad\qquad\qquad\qquad\qquad\qquad\qquad\qquad\qquad\qquad\quad\square$

### B.2.3 Proof of Lemma B.3 and Lemma B.4

Lemma B.3 directly follows from Lemma B.4 and the definition of $S_{good}$, $S_{pot}$ and $S_{bad}$ as in Definition 2. We focus on Lemma B.4 in the rest of this section. We need following lemma to give the proof of Lemma B.4.

**Lemma B.12.** *In the setting of Lemma 4, if $\left\|\bar{v}^{(t)}\right\|_\infty^2 \le \log^4(d)/d$, we have $\sum_i [\bar{v}_i^{(t)}]^4 \le c_e \log d/d$ for all $t \le t_1$.*

*Proof.* We claim that for all $t \le t_1$, there are at most $O(\log d)$ many $i \in [d]$ such that $[\bar{v}_i^{(t)}]^2 \ge c_e \log(d)/2d$. Based on this claim, we know

$$\sum_{i \in [d]} [\bar{v}_i^{(t)}]^4 \le O(\log d)\frac{\log^8 d}{d^2} + \sum_{i:[\bar{v}_i^{(t)}]^2 < c_e \log(d)/2d} [\bar{v}_i^{(t)}]^4 \le O\left(\frac{\log^9 d}{d^2}\right) + \frac{c_e \log(d)}{2d} \le \frac{c_e \log(d)}{d},$$

which gives the desired result.

In the following, we prove the above claim. From Lemma B.1, we know when $t = 0$, the claim is true. For any $[\bar{v}_k^{(0)}]^2 \le c_e \log(d)/10d$, we will show $[\bar{v}_k^{(t)}]^2 \le c_e \log(d)/2d$ for all $t \le t_1$. By (3) we have

$$\frac{\mathrm{d}[\bar{v}_k^{(t)}]^2}{\mathrm{d}t} = 8[\bar{v}_k^{(t)}]^2 \left( \tilde{a}_k^{(t)}[\bar{v}_k^{(t)}]^2 - \sum_{i \in [d]} \tilde{a}_i^{(t)}[\bar{v}_i^{(t)}]^4 \pm \frac{\left\|\Delta^{(t)}\right\|_F}{|\bar{v}_k^{(t)}|} \right).$$

In fact, we only need to show that for any $\tau_0$ such that $[\bar{v}_k^{(\tau_0)}]^2 = c_e \log(d)/10d$ and $[\bar{v}_k^{(t)}]^2 \ge c_e \log(d)/10d$ when $\tau_0 \le t \le \tau_0 + t_1$, we have $[\bar{v}_k^{(t)}]^2 \le c_e \log(d)/2d$. To show this, we have

$$\frac{\mathrm{d}[\bar{v}_k^{(t)}]^2}{\mathrm{d}t} \le 8[\bar{v}_k^{(t)}]^2 \left( \tilde{a}_k^{(t)}[\bar{v}_k^{(t)}]^2 + \frac{\left\|\Delta^{(t)}\right\|_F}{|\bar{v}_k^{(t)}|} \right) \le [\bar{v}_k^{(t)}]^2 \cdot 16\tilde{a}_k^{(t)}[\bar{v}_k^{(t)}]^2 \le [\bar{v}_k^{(t)}]^2 \cdot \frac{\beta}{1-\gamma} \cdot \frac{8c_e \log(d)}{d},$$

where we use $\left\|\Delta^{(t)}\right\|_F = O(\alpha + m\delta_1^2)$ and $\tilde{a}_k^{(t)} \le \beta/(1-\gamma)$. Therefore, with our choice of $t_1$, we know $[\bar{v}_k^{(t)}]^2 \le c_e \log(d)/2d$. This finish the proof.

$\qquad\qquad\qquad\qquad\qquad\qquad\qquad\qquad\qquad\qquad\qquad\qquad\qquad\qquad\qquad\qquad\quad\square$

We now are ready to give the proof of Lemma B.4.

**Lemma B.4.** *In the setting of Lemma 4, we have*

1. *if $[\bar{v}_k^{(0)}]^2 \le \min\{\Gamma_k - \rho_k, \Gamma_{max}\}$, then $[\bar{v}_k^{(t)}]^2 = O(\log(d)/d)$ for $t \le t_1'$.*

2. *if $S_k^{(t)} = 0$ for $t \le t_1'$, $[\bar{v}_k^{(0)}]^2 \ge \Gamma_k + \rho_k$, $[\bar{v}_i^{(0)}]^2 \le \Gamma_i - \rho_i$ for all $i \ne k$ and $\left\|\bar{v}^{(0)}\right\|_\infty^2 \le \log^4(d)/d$, then there exists $t \le t_1'$ such that $[\bar{v}_k^{(t)}]^2 \ge \log^4(d)/d$.*

*Proof.* We focus on the dynamic of $[\bar{v}_k^{(t)}]^2$. For those already fitted direction $e_k$, we have $\Gamma_k = 1/(8\lambda t_1')$, which means $\Gamma_{max} \le \Gamma_k - \rho_k$. From Lemma B.11 we know $[\bar{v}_k^{(t)}]^2 = O(\log d/d)$ for $t \le t_1'$. In the rest of proof, we focus on these unfitted direction $e_k$. By (3) we have

$$\frac{\mathrm{d}[\bar{v}_k^{(t)}]^2}{\mathrm{d}t} = 8[\bar{v}_k^{(t)}]^2 \left( \tilde{a}_k^{(t)}[\bar{v}_k^{(t)}]^2 - \sum_{i \in [d]} \tilde{a}_i^{(t)}[\bar{v}_i^{(t)}]^4 \pm \frac{\left\|\Delta^{(t)}\right\|_F}{|\bar{v}_k^{(t)}|} \right)$$

**Part 1.** Define the following dynamics $p^{(t)}$,

$$\frac{\mathrm{d}p^{(t)}}{\mathrm{d}t} = 8p^{(t)}\left(a_k p^{(t)} + \frac{a_k c_e \log d}{d}\right), \quad p^{(0)} = [\bar{v}_k^{(0)}]^2$$

Given that $\tilde{a}_i^{(t)} \leq a_i$ and $\left\|\Delta^{(t)}\right\|_F = O(\alpha + m\delta_1^2)$ is small enough, it is easy to see $[\bar{v}_k^{(t)}]^2 \leq \max\{\log(d)/d, p^{(t)}\}$. Then it suffices to bound $p^{(t)}$ to have a bound for $[\bar{v}_k^{(t)}]^2$. Consider the following dynamic $x^{(t)}$

$$\frac{\mathrm{d}x^{(t)}}{\mathrm{d}t} = \tau_1 [x^{(t)}]^2, \quad x^{(0)} = \tau_2. \tag{5}$$

We know $x^{(t)} = 1/(1/\tau_2 - \tau_1 t)$. Set $\tau_1 = 8a_k$ and $\tau_2 = 1/(\tau_1 t_1') = \Gamma_k$. Then, with our choice of $\rho_k = c_\rho \Gamma_k$, we know

1. $p^{(0)} = [\bar{v}_k^{(0)}]^2 \leq \Gamma_k - \rho_k \leq \Gamma_{max}$. As long as $\rho_k \geq \frac{2c_e \log d}{d}$ and $x^{(0)} = p^{(0)} + \rho_k/2$, we have $p^{(t)} \leq x^{(t)} - \rho_k/2$ for $t \leq t_1'$. Therefore, $p^{(t_1')} \leq x^{(t_1')} \leq 2\Gamma_k^2/\rho_k = O(\log d/d)$.

2. $p^{(0)} = [\bar{v}_k^{(0)}]^2 \leq \Gamma_{max} < \Gamma_k - \rho_k$. As long as $x^{(0)} = p^{(0)} + \frac{c_e \log d}{d}$, we have $p^{(t)} \leq x^{(t)} - \frac{c_e \log d}{d}$ for $t \leq t_1'$. Therefore, $p^{(t_1')} \leq x^{(t_1')} = O(\log d/d)$.

Together we know $[\bar{v}_k^{(t)}]^2 = O(\log d/d)$ for $t \leq t_1'$.

**Part 2.** Define the following dynamics $q^{(t)}$,

$$\frac{\mathrm{d}q^{(t)}}{\mathrm{d}t} = 8q^{(t)}\left(a_k q^{(t)} - \frac{2\beta c_e \log d}{d}\right), \quad q^{(0)} = [\bar{v}_k^{(0)}]^2.$$

Since $S_k^{(t)} = \varnothing$, we know $\tilde{a}_k^{(t)} = a_k$. Given that $\left\|\Delta^{(t)}\right\|_F = O(\alpha + m\delta_1^2)$ and Lemma B.12, it is easy to see as long as $\left\|\bar{v}^{(t)}\right\|_\infty^2 \leq \log^4 d/d$, if $q^{(0)} \geq [\bar{v}_k^{(0)}]^2 \geq \Theta(\log d/d)$ and $a_k[q^{(0)}]^2 - \frac{2\beta c_e \log d}{d} > 0$, we have $[\bar{v}_k^{(t)}]^2 \geq q^{(t)}$. Then it suffices to bound $q^{(t)}$ to get a bound on $[v_k^{(t)}]^2$. Consider the same dynamic (5) with same $\tau_1$ and $\tau_2$, as long as $q^{(0)} = [\bar{v}_k^{(0)}]^2 \geq \Gamma_k + \rho_k$, $\rho_k \geq \frac{4\beta c_e \log d}{a_k d}$ and $x^{(0)} = q^{(0)} - \rho_k/2$, we have $q^{(t)} \geq x^{(t)} + \rho_k/2$ if $\left\|\bar{v}^{(t)}\right\|_\infty^2 \leq \log^4 d/d$ holds. We can verify that $x^{(T_1')} = +\infty$, which implies there exists $t \leq t_1'$ such that $\left\|\bar{v}^{(t)}\right\|_\infty^2 > \log^4 d/d$.

$\square$

### B.2.4 Proof of Lemma B.5

**Lemma B.5.** *In the setting of Lemma 4, suppose $[\bar{v}_i^{(t_1')}]^2 = O(\log(d)/d)$. Then we have $[\bar{v}_i^{(t)}]^2 = O(\log(d)/d)$ for $t_1' \leq t \leq t_1$.*

*Proof.* Recall $t_1 - t_1' = t_1'' + t_1''' = o(d/(\beta \log d))$, it suffices to show if $[\bar{v}_i^{(t_1')}]^2 = c_1 \log(d)/d$, then $[\bar{v}_i^{(t)}]^2$ will be at most $2c_1 \log(d)/d$ in $t_{max}' = o(d/(\beta \log d))$ time. Suppose there exists time $\tau_1 \leq t_{max}'$ such that $[\bar{v}_i^{(\tau_1)}]^2 \geq 2c_1 \log(d)/d$ for the first time. We only need to show if $[\bar{v}_i^{(t)}]^2 \geq c_1 \log(d)/d$ for $t \leq \tau_1$, we have $[\bar{v}_i^{(t)}]^2 < 2c_1 \log(d)/d$. We know the dynamic of $[\bar{v}_i^{(t)}]^2$

$$\frac{\mathrm{d}[\bar{v}_i^{(t)}]^2}{\mathrm{d}t} = 8[\bar{v}_i^{(t)}]^2\left(\tilde{a}_k^{(t)}[\bar{v}_i^{(t)}]^2 - \sum_{j\in[d]} \tilde{a}_j^{(t)}[\bar{v}_j^{(t)}]^4 \pm \frac{\left\|\Delta^{(t)}\right\|_F}{|\bar{v}_i^{(t)}|}\right) \leq [\bar{v}_i^{(t)}]^2 O\left(\frac{\beta \log d}{d}\right),$$

where we use $\left\|\Delta^{(t)}\right\|_F = O(\alpha + m\delta_1^2)$ is small enough and $\tilde{a}_k^{(t)} \leq 1$. This implies $[\bar{v}_i^{(t)}]^2 \leq 2c_1 \log d/d$ as $t_{max}' = o(d/(\beta \log d))$.

$\square$

### B.2.5 Proof of Lemma B.6

**Lemma B.6** (Good component, constant correlation). *In the setting of Lemma 4, suppose $S_k^{(t)} = \varnothing$ for $t \leq t_1$, $a_k = \Omega(\beta)$. If there exists $\tau_0 \leq t_1$ such that $[\bar{v}_k^{(\tau_0)}]^2 > \log^4(d)/d$ and $[\bar{v}_i^{(\tau_0)}]^2 = O(\log(d)/d)$ for all $i \neq k$, then for any constant $c \in (0,1)$ we have $[\bar{v}_k^{(t)}]^2 > c$ and $[\bar{v}_i^{(t)}]^2 = O(\log(d)/d)$ for all $i \neq k$ when $\tau_0 + t_1'' \leq t \leq t_1$ with $t_1'' = \Theta(d/(\beta \log^3 d))$.*

*Proof.* By Lemma B.5 we know $[\bar{v}_i^{(t)}]^2$ will remain $O(\log d/d)$ for those $[\bar{v}_i^{(\tau_0)}]^2 = O(\log d/d)$.

We now show $[\bar{v}_k^{(t)}]^2$ will become constant within $t_1''$ time. We know $\sum_{i \neq k} \tilde{a}_i^{(t)} [\bar{v}_i^{(t)}]^4 \leq \beta c_1 \log d/d$ for some constant $c_1$. Hence, with the fact $S_k^{(t)} = \varnothing$, $a_k = \Omega(\beta)$, $[\bar{v}_k^{(\tau_0)}]^2 > \log^4(d)/d$ and $\left\| \Delta^{(t)} \right\|_F = O(\alpha + m\delta_1^2)$,

$$\frac{\mathrm{d}[\bar{v}_k^{(t)}]^2}{\mathrm{d}t} = 8[\bar{v}_k^{(t)}]^2 \left( \tilde{a}_k^{(t)} [\bar{v}_k^{(t)}]^2 (1 - [\bar{v}_k^{(t)}]^2) - \sum_{i \neq k} \tilde{a}_i^{(t)} [\bar{v}_i^{(t)}]^4 \pm \frac{\left\| \Delta^{(t)} \right\|_F}{|\bar{v}_k^{(t)}|} \right)$$

$$\geq 8(1 - 2c)[\bar{v}_k^{(t)}]^2 a_k [\bar{v}_k^{(t)}]^2 = [\bar{v}_k^{(t)}]^2 \Omega \left( \frac{\beta \log^4 d}{d} \right).$$

This implies that within $t_1''$ time, we have $[\bar{v}_k^{(t)}]^2 \geq c$. Since $[\bar{v}_i^{(t)}]^2$ will remain $O(\log d/d)$ for $i \neq k$ and $t \leq t_1$, following the same argument above, it is easy to see $\frac{\mathrm{d}[\bar{v}_k^{(t)}]^2}{\mathrm{d}t} \geq 0$ after $[\bar{v}_k^{(t)}]^2$ reaches $c$. Therefore, $[\bar{v}_k^{(t)}]^2 \geq c$ for $t \leq t_1$. $\qquad \square$

### B.2.6 Proof of Lemma B.7

**Lemma B.7** (Good component, norm growth). *In the setting of Lemma 4, suppose $S_k^{(t)} = \varnothing$ for $t \leq t_1$, $a_k = \Omega(\beta)$. If there exists $\tau_0' \leq t_1$ such that $[\bar{v}_k^{(\tau_0')}]^2 > c$ and $[\bar{v}_i^{(\tau_0')}]^2 = O(\log(d)/d)$ for all $i \neq k$, then we have $\left\| v^{(t)} \right\|_2 \geq \delta_1$ for some $\tau_0' \leq t \leq \tau_0' + t_1'''$ with $t_1''' = \Theta(\log(d/\alpha)/\beta)$.*

*Proof.* For $\left\| v^{(t)} \right\|_2^2$, we have

$$\frac{\mathrm{d} \left\| v^{(t)} \right\|_2^2}{\mathrm{d}t} = \left\| v^{(t)} \right\|^2 \left( 4 \sum_{i \in [d]} \tilde{a}_i^{(t)} [\bar{v}_i^{(t)}]^4 \pm \left\| \Delta^{(t)} \right\|_F - 2\lambda \right).$$

Given the fact $\left\| \Delta^{(t)} \right\|_F = O(\alpha + m\delta_1^2)$ and $\lambda$ are small enough , it is easy to see $\left\| v^{(\tau_0')} \right\|_2 \geq \delta_0/2$ as $\tau_0' \leq t_1$. We now show that there exist time $\tau_1 \leq t_1' + t_1'' + t_1''' = t_1$ such that $\left\| v^{(\tau_1)} \right\|_2 \geq \delta_1$. By Lemma B.6 we know $[\bar{v}_k^{(t)}]^2 \geq c$ after time $\tau_0 + t_1' \leq t_1' + t_1''$. And since $S_k^{(t)} = \varnothing$, we know $\tilde{a}_k^{(t)} = a_k = \Omega(\beta)$. Then with the fact that $\left\| \Delta^{(t)} \right\|_F = O(\alpha + m\delta_1^2)$ and $\lambda$ are small enough, we have

$$\frac{\mathrm{d} \left\| v^{(t)} \right\|^2}{\mathrm{d}t} \geq \left\| v^{(t)} \right\|^2 \Omega(\beta).$$

This implies that $\left\| v^{(\tau_1)} \right\|_2^2 \geq \delta_1^2$ as $t_1''' = \Theta(\log(d/\alpha)/\beta)$. $\qquad \square$

### B.2.7 Proof of Lemma B.8

**Lemma B.8.** *In the setting of Lemma 4, we have*

*1. if $\left\| \bar{v}^{(t)} \right\|_\infty^2 \leq \log^4(d)/d$ for all $t \leq t_1$, then $\left\| v^{(t)} \right\|_2 = O(\delta_0)$ for all $t \leq t_1$.*

2. *Let $\tau_0 = \inf\{t \in [0, t_1] | \|\bar{v}^{(t)}\|_\infty^2 \geq \log^4 d/d\}$. Suppose $[\bar{v}_k^{(\tau_0)}]^2 \geq \log^4 d/d$ and $[\bar{v}_i^{(\tau_0)}]^2 = O(\log d/d)$ for $i \neq k$. If there exists $\tau_1$ such that $\tau_0 < \tau_1 \leq t_1$ and $\|v^{(\tau_1)}\|_2 \geq \delta_1$ for the first time, then there exists $k \in [d]$ such that $[\bar{v}_k^{(\tau_1)}]^2 \geq 1 - \alpha^2$ if $\hat{a}_k^{(t)} \leq \alpha$ for $t \leq \tau_1$ and $[\bar{v}_k^{(\tau_1)}]^2 \geq 1 - \alpha$ otherwise.*

*Proof.* For $\|v^{(t)}\|_2^2$, we have

$$\frac{\mathrm{d}\|v^{(t)}\|_2^2}{\mathrm{d}t} = \|v^{(t)}\|^2 \left(4\sum_{i \in [d]} \tilde{a}_i^{(t)}[\bar{v}_i^{(t)}]^4 \pm \|\Delta^{(t)}\|_F - 2\lambda\right)$$

**Part 1.** By Lemma B.12 and $\|\Delta^{(t)}\|_F = O(\alpha + m\delta_1^2)$, we know

$$\frac{\mathrm{d}\|v^{(t)}\|^2}{\mathrm{d}t} \leq \|v^{(t)}\|^2 \frac{5\beta c_e \log d}{d}.$$

This implies $\|v^{(t)}\|_2^2 = O(\delta_0)$ as $t_1 = O(\frac{d}{\beta \log d})$.

**Part 2.** By Part 1, we know $\|v^{(\tau_0)}\|_2 = O(\delta_0)$ and $[v_i^{(\tau_0)}]^2 = O(\delta_0^2 \log d/d)$ for $i \neq k$. For $[\bar{v}_i^{(\tau_0)}]^2 = O(\log d/d)$, we know $[\bar{v}_i^{(t)}]^2 = O(\log d/d)$ for $\tau_0 \leq t \leq \tau_1$ by Lemma B.5. We consider following cases separately.

1. Case 1: Suppose $\hat{a}_k^{(t)} \leq \alpha$ for $t \leq \tau_1$. In the following we show there exists some constant $C$ such that for all $i \neq k$ $[v_i^{(t)}]^2 \leq C\delta_0^2 \log d/d$ for $\tau_0 \leq t \leq \tau_1$. Let $\tau_2$ be the first time that the above claim is false, which means for all $i \neq k$ $[v_i^{(t)}]^2 \leq C\delta_0^2 \log d/d$ when $t \leq \tau_2$.

   For any $i \neq k$, we only need to consider the time period $t \leq \tau_2$ whenever $[v_i^{(t)}]^2 \geq \delta_0^2 \log d/d$. By Lemma B.14, we have

   $$\begin{aligned}
   \frac{\mathrm{d}}{\mathrm{d}t}[v_i^{(t)}]^2 =& 4[v_i^{(t)}]^2 \left(2\tilde{a}_i^{(t)}[\bar{v}_i^{(t)}]^2 - \sum_{i \in [d]} \tilde{a}_i^{(t)}[\bar{v}_i^{(t)}]^4 \pm O(\alpha + m\delta_1^2)\right.\\
   &\left.\pm O\left(\frac{(\alpha^2 + d\alpha^3 + d\alpha(1 - [\bar{v}_k^{(t)}]^2)^{1.5} + m\delta_1^2)\|v^{(t)}\|}{|v_i^{(t)}|}\right)\right)\\
   \leq& [v_i^{(t)}]^2 \left(O\left(\frac{\beta \log d}{d}\right) + O\left(\frac{(\alpha^2 + \alpha(1 - [\bar{v}_k^{(t)}]^2)^{1.5} + m\delta_1^2)\|v^{(t)}\|}{|v_i^{(t)}|}\right)\right).
   \end{aligned}$$

   Since for all $i \neq k$ $[v_i^{(t)}]^2 \leq C\delta_0^2 \log d/d$, we know $\sum_{i \neq k}[v_i^{(t)}]^2 = \|v^{(t)}\|^2 (1 - [\bar{v}_k^{(t)}]^2) = O(\delta_0^2 \log d)$. Together with the fact $[v_i^{(t)}]^2 \geq \delta_0^2 \log d/d$, we have

   $$\frac{\mathrm{d}}{\mathrm{d}t}[v_i^{(t)}]^2 \leq [v_i^{(t)}]^2 O\left(\frac{\beta \log d}{d}\right).$$

   Since $t_1 = O(d/(\beta \log d))$, we know if we choose large enough $C$, it must be $\tau_2 \geq \tau_1$. Therefore, we know for all $i \neq k$ $[v_i^{(t)}]^2 \leq C\delta_0^2 \log d/d$ for $\tau_0 \leq t \leq \tau_1$. Then at time $\tau_1$ when $\|v^{(\tau_1)}\|_2 \geq \delta_1$, it must be $[\bar{v}_k^{(t)}]^2 \geq 1 - \alpha^2$ since $\delta_1 = \Theta(\delta_0 \log^{1/2}(d)/\alpha)$.

2. Case 2: We do not make assumption on $\hat{a}_k^{(t)}$. In the following we show there exists some constant $C$ such that for all $i \neq k$ $[v_i^{(t)}]^2 \leq \delta_1^2 \alpha/d$ for $\tau_0 \leq t \leq \tau_1$. Let $\tau_2$ be the first time that the above claim is false, which means for all $i \neq k$ $[v_i^{(t)}]^2 \leq \delta_1^2 \alpha/d$ when $t \leq \tau_2$.

For any $i \neq k$, we only need to consider the time period $t \leq \tau_2$ whenever $[v_i^{(t)}]^2 \geq \delta_1^2 \alpha / 2d$. We have

$$
\frac{\mathrm{d}[v_i^{(t)}]^2}{\mathrm{d}t} = 4[v_i^{(t)}]^2 \left( 2\tilde{a}_i^{(t)}[\bar{v}_i^{(t)}]^2 - \sum_{i \in [d]} \tilde{a}_i^{(t)}[\bar{v}_i^{(t)}]^4 \pm \frac{\left\|\Delta^{(t)}\right\|_F \left\|v^{(t)}\right\|_2}{|v_i^{(t)}|} \right)
$$

$$
\leq [v_i^{(t)}]^2 \left( O\left(\frac{\beta \log d}{d}\right) + O\left(\frac{\alpha + m\delta_1^2}{\alpha^{1/2} d^{-1/2}}\right) \right).
$$

Since $m\delta_1^2 = O(\alpha)$ and $t_1 = O(d/(\beta \log d))$, we know it must be $\tau_2 \geq \tau_1$. Therefore, we know for all $i \neq k$ $[v_i^{(t)}]^2 \leq \delta_1^2 \alpha / d$ for $\tau_0 \leq t \leq \tau_1$. Then at time $\tau_1$ when $\left\|v^{(\tau_1)}\right\|_2 \geq \delta_1$, it must be $[\bar{v}_k^{(t)}]^2 \geq 1 - \alpha$.

$\square$

### B.2.8 Proof of Lemma B.9

To prove Lemma B.9, we need the following calculation on $\frac{d}{dt}\left\|v^{(t)}\right\|^2$.

**Lemma B.13.** *Suppose* $v^{(t)} \in S_k^{(t)}$, *we have*

$$
\frac{d}{dt}\left\|v^{(t)}\right\|^2 = \left( 4\tilde{a}_k^{(t)} - 2\lambda \pm O(\alpha + m\delta_1^2) \right) \left\|v^{(t)}\right\|^2.
$$

*Proof.* We can write down $\frac{d}{dt}\left\|v^{(t)}\right\|^2$ as follows:

$$
\frac{d}{dt}\left\|v^{(t)}\right\|^2 = \left( 4(T^* - T^{(t)})([\bar{v}^{(t)}]^{\otimes 4}) - 2\lambda \right) \left\|v^{(t)}\right\|^2
$$

$$
= \left( 4 \sum_{i \in [d]} \tilde{a}_i^{(t)}[\bar{v}_i^{(t)}]^4 \pm \left\|\Delta^{(t)}\right\|_F - 2\lambda \right) \left\|v^{(t)}\right\|^2
$$

Since $[\bar{v}_k^{(t)}]^2 \geq 1 - \alpha$, $[\bar{v}_i^{(t)}]^2 \leq \alpha$ for any $i \neq k$ and $\left\|\Delta^{(t)}\right\|_F = O(\alpha + m\delta_1^2)$, we have

$$
\frac{d}{dt}\left\|v^{(t)}\right\|^2 = \left( 4\tilde{a}_k^{(t)} - 2\lambda \pm O(\alpha + m\delta_1^2) \right) \left\|v^{(t)}\right\|^2.
$$

$\square$

Now we are ready to prove Lemma B.9.

**Lemma B.9.** *If* $S_k^{(0)} = \varnothing$ *and* $S_k^{(t')} \neq \varnothing$ *for some* $t' \in (0, t_1]$, *we have* $S_k^{(t_1)} \neq \varnothing$ *and* $\hat{a}_k^{(t_1)} \geq \delta_1^2$.

*Proof.* If $\tilde{a}_k^{(t)} = \Omega(\lambda)$ through Phase 1, according to Lemma B.13, we know $\left\|v^{(t)}\right\|^2$ will never decrease for any $v^{(t)} \in S_k^{(t)}$. So, we have $S_k^{(t_1)} \neq \varnothing$ and $\hat{a}_k^{(t_1)} \geq \delta_1^2$.

If $\tilde{a}_k^{(t)} = O(\lambda)$ at some time in Phase 1, according to Lemma A.18, it's not hard to show at the end of Phase 1 we still have $a_k - \hat{a}_k^{(t_1)} = O(\lambda)$. This then implies $\hat{a}_k^{(t_1)} = \Omega(\frac{\epsilon}{\sqrt{d}})$. Note that we only re-initialize the components that have norm less than $\delta_1$. As long as $\delta_1^2 = O(\frac{\epsilon}{m\sqrt{d}})$, we ensure that after the re-initialization, we still have $\hat{a}_k^{(t_1)} = \Omega(\frac{\epsilon}{\sqrt{d}})$, which of course means $S_k^{(t_1)} \neq \varnothing$. $\square$

### B.2.9 Technical Lemma

**Lemma B.14.** *In the setting of Lemma B.8, suppose $\hat{a}_k^{(t)} \le \alpha$. We have for $i \ne k$*

$$\frac{\mathrm{d}}{\mathrm{d}t}[v_i^{(t)}]^2 = 4[v_i^{(t)}]^2 \left( 2\tilde{a}_i^{(t)}[\bar{v}_i^{(t)}]^2 - \sum_{i \in [d]} \tilde{a}_i^{(t)}[\bar{v}_i^{(t)}]^4 \pm O(\alpha + m\delta_1^2) \right.$$
$$\left. \pm O\left( \frac{(\alpha^2 + \alpha(1 - [\bar{v}_k^{(t)}]^2)^{1.5} + m\delta_1^2)\left\|v^{(t)}\right\|}{|v_i^{(t)}|} \right) \right).$$

*Proof.* In order to prove this lemma, we need a more careful analysis on $\frac{d}{dt}[v_i^{(t)}]^2$. Recall we can decompose $T^{(t)}$ as $\sum_{i \in [d]} T_i^{(t)} + T_\varnothing^{(t)}$ and further write each $T_i^{(t)}$ as $\hat{a}_i^{(t)} e_i^{\otimes 4} + (T_i^{(t)} - \hat{a}_i^{(t)} e_i^{\otimes 4})$. Note that $\left\|(T_i^{(t)} - \hat{a}_i^{(t)} e_i^{\otimes 4})\right\|_F = O(\hat{a}_i^{(t)} \alpha)$ and $\left\|T_\varnothing^{(t)}\right\|_F \le m\delta_1^2$. We can write down $\frac{d}{dt}[v_i^{(t)}]^2$ in the following form:

$$\frac{\mathrm{d}}{\mathrm{d}t}[v_i^{(t)}]^2 = 4[v_i^{(t)}]^2 \left( 2a_i[\bar{v}_i^{(t)}]^2 - \sum_{i \in [d]} a_i[\bar{v}_i^{(t)}]^4 \right)$$
$$- 8v_i^{(t)}\left\|v^{(t)}\right\| \sum_{j \in [d]} \left[ T_j^{(t)}([\bar{v}^{(t)}]^{\otimes 3}, I) \right]_i - 8v_i^{(t)}\left\|v^{(t)}\right\| \left[ T_\varnothing^{(t)}([\bar{v}^{(t)}]^{\otimes 3}, I) \right]_i$$
$$+ 4v_i^{(t)} \sum_{j \in [d]} \left[ T_j^{(t)}([\bar{v}^{(t)}]^{\otimes 4})v^{(t)} \right]_i + 4v_i^{(t)} \left[ (T_\varnothing^{(t)}([\bar{v}^{(t)}]^{\otimes 4})v^{(t)} \right]_i$$
$$= 4[v_i^{(t)}]^2 \left( 2a_i[\bar{v}_i^{(t)}]^2 - \sum_{i \in [d]} a_i[\bar{v}_i^{(t)}]^4 \right)$$
$$- 8v_i^{(t)}\left\|v^{(t)}\right\| \sum_{j \in [d]} \left[ T_j^{(t)}([\bar{v}^{(t)}]^{\otimes 3}, I) \right]_i \pm v_i^{(t)}\left\|v^{(t)}\right\| O(m\delta_1^2)$$
$$+ 4[v_i^{(t)}]^2 \sum_{j \in [d]} T_j^{(t)}([\bar{v}^{(t)}]^{\otimes 4}) \pm [v_i^{(t)}]^2 O(m\delta_1^2)$$
$$= 4[v_i^{(t)}]^2 \left( 2a_i[\bar{v}_i^{(t)}]^2 - \sum_{i \in [d]} (a_i - \hat{a}_i)[\bar{v}_i^{(t)}]^4 \pm O(\alpha + m\delta_1^2) \right)$$
$$- 8v_i^{(t)}\left\|v^{(t)}\right\| \sum_{j \in [d]} \left[ T_j^{(t)}([\bar{v}^{(t)}]^{\otimes 3}, I) \right]_i \pm v_i^{(t)}\left\|v^{(t)}\right\| O(m\delta_1^2).$$

We now bound the term $\left[ T_j^{(t)}([\bar{v}^{(t)}]^{\otimes 3}, I) \right]_i$.

1. Case 1: $j = i$. If $\hat{a}_i^{(t)} = 0$, we know $T_i^{(t)} = 0$. Otherwise, denote $x = \left\langle \bar{w}_{-i}, \bar{v}_{-i}^{(t)} \right\rangle$, we have

$$\left[ T_i^{(t)}([\bar{v}^{(t)}]^{\otimes 3}, I) \right]_i$$
$$= \hat{a}_i^{(t)} \mathbb{E}_{i,w}^{(t)} \bar{w}_i \left\langle \bar{w}, \bar{v}^{(t)} \right\rangle^3$$
$$= \hat{a}_i^{(t)} \mathbb{E}_{i,w}^{(t)} \bar{w}_i \left( (\bar{w}_i \bar{v}_i^{(t)})^3 + (\bar{w}_i \bar{v}_i^{(t)})^2 x + (\bar{w}_i \bar{v}_i^{(t)})x^2 + x^3 \right)$$
$$\le \hat{a}_i^{(t)}[\bar{v}_i^{(t)}]^3 + \hat{a}_i^{(t)}|\bar{v}_i^{(t)}|\mathbb{E}_{i,w}^{(t)}|x| + \hat{a}_i^{(t)}|\bar{v}_i^{(t)}|\mathbb{E}_{i,w}^{(t)}x^2 + \hat{a}_i^{(t)}\mathbb{E}_{i,w}^{(t)}x^3.$$

Since $|x| \le \|\bar{w}_{-1}\|$ and $\mathbb{E}_{i,w}^{(t)}\|\bar{w}_{-i}\| \le (\mathbb{E}_{i,w}^{(t)}\|\bar{w}_{-i}\|^2)^{1/2} = O(\alpha)$, we have $\left[ T_i^{(t)}([\bar{v}^{(t)}]^{\otimes 3}, I) \right]_i = \hat{a}_i^{(t)}[\bar{v}_i^{(t)}]^3 + \hat{a}_i^{(t)}|\bar{v}_i^{(t)}|O(\alpha) + \hat{a}_i^{(t)}O(\alpha^{2.5}).$

2. Case 2: $j = k$. We have $\left[T_k^{(t)}([\bar{v}^{(t)}]^{\otimes 3}, I)\right]_i = \hat{a}_k^{(t)} \mathbb{E}_{k,w}^{(t)} \bar{w}_i \left\langle \bar{w}, \bar{v}^{(t)} \right\rangle^3 \le \hat{a}_k^{(t)} \mathbb{E}_{k,w}^{(t)} |\bar{w}_i| = O(\alpha^2)$, since $\hat{a}_k^{(t)} \le \alpha$ and $\mathbb{E}_{k,w}^{(t)} |\bar{w}_i| \le (\mathbb{E}_{k,w}^{(t)} |\bar{w}_i|^2)^{1/2} = O(\alpha)$.

3. Case 3: $j \ne i, k$. $j \ne i, k$. If $\hat{a}_j^{(t)} = 0$, we know $T_j^{(t)} = 0$. Otherwise, we can write $T_j^{(t)}$ as $\hat{a}_j^{(t)} \mathbb{E}_{j,w}^{(t)} \bar{w}^{\otimes 4}$. So we just need to bound $\mathbb{E}_{j,w}^{(t)} \bar{w}_i \left\langle \bar{w}, \bar{v}^{(t)} \right\rangle^3$. We know $\left| \left\langle \bar{w}, \bar{v}^{(t)} \right\rangle \right| = \left| \left\langle \bar{w}_{-j}, \bar{v}_{-j}^{(t)} \right\rangle + \bar{w}_j \bar{v}_j^{(t)} \right| \le \|\bar{w}_{-j}\| + \sqrt{1 - [\bar{v}_k^{(t)}]^2}$. So we have

$$\mathbb{E}_{j,w}^{(t)} \bar{w}_i \left\langle \bar{w}, \bar{v}^{(t)} \right\rangle^3 = \mathbb{E}_{j,w}^{(t)} \bar{w}_i O\left( \|\bar{w}_{-j}\|^3 + (1 - [\bar{v}_k^{(t)}]^2)^{1.5} \right)$$
$$\le O\left( \alpha^3 + \alpha(1 - [\bar{v}_k^{(t)}]^2)^{1.5} \right),$$

where in the lase line we use $\mathbb{E}_{j,w}^{(t)} \bar{w}_i \le (\mathbb{E}_{j,w}^{(t)} \bar{w}_i^2)^{1/2} = O(\alpha)$.

Recall that $\tilde{a}_i^{(t)} = a_i - \hat{a}_i^{(t)}$. We now have

$$\frac{\mathrm{d}}{\mathrm{d}t}[v_i^{(t)}]^2 = 4[v_i^{(t)}]^2 \left( 2\tilde{a}_i^{(t)}[\bar{v}_i^{(t)}]^2 - \sum_{i \in [d]} \tilde{a}_i^{(t)}[\bar{v}_i^{(t)}]^4 \pm O(\alpha + m\delta_1^2) \right.$$
$$\left. \pm O\left( \frac{(\alpha^2 + \alpha(1 - [\bar{v}_k^{(t)}]^2)^{1.5} + m\delta_1^2) \|v^{(t)}\|}{|v_i^{(t)}|} \right) \right).$$

$\square$

## C  Proofs for Phase 2

The goal of this section is to show that all discovered directions can be fitted within time $t_2^{(s)} - t_1^{(s)}$ and the reinitialized components will not move significantly. Namely, we prove the following lemma.

**Lemma 6** (Main Lemma for Phase 2). *In the setting of Theorem 1, suppose Proposition 1 holds at $(s, t_1^{(s)})$, we have for $t_2^{(s)} - t_1^{(s)} := O(\frac{\log(1/\delta_1) + \log(1/\lambda)}{\beta^{(s)}})$*

1. *Proposition 1 holds at $(s, t)$ for any $t_1^{(s)} \le t \le t_2^{(s)}$.*

2. *If $S_k^{(s,t_1^{(s)})} \ne \varnothing$, we have $a_k - \hat{a}_k^{(s,t_2^{(s)})} \le 2\lambda$.*

3. *For any component $v$ that was reinitialized at $t_1^{(s)}$, we have $\left\| v^{(s,t_2^{(s)})} \right\|^2 = \Theta(\delta_0^2)$ and $\left[ \bar{v}_i^{(s,t_2^{(s)})} \right]^2 = \left[ \bar{v}_i^{(s,t_1^{(s)})} \right]^2 \pm o\left( \frac{\log d}{d} \right)$ for every $i \in [d]$.*

Note that since $\delta_1^2 = \mathrm{poly}(\varepsilon)/\mathrm{poly}(d)$ and $\log(d/\varepsilon) = o(d/\log d)$, we have $t_2^{(s)} - t_1^{(s)} = \frac{o(d/\log d)}{\beta^{(s)}}$.

**Notations**  As in Sec. A, to simplify the notations, we shall drop the superscript of epoch $s$, and write $z^{(t)} := \left\langle \bar{v}^{(t)}, \bar{w}^{(t)} \right\rangle$ and $\tilde{a}_k^{(t)} := a_k - \hat{a}_k^{(t)}$. Within this section, we write $T := t_2^{(s)} - t_1^{(s)}$.

**Proof overview**  The first part is proved using the analysis in Appedix A. Note that we should view the analysis in this section and the analysis in Appendix A as a whole induction/continuity argument. It's easy to verify that at any time $t_1^{(s)} \le t \le t_2^{(s)}$, Assumption 1 holds and Proposition 1 holds.

The second part is a simple corollary of Lemma A.18 that gives a lower bound for the increasing speed of $\hat{a}_k^{(t)}$.

For the third part, we proceed as follows. At the beginning of phase 2, for any reinitialized component $v^{(t)}$, we know there exists some universal constant $C > 0$ s.t. $[\bar{v}_k^{(t)}]^2 \le C \log d/d$ for all $k \in [d]$. Let $T'$ be the minimum time needed for some $[\bar{v}_k^{(t)}]^2$ to reach $2C \log d/d$. For any $t \le T' + t_1^{(s)}$, we have $[\bar{v}_k^{(t)}]^2 \le 2C \log d/d$ and then we can derive an upper bound on the movement speed of $v^{(t)}$, with which we show the change of $[\bar{v}_k^{(t)}]^2$ is $o(\log d/d)$ within time $T$. (Also note this automatically implies that $T' > T$.) To bound the change of the norm, we proceed in a similar way but with $T'$ being the minimum time needed for some $\|v^{(t)}\|$ to reach $2\delta_0$. (Strictly speaking, the actual $T'$ is the smaller one between them.)

**Lemma C.1.** *If $S_k^{(s, t_1^{(s)})} \ne \varnothing$, then after at most $\frac{4}{a_k} \log\left(\frac{a_k}{2\delta_1^2}\right)$ time, we have $\tilde{a}_k^{(t)} \le \lambda$.*

*Proof.* Recall that Lemma A.18 says [8]

$$\frac{1}{\hat{a}_k^{(t)}} \frac{\mathrm{d}}{\mathrm{d}t} \hat{a}_k^{(t)} \ge 2\tilde{a}_k^{(t)} - \lambda - O\left(\alpha^2\right).$$

As a result, when $\tilde{a}_k^{(t)} < 2\lambda/3$, we have $\frac{\mathrm{d}}{\mathrm{d}t}\hat{a}_k^{(t)} \ge \tilde{a}_k^{(t)}\hat{a}_k^{(t)}$ or, equivalently, $\frac{\mathrm{d}}{\mathrm{d}t}\tilde{a}_k^{(t)} \le -\tilde{a}_k^{(t)}\hat{a}_k^{(t)}$. When $\hat{a}_k^{(t)} \le a_k/2$, we have $\frac{\mathrm{d}}{\mathrm{d}t}\hat{a}_k^{(t)} \ge a_k\hat{a}_k^{(t)}/2$, whence it takes at most $\frac{2}{a_k}\log\left(\frac{a_k}{2\delta_1^2}\right)$ time for $\hat{a}_k^{(t)}$ to grow from $\delta_1^2$ to $a_k/2$. When $\hat{a}_k^{(t)} \ge a_k/2$, we have $\frac{\mathrm{d}}{\mathrm{d}t}\tilde{a}_k^{(t)} \le -a_k\tilde{a}_k^{(t)}/2$, whence it takes at most $\frac{2}{a_k}\log\left(\frac{a_k}{2\lambda}\right)$. Hence, the total amount of time is upper bounded by $\frac{2}{a_k}\left(\log\left(\frac{a_k}{2\delta_1^2}\right) + \log\left(\frac{a_k}{2\lambda}\right)\right)$. Finally, use the fact $\lambda > \delta_1^2$ to complete the proof. $\square$

**Lemma C.2.** *For any $k \in [d]$ and $\bar{v}^{(t)}$ with $\|\bar{v}^{(t)}\|_\infty^2 \le O(\log d/d)$, we have $\mathbb{E}_{k,w}^{(t)}[z^{(t)}]^4 = [\bar{v}_k^{(t)}]^4 \pm O\left(\frac{\log d}{d}\alpha\right)$. Meanwhile, for each $\bar{w}^{(t)} \in S_k^{(t)}$, we have $|z^{(t)}| \le O\left(\sqrt{\frac{\log d}{d}}\right)$.*

*Proof.* For simplicity, put $x^{(t)} = \left\langle \bar{w}_{-k}^{(t)}, \bar{v}_{-k}^{(t)} \right\rangle$. Then we have

$$\mathbb{E}_{k,w}^{(t)}[z^{(t)}]^4 = \mathbb{E}_{k,w}^{(t)}\left\{ [\bar{w}_k^{(t)}]^4[\bar{v}_k^{(t)}]^4 + 4[\bar{w}_k^{(t)}]^3[\bar{v}_k^{(t)}]^3 x^{(t)} + 6[\bar{w}_k^{(t)}]^2[\bar{v}_k^{(t)}]^2[x^{(t)}]^2 \right.$$
$$\left. + 4\bar{w}_k^{(t)}\bar{v}_k^{(t)}[x^{(t)}]^3 + [x^{(t)}]^4 \right\}.$$

For the first term, we have $[\bar{v}_k^{(t)}]^4\mathbb{E}_{k,w}^{(t)}[\bar{w}_k^{(t)}]^4 = [\bar{v}_k^{(t)}]^4\left(1 \pm O(\alpha^2)\right)$. To bound the rest terms, we compute

$$\mathbb{E}_{k,w}^{(t)}\left\{[\bar{w}_k^{(t)}]^3[\bar{v}_k^{(t)}]^3 x^{(t)}\right\} \le O(1)\left(\frac{\log d}{d}\right)^{1.5} \mathbb{E}_{k,w}^{(t)}\sqrt{1 - [\bar{w}_k^{(t)}]^2} \le O(1)\left(\frac{\log d}{d}\right)^{1.5}\alpha,$$

$$\mathbb{E}_{k,w}^{(t)}\left\{[\bar{w}_k^{(t)}]^2[\bar{v}_k^{(t)}]^2[x^{(t)}]^2\right\} \le O(1)\frac{\log d}{d}\alpha^2$$

$$\mathbb{E}_{k,w}^{(t)}\left\{\bar{v}_k^{(t)}[x^{(t)}]^3\right\} \le O(1)\sqrt{\frac{\log d}{d}}\alpha^{2.5}$$

$$\mathbb{E}_{k,w}^{(t)}\left\{[x^{(t)}]^4\right\} \le O(1)\alpha^3.$$

Use the fact $\alpha \le \log d/d$ and we get

$$\mathbb{E}_{k,w}^{(t)}[z^{(t)}]^4 = [\bar{v}_k^{(t)}]^4\left(1 \pm O(\alpha^2)\right) \pm O(1)\frac{\log d}{d}\alpha = [\bar{v}_k^{(t)}]^4 \pm O\left(\frac{\log d}{d}\alpha\right).$$

For the individual bound, it suffices to note that

$$\left|z^{(t)}\right| \le \left|\bar{v}_k^{(t)}\right| + \sqrt{1 - [\bar{w}_k^{(t)}]^2} \le O\left(\sqrt{\frac{\log d}{d}}\right) + \sqrt{\alpha} = O\left(\sqrt{\frac{\log d}{d}}\right).$$

$\square$

---

[8] $\alpha^2 = o(\lambda)$.

**Lemma C.3** (Bound on the tangent movement). *In Phase 2, for any reinitialized component $v^{(t)}$ and $k \in [d]$, we have $[\bar{v}_k^{(t_2)}]^2 = [\bar{v}_k^{(t_1)}]^2 + o(\log d/d)$.*

*Proof.* Recall the definition of $G_1$, $G_2$ and $G_3$ from Lemma A.7. By Lemma C.2, we have

$$G_1 \leq 8\tilde{a}_k^{(t)} \left(1 - [\bar{v}_k^{(t)}]^2\right) [\bar{v}_k^{(t)}]^4 + O(1)a_k \frac{\log d}{d}\alpha + 8\hat{a}_k^{(t)} \mathbb{E}_{k,w}^{(t)} \left\{[z^{(t)}]^3 \langle \bar{w}_{-k}, \bar{v}_{-k} \rangle\right\}$$

$$\leq 8\tilde{a}_k^{(t)} \left(1 - [\bar{v}_k^{(t)}]^2\right) [\bar{v}_k^{(t)}]^4 + O\left(a_k \frac{\log d}{d}\alpha\right),$$

where the second line comes from

$$\mathbb{E}_{k,w}^{(t)} \left\{[z^{(t)}]^3 \langle \bar{w}_{-k}, \bar{v}_{-k} \rangle\right\} \leq O(1) \frac{\log d}{d} \mathbb{E}_{k,w}^{(t)} \sqrt{1 - [\bar{w}_k^{(t)}]^2} \leq O\left(\frac{\log d}{d}\alpha\right).$$

Similarly, we have $|G_2| \leq O(1) \sum_{i \neq k} a_i \frac{\log d}{d}\alpha$. For $G_3$, by Lemma C.2, we have

$$a_i[\bar{v}_i^{(t)}]^4 - \hat{a}_i^{(t)} \mathbb{E}_{i,w}^{(t)} \left\{[z^{(t)}]^4\right\} = \tilde{a}_i^{(t)} [\bar{v}_i^{(t)}]^4 \pm O\left(a_i \frac{\log d}{d}\alpha\right).$$

Therefore

$$|G_3| \leq 8[\bar{v}_k^{(t)}]^2 \sum_{i \neq k} \left(\tilde{a}_i^{(t)} [\bar{v}_i^{(t)}]^4 \pm O\left(a_i \frac{\log d}{d}\alpha\right)\right)$$

$$\leq 8[\bar{v}_k^{(t)}]^2 \left(\left(\max_{i \neq k} \tilde{a}_i^{(t)}\right) O\left(\frac{\log d}{d}\right) + O\left(\frac{\log d}{d}\alpha\right)\right)$$

$$\leq O\left(\beta^{(s)} \frac{\log^2 d}{d^2}\right).$$

Thus[9],

$$\frac{\mathrm{d}}{\mathrm{d}t}[\bar{v}_k^{(t)}]^2 \leq 8\tilde{a}_k^{(t)} [\bar{v}_k^{(t)}]^4 + O\left(\frac{\log d}{d}\alpha\right) + O\left(\beta^{(s)} \frac{\log^2 d}{d^2}\right)$$

$$\leq O\left(\beta^{(s)} \frac{\log^2 d}{d^2}\right).$$

Integrate both sides and recall that $T = \frac{o(d/\log d)}{\beta^{(s)}}$. Thus, the change of $[\bar{v}_k^{(t)}]^2$ is $o(\log d/d)$. $\square$

**Lemma C.4** (Bound on the norm growth). *In Phase 2, for any reinitialized component $v^{(t)}$ and $k \in [d]$, we have $\left|\left\|v^{(t_2)}\right\|^2 - \left\|v^{(t_2)}\right\|^2\right| = o(\delta_0^2)$.*

*Proof.* By Lemma A.6 and Lemma C.2, we have

$$\frac{1}{2\left\|v^{(t)}\right\|^2} \frac{\mathrm{d}}{\mathrm{d}t} \left\|v^{(t)}\right\|^2 \leq \sum_{i=1}^d \left(a_i[\bar{v}_i^{(t)}]^4 - \hat{a}_i^{(t)} \mathbb{E}_{i,w}^{(t)}[z^{(t)}]^4\right)$$

$$\leq \sum_{i=1}^d \left(\tilde{a}_i^{(t)} [\bar{v}_i^{(t)}]^4 + a_i O\left(\frac{\log d}{d}\alpha\right)\right)$$

$$\leq \left(\max_{i \in [d]} \tilde{a}_i^{(t)}\right) O\left(\frac{\log d}{d}\right) + O\left(\frac{\log d}{d}\alpha\right)$$

$$= \left(\max_{i \in [d]} \tilde{a}_i^{(t)}\right) O\left(\frac{\log d}{d}\right).$$

Recall that $\max_{i \in [d]} \tilde{a}_i^{(t)} \leq O(\beta^{(s)})$ and $\|v^{(t)}\| \leq O(\delta_0)$. Hence,

$$\frac{\mathrm{d}}{\mathrm{d}t} \left\|v^{(t)}\right\|^2 \leq O\left(\beta^{(s)} \frac{\log d}{d}\right) \delta_0^2.$$

Integrate both sides, use the fact $T = \frac{o(d/\log d)}{\beta^{(s)}}$, and then we complete the proof. $\square$

---

[9] $\alpha \leq O(\beta^{(s)} \log d/d)$

**Proof of Lemma 6.** Lemma 6 follows by combining the above lemmas with the analysis in Appendix A. □

## D  Proof for Theorem 1

In the section, we give a proof of Theorem 1.

**Theorem 1.** *For any $\epsilon \geq \exp(-o(d/\log d))$, there exists $\gamma = \Theta(1)$, $m = poly(d)$, $\lambda = \min\{O(\log d/d), O(\epsilon/d^{1/2})\}$), $\alpha = \min\{O(\lambda/d^{3/2}), O(\lambda^2), O(\epsilon^2/d^4)\}$, $\delta_1 = O(\alpha^{3/2}/m^{1/2})$, $\delta_0 = \Theta(\delta_1\alpha/\log^{1/2}(d))$ such that with probability $1 - 1/poly(d)$ in the (re)-initializations, Algorithm 2 terminates in $O(\log(d/\epsilon))$ epochs and returns a tensor $T$ such that*

$$\|T - T^*\|_F \leq \epsilon.$$

Note that Proposition 1 guarantees any ground truth component with $a_i \geq \beta^{(s)}/(1 - \gamma)$ must have been fitted before epoch $s$ starts. When $\beta^{(s)}$ decreases below $O(\epsilon/\sqrt{d})$, all the ground truth components larger than $O(\epsilon/\sqrt{d})$ have been fitted and the residual $\|T - T^*\|_F$ must be less than $\epsilon$. Since $\beta^{(s)}$ decreases in a constant rate, the algorithm must terminate in $O(\log(d/\epsilon))$ epochs.

*Proof.* According to Lemma 4 and Lemma 6, we know Proposition 1 holds through the algorithm. We first show that $\beta^{(s)}$ is always lower bounded by $\Omega(\epsilon/\sqrt{d})$ before the algorithm ends. For the sake of contradiction, assume $\beta^{(s)} \leq O(\frac{\epsilon}{\sqrt{d}})$. We show that $\left\|T^{(s,0)} - T^*\right\|_F < \epsilon$, which is a contradiction because our algorithm should have terminated before this epoch. For simplicity, we drop the superscript on epoch $s$ in the proof.

We can upper bound $\left\|T^* - T^{(t)}\right\|_F$ by splitting $T^*$ into $\sum_{i\in[d]} T_i^*$ and splitting $T^{(t)}$ into $\sum_{i\in[d]} T_i^{(t)} + T_\varnothing^{(t)}$. Then, we have

$$\left\|T^* - T^{(t)}\right\|_F \leq \left\|\sum_{i\in d}(a_i - \hat{a}_i^{(t)})e_i^{\otimes 4}\right\|_F + \sum_{i\in[d]} \left\|T_i^{(t)} - \hat{a}_i^{(t)}e_i^{\otimes 4}\right\|_F + \left\|T_\varnothing^{(t)}\right\|_F$$

$$\leq O\left(\sqrt{d}\max\left(\beta^{(s)}, \lambda\right)\right) + O(\alpha + m\delta_1^2),$$

where the second inequality holds because $(a_i - \hat{a}_i^{(t)}) \leq O(\max\left(\beta^{(s)}, \lambda\right))$, $\left\|T_i^{(t)} - \hat{a}_i^{(t)}e_i^{\otimes 4}\right\|_F \leq O(\hat{a}_i^{(t)}\alpha)$ and $\left\|T_\varnothing^{(t)}\right\|_F \leq m\delta_1^2$. Choosing $\lambda, \alpha = O(\frac{\epsilon}{\sqrt{d}})$ and $\delta_1^2 = O(\frac{\epsilon}{m\sqrt{d}})$, we have

$$\left\|T^* - T^{(t)}\right\|_F < \epsilon.$$

Since $\beta^{(s)}$ starts from $O(1)$ and decreases by a constant factor at each epoch, it will decrease below $O(\frac{\epsilon}{\sqrt{d}})$ after $O(\log(d/\epsilon))$ epochs. This means our algorithm terminates in $O(\log(d/\epsilon))$ epochs. □

## E  Experiments

In Section E.1, we give detailed settings for our experiments in Figure 1. Then, we give additional experiments on non-orthogonal tensors in Section E.2.

### E.1  Experiment settings for orthogonal tensor decomposition

We chose the ground truth tensor $T^*$ as $\sum_{i\in[5]} a_i e_i^{\otimes 4}$ with $e_i \in \mathbb{R}^{10}$ and $a_i/a_{i+1} = 1.2$. We normalized $T^*$ so its Frobenius norm equals 1.

Our model $T$ was over-parameterized to have 50 components. Each component $W[:, i]$ was randomly initialized from $\delta_0\text{Unif}(\mathbb{S}^{d-1})$ with $\delta_0 = 10^{-15}$.

The objective function is $\frac{1}{2}\|T - T^*\|_F^2$. We ran gradient descent with step size 0.1 for 2000 steps. We repeated the experiment from 5 different experiments and plotted the results in Figure 1. Our experiments was ran on a normal laptop and took a few minutes.

### E.2 Additional results on non-orthogonal tensor decomposition

In this subsection, we give some empirical observations that suggests non-orthogonal tensor decomposition may not follow the greedy low-rank learning procedure in Li et al. (2020b).

**Ground truth tensor** $T^*$: The ground truth tensor is a $10 \times 10 \times 10 \times 10$ tensor with rank 5. It's a symmetric and non-orthogonal tensor with $\|T^*\|_F = 1$. The specific ground truth tensor we used is in the code.

**Greedy low-rank learning (GLRL):** We first generate the trajectory of the greedy low-rank learning. In our setting, GLRL consists of 5 epochs. At initialization, the model has no component. At each epoch, the algorithm first adds a small component (with norm $10^{-60}$) that maximizes the correlation with the current residual to the model, then runs gradient descent until convergence.

To find the component that has best correlation with residual $R$, we ran gradient descent on $R(w^{\otimes 4})$ and normalize $w$ after each iteration. In other words, we ran projected gradient descent to solve $\min_{w|\|w\|=1} R(w^{\otimes 4})$. We repeated this process from 50 different initializations and chose the best component among them.

In the experiment, we chose the step size as $0.3$. And at the $s$-th epoch, we ran $s \times 2000$ iterations to find the best rank-one approximation and also ran $s \times 2000$ iterations on our model after we included the new component. After each epoch, we saved the current tensor as a saddle point. We also included the zero tensor as a saddle point so there are 6 saddles in total.

Figure 2 shows that the loss decreases sharply in each epoch and eventually converges to zero.

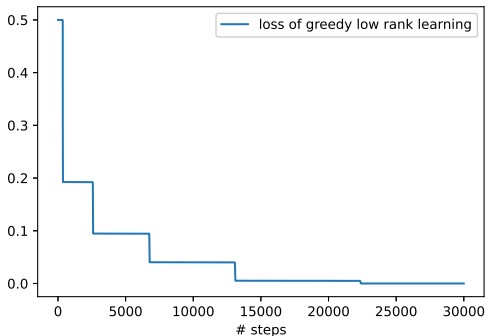

Figure 2: Loss trajectory of greedy low-rank learning.

**Over-parameterized gradient descent:** If the over-parameterized gradient descent follows the greedy low-rank learning procedure, one should expect that the model passes the same saddles when the tensor rank increases. To verify this, we ran experiments with gradient descent and computed the distance to the closest GLRL saddles at each iteration.

Our model has 50 components and each component is initialized from $\delta_0 \text{Unif}(\mathbb{S}^{d-1})$ with $\delta_0 = 10^{-60}$. We ran gradient descent with step size $0.3$ for 1000 iterations.

Figure 3 (left) shows that after fitting the first direction, over-parameterized gradient descent then has a very different trajectory from GLRL. After roughly $450$ iterations, the loss continues decreasing but the distance to the closest saddle is high. After $800$ iterations, gradient descent converges and the distance to the closest saddle (which is $T^*$) becomes low.

In Figure 3 (right), we plotted the norm trajoeries for 10 of the components. The figure shows that some of the already large components become even larger at roughly $450$ iterations, which corresponds to the second drop of the loss. We picked two of these components and found that their correlation $\langle \bar{w}, \bar{v} \rangle$ drops from 1 at the 400-th iteration to 0.48 at the 550-th iteration. This suggests that two large component in the same direction can actually split into two directions in the training.

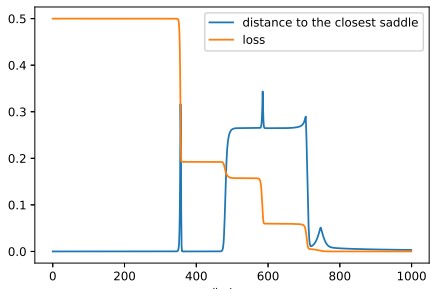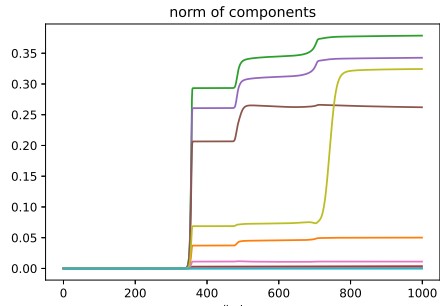

Figure 3: Non-orthogonal tensor decomposition with number of components $m = 50$ and initialization scale $\delta_0 = 10^{-60}$. The left figure shows the loss trajectory and the distance to the closest GLRL saddles; the right figures shows the norm trajectory of different components.

One might suspect that this phenomenon would disappear if we use more aggressive over-parameterization and even smaller initialization. We then let our model have $1000$ components and let the initialization size to be $10^{-100}$ and re-did the experiments. We observed almost the same behavior as before. Figure 4 (left) shows the same pattern for the distance to closest GLRL saddles as in Figure 3. In Figure 4 (right), we randomly chose 10 of the 1000 components and plotted their norm change, and we again observe that one large component becomes even larger at roughly iteration 700 that corresponds to the second drop of the loss function.

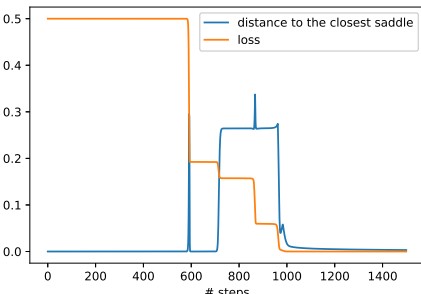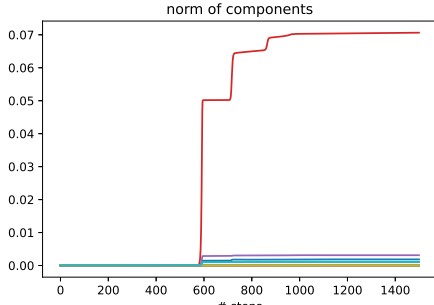

Figure 4: Non-orthogonal tensor decomposition with number of components $m = 1000$ and initialization scale $\delta_0 = 10^{-100}$. The left figure shows the loss trajectory and the distance to the closest GLRL saddles; the right figures shows the norm trajectory of different components.