# OpenReview forum: "Understanding Deflation Process in Over-parametrized Tensor Decomposition"
_NeurIPS.cc/2021/Conference — NeurIPS 2021 Poster_

### Official Review · Reviewer_ubkb · 2021-06-27

**Rating:** 7
**Confidence:** 4

**Summary:**

The paper analyzes the dynamics of (a slightly modified) gradient flow for overparameterized symmetric CP tensor decomposition of order 4, where the minimized objective is the L2 distance from an orthogonally decomposable target tensor. The main technical result (Theorem 1) establishes that the modified gradient flow follows a trajectory similar to that of a tensor deflation process, in which components are learned one after the other, recovering the target tensor. Empirical evaluations corroborate the theoretical analysis.

**Limitations And Societal Impact:**

N/A --- This theoretical work does not have apparent negative societal impacts.

**Main Review:**

I find the paper to be well written and technically sound (though I did not verify proofs in the appendix). In particular, the proof sketch for the main theorem is laid out in a manner that allows following the big picture, as well as specific details. Although the analyzed setting is rather restrictive --- symmetric order 4 decomposition, fully observed orthogonally decomposable target tensor, and a modified version of gradient flow --- the paper rigorously establishes greedy low rank learning under these conditions, similar to that described in Li et al. 2021 for matrices and Razin et al. 2021 for tensors. Furthermore, proofs of convergence for such non-convex problems are often highly non-trivial, even under simplifying assumptions. Overall, the contributions made improve upon current understanding of gradient-based dynamics for tensor factorizations.

There are a few drawbacks I see in this work besides the restricted setting (discussed below), yet I believe it is a worthy contribution and tend to recommend acceptance (given that these concerns are adequately addressed).

Drawbacks:
1. In Theorem 1, it seems that the approximation error cannot be arbitrarily small, but rather is bounded away from zero by a term dependent on the dimension. That is, for a certain fixed dimension it does not guarantee recovery of the ground truth to an arbitrary precision. Is this indeed the case? If so, I believe it is worth explicitly mentioning and clarifying. For example, in the introduction it is stated that recovery is to a desired accuracy.

2. In my opinion the works below are relevant, yet are not discussed in the related work section:
    - In addition to Li et al. 2021 which is referred to, Gidel et al. 2019, Gissin et al. 2020 , and Chou et al. 2020 also show a form of greedy low rank search for matrices, under certain conditions.
    - Razin & Cohen 2020 empirically demonstrates an implicit regularization towards low tensor rank for CP decompositions.


Additional (more minor) comments:
- In Equation (1), I believe the notation used for R over three inputs was not previously defined. Though one can extrapolate from the definition for two inputs, it might be worth explicitly defining.
- In Lemma 3 and Definition 2 the components are denoted by v instead of w as done up until that point.


Questions to authors:
1. Can the results be extended from order 4 tensors to arbitrary orders? If so, what is the dependence on the order, e.g. in the number of components necessary and the convergence rate?

2. In Theorem 1, what exactly is the condition on the error epsilon? It seems that it cannot be arbitrarily small, but rather bounded away from zero by a term dependent on the dimension.



References:

Li, Z., Luo, Y., and Lyu, K. Towards resolving the implicit bias of gradient descent for matrix factorization: Greedy low-rank learning. International Conference on Learning Representations, 2021.

Razin, N., Maman, A., and Cohen N. Implicit regularization in tensor factorization. International Conference on Machine Learning, 2021.

Gidel, G., Bach, F., and Lacoste-Julien, S. Implicit regularization of discrete gradient dynamics in linear neural networks. In Advances in Neural Information Processing Systems, 2019.

Gissin, D., Shalev-Shwartz, S., and Daniely, A. The implicit bias of depth: How incremental learning drives generalization. International Conference on Learning Representations, 2020.

Chou, H.-H., Gieshoff, C., Maly, J., and Rauhut, H. Gradient descent for deep matrix factorization: Dynamics and implicit bias towards low rank. arXiv preprint arXiv:2011.13772, 2020.

Razin, N. and Cohen, N. Implicit regularization in deep learning may not be explainable by norms. In Advances in Neural Information Processing Systems, 2020.



**Time Spent Reviewing:**

6

---

> ### Author Response · Authors · 2021-08-10
> **Thank you so much for your positive review and valuable suggestions!**
>
> Thank you so much for your positive review and valuable suggestions! We are glad to answer your questions as below.
>
> *“Approximation error cannot be arbitrarily small:”*
>
> We admit that the smallest approximation error has a dependency on dimension d and cannot be made arbitrarily small due to some technical challenges. But we want to emphasize that this dependency is very mild and the error can be exponentially small ($\exp(-\Omega(d/\log^2 d)$) to be precise). In the real applications, the dimension is usually large, so this error lower bound can easily drop below the numerical precision.
>
> *“Generalize to higher order tensors? What’s the dependency on the tensor order?”*
>
> Our result can be easily generalized to higher order tensors. We believe the number of components required is independent with the tensor order but the running time is exponential in the tensor order. We will add more discussion on higher order tensors in the later version.
>
> We will also include the related works mentioned and fix other minor issues in the later version. Thanks for the suggestions!

---

> > ### Comment · Reviewer_ubkb · 2021-08-10
> > **Small Follow Up Question**
> >
> > Thank you for the response. I've read it and other reviewer's comments carefully. I agree that the limitation on the approximation error $\epsilon$ is a rather minor issue. In my opinion, it might still be worth clarifying in the paper to avoid confusion (even if just in a footnote). Specifically, I find the possible sources for confusion to be:
> >
> > (i) The phrase "to desired accuracy" in line 36.
> >
> > (ii) Posing the condition on $\epsilon$ in Theorem 1 as $\log (1 / \epsilon) = o (d / \log d)$ instead of the more direct $\epsilon = \exp ( - \Omega ( d / \log^2 d ))$.
> >
> > Though, this may be a bit nitpicking. Out of interest, I would appreciate it if you can briefly elaborate here on the source of this limitation. That is, why can't $\epsilon$ be arbitrarily small?
> >
> > Regardless, I would like to keep my initial positive assessment.

---

> > > ### Author Response · Authors · 2021-08-10
> > > **Source of the limitation of accuracy**
> > >
> > > Thanks for your suggestions! We will fix the phrasing on the accuracy and add more clarifications.
> > >
> > > Regarding the source of the accuracy limitation, one major issue is that our argument relies on the fact that Phase 2 is short, which ensures the movement of the re-initialized components is negligible (line 305 in Lemma 6). However, if $\epsilon$ is exponentially small, the norm of the discovered component at the beginning of Phase 2 is exponentially small and it takes a long time for these components to grow up to fit the corresponding ground truth components.

---

> > > > ### Comment · Reviewer_ubkb · 2021-08-10
> > > > **Concerns Fully Addressed**
> > > >
> > > > I understand. Thank you for fully addressing my concerns and questions.

---

### Official Review · Reviewer_tiKB · 2021-07-09

**Rating:** 5
**Confidence:** 3

**Summary:**

This work studied a special training behavior of over-parametrized tensor decomposition by gradient-based methods. The paper pointed out this phenomena as the tensor deflation process and rigorously proved its convergence (Thm. 1) and the behaviors of factors (Prop. 1 and etc.) during the training. The implicit regularization by gradient method is an interesting and important issue in deep learning, and this work shows it would also happen in tensor decomposition. It may help the tensor researchers to develop more effective methods to seek for low-tensor-rank approximations, especially when the tensor model owns complex structures like tensor networks. However, unlike the closed work (li et al., 2020b) on matrices, several additional conditions and assumptions are imposed in the paper (norm regularization and re-initialization shown in Algo. 2), and the theoretical results seem to be only held for the decomposition task (the Fro. objective) and to be difficult for people to extend it into more general forms like tensor learning. It therefore limits the impact on the both tensor and machine learning community.

**Limitations And Societal Impact:**

Yes.

**Main Review:**

Pros:

- The paper clearly discussed the tensor deflation process, which is interesting and somehow important.
- The proof thoughts and tricks like the *continuity argument* is instructive for me though I'm not mathematician.

Cons:

- It's difficult for me to evaluate if those mentioned modification is *mild* or not. If they are necessary for the both algorithm and theory. It will severely limit its application and extension on more sophisticated models in *machine learning*.
- Furthermore, there is also no discussion on if such implicit regularization on tensor decomposition outperforms the existing low-rank approximation methods or not, even though the  phenomena itself is indeed interesting.

### Comments

After reading the supp, I was convinced that the proof is a tough work and many tricks are impressive for me (engineering background). However, as a tensor researcher, I was always thinking about how *much new information or insight I can obtain from this impressive work?* For example, as the main contribution of the work, Thm. 3 only proves the convergence *if a symmetric orthogonal order-4 tensor is decomposed by Algo. 2,* which is closed to yet different from a trivial gradient method*.* It makes me consider why I need such a theorem, of which the form is very specific and the insight has been discussed by (li et al., 2020b).  As for Prop. 1 and other results, I partially understand they reflect the training behaviour of the factors, but I'm not sure if those results can fully support the claims given in the paper. For example, in line 238-239, the authors claim that the condition (c) shows a ground-truth component with large enough a_k can be always fitted. I think this claim is based on the upper-bound given at the end of line 233. Although in supp. there is discussion on lambda, I'm still confused if the bound is trivial (tight enough) or not and if or not it really can be used to support that those components can be always fitted. Therefore, it would be better to give more interpretation on each claim and more discussion to guide ML/AI researchers to exploit your results.

### Questions

1. What happens for your results if the tensor order equals two? In this case, what is the main difference from the work in (li et al., 2020b)? I have this question because the orth. assumption on factors makes the property of tensor decomposition be closed to its matrix counterpart. I therefore curious if your results can bring something new, which is significantly different from matrix.

**Time Spent Reviewing:**

5-10

---

> ### Author Response · Authors · 2021-08-10
> **Thanks a lot for your detailed review and valuable suggestions!**
>
> Thanks a lot for your detailed review and valuable suggestions! We are glad to take this opportunity to address your concerns as below.
>
> *“Modification to vanilla gradient flow and restriction to decomposition tasks?”*
>
> Our modifications to the algorithm are mostly motivated by the theoretical challenges in the proof. These modifications are mild and do not change the empirical behavior of the algorithm, in particular the tensor deflation phenomenon. Modification as norm regularization and re-initialization are also very common in the non-convex optimization literature.
>
> Regarding the restriction to fully-observable setting (tensor decomposition), we are aware that previous work (Razin et al 2021) studied the partially-observable setting (tensor learning), but they only characterized the learning of the top direction (corresponds to the first epoch in our paper). Instead, we focused on a simpler setting with fully-observable ground truth tensor and
> gave a complete analysis of fitting all the ground truth components. The proof in this simple setting already requires highly non-trivial analysis and yields the interesting tensor deflation phenomenon. We hope our techniques will be useful in the future study on tensor learning.
>
> *“Implicit regularization on tensor decomposition outperforms the existing low-rank approximation methods?”*
>
> The goal of our paper is not to propose a new algorithm that outperforms current ones in tensor decomposition, but rather to understand the training trajectory of gradient flow. We view our result as a first step towards understanding the implicit regularization of over-parameterized models for low-rank tensors.
>
> *“The insights in Theorem 1 have been discussed in Li et al 2020b:”*
>
> We agree that the phenomenon of greedy low-rank learning has been discussed by Li et al, but they focused on matrix sensing and only fully analyzed the fitting of the top direction. Tensor decomposition is a very different problem compared to matrix factorization (even when we assume tensor components are orthogonal, see answer to last question). We considered the tensor decomposition problem and fully characterized the fitting of all the ground truth components.
>
> *“Line 233, how large is lambda?”*
>
> In the main theorem, we choose the lambda as small as eps/\sqrt{d}, so the claim in line 233 proved that for any large enough component, the residual is bounded by eps/sqrt(d). We will add more clarification on this.
>
> *“What happens when the tensor order equals two? What’s the main difference from work in li et al 2020b?”*
>
> When the tensor order equals two, the problem reduces to matrix decomposition, which has been studied by Li et al. They proved that the model fits the singular values of the ground truth matrix one by one in the decreasing order. Our analysis actually does not work for matrix decomposition. In matrix decomposition, if all the singular values are equal, due to the rotation invariance, we cannot expect each component in our model to align with one ground truth component. In the 4-th order tensor decomposition problem, we proved every large component in our model aligns with one ground truth component. The analysis for high order tensors is also much harder than that for matrix decomposition problems.
>
> We hope our response answers your questions. Again, thanks for your valuable suggestions, we will revise our paper accordingly in the later version.

---

> > ### Comment · Reviewer_tiKB · 2021-08-13
> > **Concern addressed**
> >
> > Thanks for addressing my concerns. I'm convinced and agree with the contribution of the work after reading the response and other reviewers' comments.

---

### Official Review · Reviewer_2dDS · 2021-07-10

**Rating:** 7
**Confidence:** 4

**Summary:**

The paper analyzes gradient descent on the orthogonal tensor decomposition problem. More specifically, they consider a ground 4th order tensor with r orthonormal factors and an overparametrized estimate with m>r factors, and analyze the performance of (a variant of) gradient descent on the problem.

The paper establishes a close relationship between the dynamics of gradient descent and deflation based tensor decomposition. In deflation based tensor decomposition, the factors are recovered one at a time, and the factors recovered so far are subtracted from the original tensor to recover the next factor. The paper shows that the dynamics of a modified gradient descent algorithm are similar: the algorithm also recovers factors one at a time.

The analysis uses the following key ideas, some of which are new and some are variants of previous techniques. First, the paper states a straightforward relation between gradient descent and the well-known tensor power method. This relation ensures that a non-negligible correlation with one of the original factors will get amplified on performing the gradient descent updates (similar to the tensor or matrix power methods which are well-analyzed). Next, the paper shows a local stability condition to ensure that once an iterate converges to one of the original factors, then it does not move away. This is done by adding a regularizer. Finally, the paper needs to do a periodic re-randomization of the iterates which have not converged so far to remove dependencies. To identify which factors have not converged so far, the paper establishes a relation between the norm of a factor and its correlation with any of the true factors. It appears that this is the most novel and important part of the proof. While periodic re-randomization has been introduced in the context of tensor power method and alternating least squares (ALS) and their variants, establishing a link between norm and correlation appears novel in this context.

The paper also does an empirical validation of the claims, but this is more to support the theory than a contribution in itself.

**Limitations And Societal Impact:**

Other than the suggestions above, the authors adequately discuss the other limitations of the work.

**Main Review:**

The paper appears to be mainly looking at the orthogonal tensor decomposition problem from the perspective of understanding neural networks and overparameterization. In this context, the main takeaway is that gradient descent recovers the true factors in the orthogonal case despite overparametrization. This is an interesting and novel contribution, though it would be more striking if there were more deeper takeways or implications. For instance, is it the case that that overparametrization is in fact necessary for recovering all the true factors? This is possible because one might imagine that multiple recovered factors could converge to the same original factors, and therefore if m=r then all the original factors will no longer be recovered. If true, such a result could complete the picture and highlight the role of overparametrization more.

While the paper is mainly focussing on the results in the context of deep learning theory, I think it would be useful to also put it in context of the known theory for tensor decomposition. Stronger guarantees than in the present paper (for example, extending to the non-orthogonal case) are known for the tensor power method, ALS, orthogonalized-ALS, simultaneous
diagonalization and higher order SVD etc., and it would be good to discuss at least some of the more relevant ones.

The paper is quite well-written. The algorithms and theoretical results are clearly described with ample intuition. The experiments also do a good job of explaining what is happening, particularly Fig. 1.

Overall this appears to be a good paper, though I wouldn't argue strongly about it at this point.

----after author response-----

Thank you for your response, it would be nice to add some discussion for why overparametrization is necessary and some discussion of related work on tensor decomposition. I'll keep my score (7) and advocate for acceptance.

**Time Spent Reviewing:**

4

---

> ### Author Response · Authors · 2021-08-10
> **Thank you so much for your positive review and valuable suggestions!**
>
> Thank you so much for your positive review and valuable suggestions! We are glad to answer your questions as below.
>
> *“Overparameterization is in fact necessary for recovering all the true factors?”*
>
> Overparameterization is necessary in theory for us to prove the convergence result. In practice, we also observed that if our model just has the same number of components as the ground truth, gradient flow would usually get stuck at a suboptimal solution (note that a similar observation for 2-layer neural network was made in “Spurious local minima are common in two-layer ReLu neural networks” by Safran and Shamir). Basically, all the components converge to several of the largest ground truth components and there is no remaining component to recover the smaller directions.
>
> *“Put in context of the known theory for tensor decomposition, stronger guarantees are known by other algorithms: ”*
>
> The goal of this paper is to understand the training trajectory of gradient flow on tensor decomposition, but not to design a better algorithm. That being said, we do agree that we should add more discussion on the theoretical guarantees of other non-gradient based algorithms. Thanks for the suggestions, we will cite these results in the later version.

---

### Official Review · Reviewer_b9Va · 2021-07-14

**Rating:** 7
**Confidence:** 3

**Summary:**

A connection is established between gradient descent in an over-parameterized symmetric CP decomposition and a corresponding tensor deflation process. In particular, a gradient-based learning algorithm is introduced which admits a probabilistic convergence guarantee, as well as an analysis characterizing the overlap between the learned tensor and various low-rank components of the original (orthogonally decomposable) tensor.

**Limitations And Societal Impact:**

While the limitations of the current work are clearly mentioned, no potential negative societal impacts are given. This is understandable, given the technical nature of the work.

**Main Review:**

The current paper is one of many works giving rigorous bounds on the correctness of gradient-based learning in different linear-algebraic settings, and does a good job of clearly positioning these new results within this growing area of research. The main novelty here is a characterization of the trajectory of the learned tensor, which is shown to exhibits a strong similarity with tensor deflation methods (such as the tensor power method of [Anandkumar et al., 2014]). Although the results and methods are quite technical, the paper is well-written, and the presentation makes the high-level takeaways of the results apparent to non-experts.

That being said, I have a few suggestions for how the authors could improve the manuscript. First, it would be helpful to give a more thorough comparison with [Wang et al. 2020], which has significant overlap with the current work. While a brief comparison is given in the paper, it would be useful to expand on this further.

Secondly, the authors' findings seem very similar in nature to the seminal work of [Saxe et al. 2013, "Exact solutions to the nonlinear dynamics of learning in deep linear neural networks"], wherein a deep linear neural network trained by gradient descent was shown to capture progressively larger singular values of a target matrix. Although the methods used are completely different from the current work, the final results exhibit strong qualitative similarities, especially considering the connection between singular values of a matrix and magnitudes of the components of an orthogonally decomposable tensor. If the authors see any merit in this connection, it would be useful to mention it to readers.

Finally, could the authors comment on the restriction to fourth-order tensors in the paper? While the restriction to orthogonally decomposable tensors seems to be necessary to attain the main results, it isn't clear why only fourth-order tensors were studied. It would be useful to readers to indicate in the manuscript whether or not the results could be easily generalized to higher-order tensors (as was the case in [Wang et al. 2020]).

**Time Spent Reviewing:**

3

---

> ### Author Response · Authors · 2021-08-10
> **Thank you so much for your positive review and valuable suggestions!**
>
> Thank you so much for your positive review and valuable suggestions! We are glad to answer your questions below.
>
> *“More thorough comparison with Wang et al. 2020:”*
>
> Wang et al. studied tensor decomposition with a general ground truth tensor and proved that gradient descent can minimize the loss to zero. But they did not characterize the training trajectory. In particular, their result only guarantees that the learned tensor $T$ is close to the ground truth $T^*$, but does not guarantee that the component in $T$ aligns with some ground truth component in $T^*$. We studied a more restricted setting (orthogonal ground truth tensor), but we fully characterized the training trajectory (tensor deflation process) and the learned tensor $T$ (every large component aligns with a ground truth component). We will add more comparison with Wang et al. in our later version.
>
> *“Saxe et al 2013:”*
>
> Thanks for pointing out this paper. We agree that the phenomenon shown in this paper is qualitatively similar to the tensor deflation process that we considered in our paper. They also showed that gradient flow will first recover the larger signal (singular values/components) and then the smaller ones. However, the problems considered in Saxe et al. are very different from the settings of our paper. The setting in our paper is more challenging and requires very different proof techniques. We will include this paper and add discussions in our later version.
>
> *“Restriction to fourth-order tensors?”*
>
> For simplicity of the presentation, we have restricted our setting to fourth-order tensor decomposition. But our result can indeed be easily generalized to higher order tensors. We will mention this in the later version.

---

> > ### Comment · Reviewer_b9Va · 2021-08-16
> > **Thanks!**
> >
> > Thanks for those helpful responses, that addresses everything I had asked.

---

### Decision · Program_Chairs · 2021-09-27

**Decision:**

Accept (Poster)

**Comment:**

All the reviewers reached the consensus that the paper makes a valuable interesting contribution and should be accepted to the conference.  In the beginning of the discussion period, some concerns were raised about the practical utility of the results but after discussion among reviewers, the reviewers all agreed that the contribution is a nice technical addition to the growing literature on implicit regularization in factorization method.